https://doi.org/10.1038/s41467-020-20703-1　　**OPEN**

# Recruitment of upper-limb motoneurons with epidural electrical stimulation of the cervical spinal cord

Nathan Greiner[1,2✉], Beatrice Barra[2], Giuseppe Schiavone [3], Henri Lorach[1,4], Nicholas James[1], Sara Conti[2], Melanie Kaeser[2], Florian Fallegger[3], Simon Borgognon[1,2], Stéphanie Lacour [3], Jocelyne Bloch[4,5], Grégoire Courtine [1,4,5] & Marco Capogrosso [2,6,7✉]

Epidural electrical stimulation (EES) of lumbosacral sensorimotor circuits improves leg motor control in animals and humans with spinal cord injury (SCI). Upper-limb motor control involves similar circuits, located in the cervical spinal cord, suggesting that EES could also improve arm and hand movements after quadriplegia. However, the ability of cervical EES to selectively modulate specific upper-limb motor nuclei remains unclear. Here, we combined a computational model of the cervical spinal cord with experiments in macaque monkeys to explore the mechanisms of upper-limb motoneuron recruitment with EES and characterize the selectivity of cervical interfaces. We show that lateral electrodes produce a segmental recruitment of arm motoneurons mediated by the direct activation of sensory afferents, and that muscle responses to EES are modulated during movement. Intraoperative recordings suggested similar properties in humans at rest. These modelling and experimental results can be applied for the development of neurotechnologies designed for the improvement of arm and hand control in humans with quadriplegia.

---

[1] Center for Neuroprosthetics and Brain Mind Institute, School of Life Sciences, École Polytechnique Fédérale de Lausanne (EPFL), Geneva, Switzerland. [2] Department of Neuroscience and Movement Science, Faculty of Science and Medicine, University of Fribourg, Fribourg, Switzerland. [3] Bertarelli Foundation Chair in Neuroprosthetic Technology, Laboratory for Soft Bioelectronics Interface, Institute of Microengineering, Institute of Bioengineering, Centre for Neuroprosthetics, Ecole Polytechnique Fédérale de Lausanne, Geneva, Switzerland. [4] Defitech Center for Interventional Neurotherapies (NeuroRestore), Lausanne, Switzerland. [5] Department of Neurosurgery, Lausanne University Hospital (CHUV) and University of Lausanne (UNIL), Lausanne, Switzerland. [6] Department of Neurological Surgery, University of Pittsburgh, Pittsburgh, PA, USA. [7] Rehab and Neural Engineering Labs, University of Pittsburgh, Pittsburgh, PA, USA. ✉email: nathan.greiner@m4x.org; mcapo@pitt.edu

Two decades of preclinical and clinical studies have demonstrated that the delivery of epidural electrical stimulation (EES) to the lumbosacral spinal cord can reactivate spinal sensorimotor circuits after spinal cord injury (SCI)[1–9]. Computational and experimental studies conducted in animal models and humans[10–16] have brought evidence that EES applied over the lumbosacral spinal cord primarily engages large myelinated afferent fibers running in the dorsal roots and dorsal columns of the spinal cord. These fibers form synaptic connections with lumbosacral spinal interneurons and motoneurons, thereby constituting a gateway to the motor circuits controlling leg muscles[17,18].

Due to their branching morphology[19–21], the artificial recruitment of these fibers supplies synaptic inputs to multiple spinal segments. This divergence of inputs may limit the ability to modulate specific motor nuclei with EES, which could be particularly detrimental to applications aiming at restoring arm and hand function. Nonetheless, the distribution of sensory afferents in the dorsal roots can be exploited, by targeting individual roots, to direct the modulation exerted by EES toward specific motor nuclei. Indeed, evidence suggests that stimulating individual roots predominantly affects the motor nuclei located in the corresponding spinal segments, notably via group-Ia afferent fibers, which form monosynaptic excitatory connections with motoneurons[10,16,22]. In the context of paraplegia and locomotion, this principle was used to define stimulation protocols that target individual lumbosacral dorsal roots independently, with timings corresponding to the natural dynamics of the segmental motoneuronal activity underlying walking. Such spatiotemporal patterns of targeted EES induced immediate mitigation of lower-limb motor deficits in both animals and humans[3,8,9]. Furthermore, the trans-synaptic nature of the modulation exerted by EES on lumbosacral motor nuclei was identified as a central factor in these functional outcomes. Trans-synaptic recruitment increased the activity of naturally active motor nuclei without inducing parasitic activity in motor nuclei naturally silent during movement[18].

Similarly to the lumbosacral circuits, the cervical circuits controlling upper-limb muscles are organized with dorsal root afferent fibers innervating spinal motoneurons and interneurons, and with segmentally-clustered motor nuclei[23,24]. This suggests that the above mechanisms underlying the capacity of EES to support neural activity of the lumbosacral spinal segments may translate to the cervical segments. If so, EES of the cervical spinal cord could also facilitate arm and hand function in people with cervical SCI[25].

Here, we investigated the mechanisms underlying the recruitment of arm and hand motoneurons with EES of the cervical spinal cord and we inferred the properties that epidural implants should have to direct the modulation exerted by EES toward specific upper-limb motor nuclei. To this end, we implemented a detailed computational model capable of estimating the recruitment of various populations of cervical nerve fibers and neurons in response to single pulses of EES. Numerical simulations suggested that cervical EES primarily recruits large myelinated afferent fibers in the dorsal roots and in the dorsal columns and is unlikely to recruit motor axons directly. The model also suggested that the selective recruitment of individual dorsal roots (and thus, to a lesser extent, of upper-limb motor nuclei) is achievable with monopolar epidural electrodes, but contingent on the precise mediolateral and rostro-caudal position of the electrodes. Electrophysiological experiments in monkeys and humans revealed that: (a) cervical EES recruits upper-limb motoneurons trans-synaptically; (b) lateral electrodes engage upper-limb motor nuclei with a rostro-caudal order that reflects their segmental innervation in the cervical spinal cord; and (c) the modulation of upper-limb muscle activity exerted by EES in behaving monkeys is movement-phase dependent.

These modeling and experimental results provide a conceptual framework to design novel EES technologies to improve upper-limb motor control after cervical SCI.

## Results

**Computational model of the cervical spinal cord.** We combined a volume conductor model of the cervical spinal cord with neurophysical models of nerve fibers and neurons to predict the recruitment of fibers and neurons during cervical EES. We used the finite element method to compute tridimensional electric potential distributions in the volume conductor[26] and we used the estimated distributions to simulate the electrical behavior of neurons and fibers and assess the occurrence of action potentials induced by electrical stimuli[10,27,28].

The volume conductor comprised the following compartments: grey and white matter, spinal roots, dural sac, dura mater, epidural tissue, vertebrae, electrode contacts, insulating electrode paddle, and a wrapping cylinder of conductive material. We developed a parametric geometrical model to describe these compartments and a corresponding software suite to generate numerical instances automatically from user specifications. We determined the geometrical parameters of the cervical spinal cord of macaque monkeys by combining the anatomical analysis of two preserved spines and computed tomography images (see "Methods").

Anatomical analysis (Fig. 1b) revealed consistent cross-section widths across the two analyzed spinal cords (largest segments were C6–C8, >7.5 mm). The length of the spinal segments exhibited a greater degree of variability albeit with a trend of decreasing length from rostral to caudal segments. We found that the trajectories of the dorsal and ventral roots from their entrance in the spinal cord to their exit from the spine through the intervertebral foramina were almost perpendicular to the rostro-caudal axis of the spine for spinal segments C2–C3. These trajectories became more and more downward oriented for the more caudal segments, exiting the spine at markedly more caudal levels than their segment of connection (Fig. 1a). Consequently, the T1 spinal segment was approximately located at the same rostro-caudal level as the C7 vertebra.

Each modelled tissue was assigned a specific electrical conductivity tensor. Importantly, we generated curvilinear coordinates in the white matter and along the dorsal and ventral roots in order to appropriately map their anisotropic tensor onto their local longitudinal and transversal directions (see "Methods").

We then elaborated realistic tridimensional trajectory models for group-Ia afferent fibers (Aα diameter class), group-II afferent fibers (Aβ diameter class), motoneurons with their efferent axons (Aα), dorsal spinocerebellar tract fibers (ST, Aβ), lateral corticospinal tract fibers (CST, Aβ) and dorsal column fibers (DC, Aβ) (Fig. 1c, d). Fibers from other tracts were not represented due to their depth in the spinal cord and their distance from the stimulation electrodes: a negligible recruitment could be predicted on the basis of previous studies[26,27]. The trajectory models were based on documented morphological analyses[20,21,29], and numerical instances were generated at random using stochastic parameters elaborated from these analyses. The diameters of the fibers and motoneurons were also randomized, and motoneurons included dendritic arborizations represented by binary trees of tapered cylinders also chosen at random (see "Methods"). Their electrical behavior in response to extracellular stimulation was emulated with neurophysical compartmental models[28,30,31] using NEURON v7.5[32].

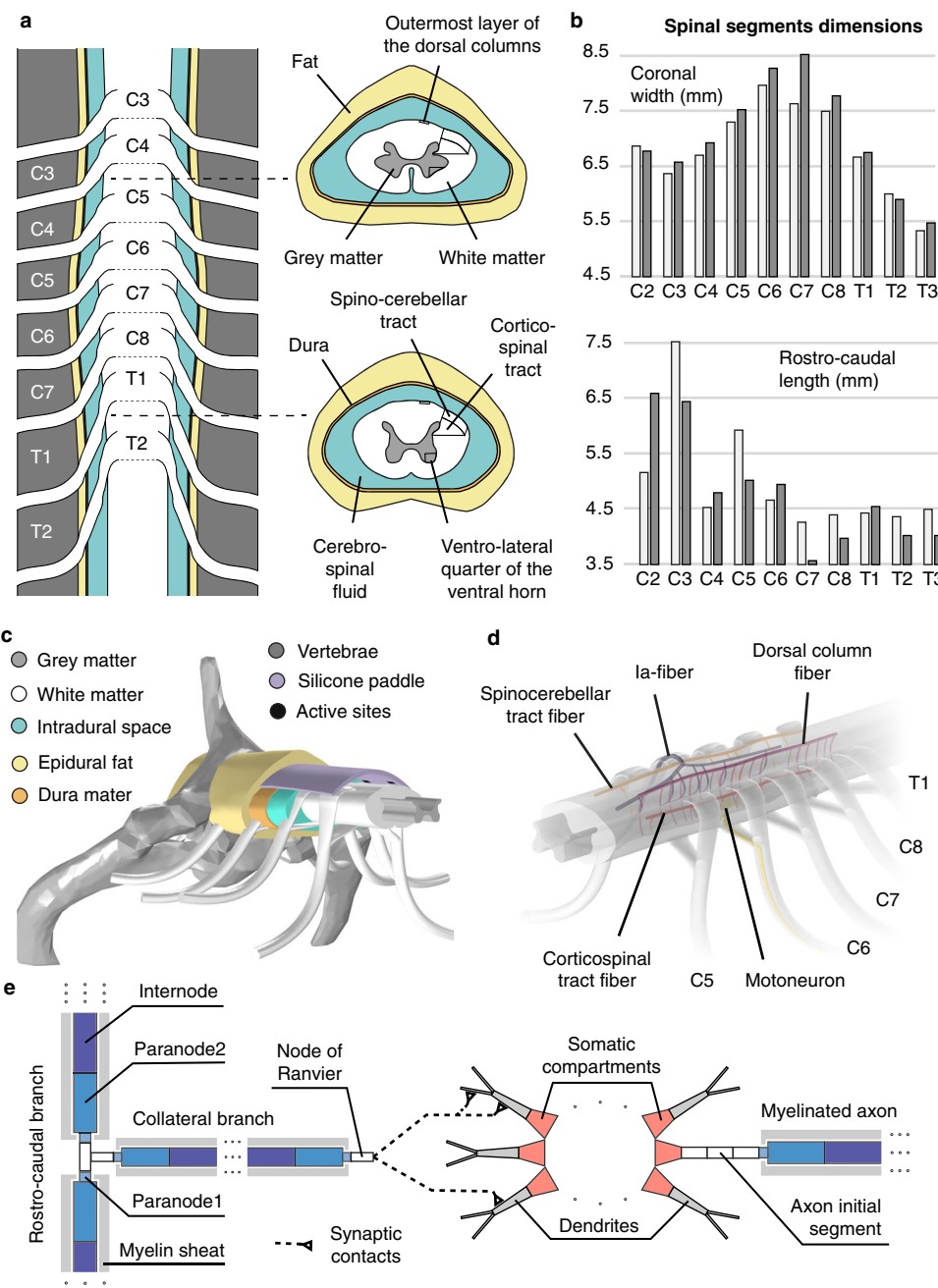

**Fig. 1 Morphology and computational model of the monkey cervical spinal cord. a** Macroscopic organization of the cervical spinal cord. Left: relationships between spinal segments, spinal roots, and vertebrae. Right: cross-sections at the C5 and T1 segmental levels showing the internal compartmentalization of the spinal cord. **b** Spinal segments dimensions. The two shades of gray indicate measurements coming from two different spinal cord dissections. **c** Tridimensional view of the volume conductor. **d** Trajectories of virtual nerve fibers and motoneurons. **e** Compartmentalization of myelinated nerve fibers and motoneurons used in neurophysical simulations[32,37,39].

**Computational analysis of the primary targets of EES applied to the cervical spinal cord.** We used this simulation environment to estimate the relative recruitment of nerve fibers and cells induced by electrical stimuli delivered from lateral and medial positions of the epidural space. We hypothesized that: (1) laterally-positioned electrodes would primarily recruit large myelinated fibers running in the nearest dorsal roots; (2) medially-located electrodes would primarily recruit dorsal column fibers; and (3) given the dimensions of the primate cervical spinal cord, recruitment of motor axons would not occur within the range of stimulation amplitudes that is necessary to recruit dorsal root afferent fibers.

Our simulations indicate that when placing electrodes laterally, facing a single dorsal root, the strongest generated electrical currents are found directly next to and inside the root (Fig. 2a). Furthermore, the generated currents do not substantially penetrate the spinal cord but predominantly flow in the less resistive cerebrospinal fluid. As a result, electric potentials are largest along the dorsal root fibers (DR-fibers, Fig. 2a), which are recruited at the lowest stimulation amplitudes (Fig. 2b). DR-fibers are followed by longitudinal fibers running in the spinocerebellar tract (ST-fibers), in the dorsal columns (DC-fibers), and in the lateral corticospinal tract (CST-fibers). Finally, the direct recruitment of motoneurons is predicted to remain null even at

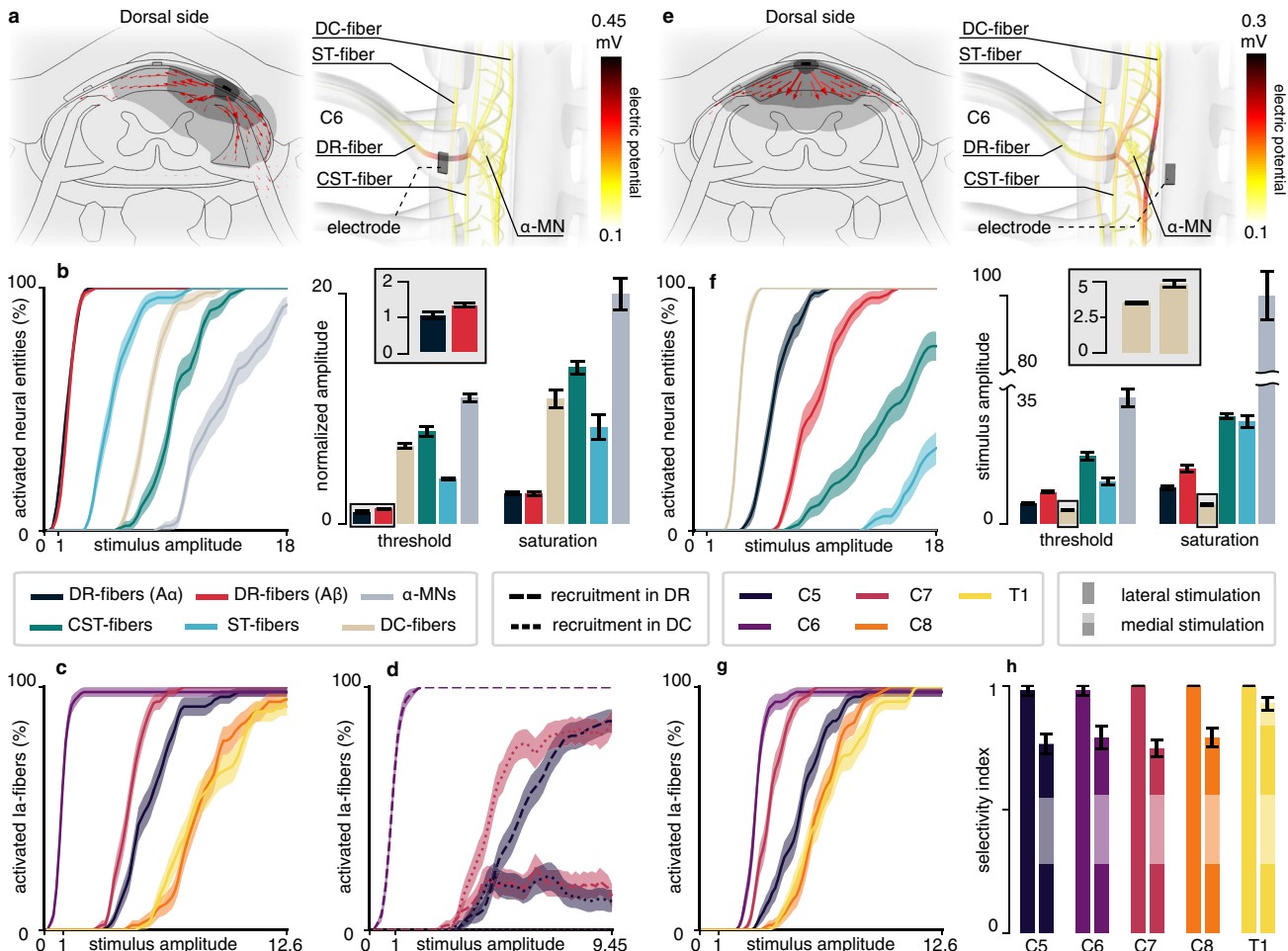

**Fig. 2 Computational analysis of the direct targets of EES of the monkey cervical spinal cord. a** Electric currents and potential distribution ($\phi$) generated by a lateral electrode contact at the C6 spinal level for a stimulation current of 1 µA estimated with the finite element method. Left: transversal cross-section cutting the electrode contact in two halves. Red arrows: current density vectors. Dark gray surface: $\phi \geq 2$ mV. Mild gray: $\phi \geq 1$ mV. Light gray: $\phi \geq 0.8$ mV. Right: $\phi$ along trajectories of virtual nerve fibers and motoneurons. DC: dorsal columns. ST: spinocerebellar tract. DR: dorsal root. CST: corticospinal tract. MN: motoneuron. **b** Direct recruitment of nerve fibers and motoneurons estimated from neurophysical simulations using the potential distribution of (**a**). Left: recruitment curves (dark blue and red curves are almost superimposed). Stimulus amplitudes are expressed as multiples of the threshold amplitude (10% recruitment) for DR-Aα-fibers. Right: threshold amplitudes and saturation amplitudes (90% recruitment) for the different neural entities, expressed as multiples of the threshold for DR-Aα-fibers. Inset: threshold for DR-Aα-fibers and DR-Aβ-fibers. **c** Simulated recruitment of Ia-fibers of individual dorsal roots using the potential distribution of (**a**). Amplitudes are expressed as multiples of the threshold for the Ia-fibers of the C6 root. **d** Recruitment for the C5, C6, and C7 roots split by recruitment localization. Amplitudes are expressed as multiples of the threshold for the Ia-fibers of the C6 root. **e** Same as (**a**) for a medially-positioned electrode contact. **f** Same as (**b**) with the potential distribution of (**e**). Inset: threshold and saturation amplitudes for the DC-fibers. Amplitudes are expressed as in (**b**). **g** Same as (**c**) with the potential distribution of (**e**). **h** Maximal selectivity indexes for each root using lateral or medial electrodes (see "Methods"). Recruitment curves (panels **b** [left], **c**, **d**, **f** [left], **g**): curves are made of 80 data points (except for **d**, 60 data points) consisting in the mean and standard deviation of the recruitment computed across 10,000 bootstrapped populations (see "Methods"). Lines and filled areas represent the moving average over three consecutive data points. Threshold, saturation, and selectivity bars and whiskers (panels **b** [right], **f** [right], **h**): mean ± standard deviation of the represented quantity computed across 10,000 bootstrapped populations (see "Methods").

amplitudes ~10 times higher than the threshold for Aα-DR-fibers and twice higher than the saturation amplitude for both Aα- and Aβ-DR-fibers and DC-fibers (Fig. 2b). Surprisingly, for lateral electrodes, the larger diameter of the Aα-DR-fibers compared to Aβ-DR-fibers does not seem to confer them a substantially lower excitation threshold, as could be expected[33,34] (see "Discussion").

We then sought to understand whether DR-fibers of adjacent roots are mostly recruited via the spread of the electric potential toward the adjacent roots or via the recruitment of the dorsal column projections of these fibers. To this end, we analyzed the recruitment order and action potential initiation sites of Ia-afferents of the C5, C6, and C7 roots in response to stimulation

targeting the C6 root. Our results indicate that Ia-afferents running in the targeted root (C6) are recruited at significantly lower stimulation amplitudes than those running in the adjacent or more distal roots (C5, C7, C8, and T1, Fig. 2c). In other words, our simulations suggest that near full and exclusive recruitment of individual roots can be achieved with individual monopolar lateral electrodes. Additionally, they indicate that action potentials in the fibers of the targeted root are initiated exclusively within their dorsal root branches (Fig. 2d). By contrast, action potential initiation sites of fibers belonging to the adjacent roots are partly found within their dorsal root branches and partly within their dorsal column projections. Specifically, recruitment

of caudal afferents (C7) is mostly induced in their dorsal columns branches, while the situation is reversed for the fibers rostral to the stimulation site (C5). This can be explained by the fact that caudal branches had smaller diameters compared to the rostral branches[19], increasing their excitation threshold[33].

Instead, with medial electrodes, electrical currents were predominantly directed toward the dorsal columns (Fig. 2e), resulting in DC-fibers exhibiting the lowest recruitment threshold (Fig. 2f), followed by Aα- and Aβ-fibers in the dorsal roots, and finally by ST-fibers. The threshold for CST-fibers is predicted to be ~5 times that of DC-fibers, while that for motor axons to be as high as ~9 times this value (Fig. 2f). In addition, contrary to lateral stimulation, the threshold of Aβ-DR-fibers to medial stimulation is significantly higher than that of Aα-DR-fibers (almost twice as high), as could be expected for an electrode position appreciably far from the fibers[33,34].

Furthermore, the ability to selectively recruit individual roots was predicted poorer with medial than with lateral electrodes (Fig. 2h, see "Methods"), and this poorer selectivity seems to be mainly due to a less efficient recruitment of the Ia-fibers of the targeted segment (Fig. 2c and g).

**Design of cervical spinal implants**. Our simulations indicate that, using lateral electrodes, it should be possible to recruit the large myelinated fibers of a single root without recruiting fibers of other tracts or roots but marginally. We thus hypothesized that targeting a single root may allow for the selective recruitment of motoneurons innervated by the Ia-fibers of that root. On the other hand, medial stimulation should lead to the activation of a wider range of motoneurons, notably from segments far from the rostro-caudal location of the stimulating electrode.

To obtain experimental evidence supporting this hypothesis, we designed and manufactured two types of multi-electrode spinal implants that were tailored to the dimensions of the spinal cord of macaque monkeys. The first design allowed the delivery of electrical stimuli from both medial and lateral locations of the epidural space. It comprised 2 columns of 5 stimulation active sites: one medial, facing the dorsal columns; and the other lateral, with one active site facing each of the C5 to T1 spinal roots (Fig. 3). The second design comprised a single, laterally-positioned column of 7 stimulation contacts, and one medial stimulation contact spanning three short-circuited active sites (Fig. S1g, h). Details on the fabrication and stability of the implants have been reported previously[35].

**Experimental recruitment of cervical motoneurons with epidural stimulation in macaque monkeys**. We conducted electrophysiological experiments in $n = 5$ anaesthetized monkeys. We implanted the spinal implants (Fig. 3, $n = 2$ design-1, and $n = 3$ design-2) and delivered single pulses of EES of increasing amplitudes from individual electrodes. In parallel, we recorded bipolar electromyographic (EMG) activity from $n = 8$ muscles of the left arm and hand (see "Methods").

We first analyzed the muscular recruitment induced by the lateral electrodes of the electrode arrays (available for the 5 animals). The results are presented in Fig. 4.

The obtained muscular recruitment profiles are not indicative of a recruitment pattern based on muscle synergies[36]. Instead, they indicate a segmental, rostro-caudal recruitment pattern which reflects the distribution of the upper-limb motor nuclei in the spinal cord (Fig. 3).

This rostro-caudal pattern is exemplified by the muscular recruitment curves from monkey Mk-Li, reported in Fig. 4b. Rostral stimulation (approximately at the C5–C6 level) induced activation predominantly of the deltoid, biceps, and extensor

carpi radialis muscles (Fig. 4b, top panel), all of which are innervated in the C5 and C6 segments. Caudal stimulation (C8–T1 level) mainly recruited the extensor digitorum communis, flexor digitorum profundis, flexor carpi radialis and abductor pollicis brevis muscles (Fig. 4b, bottom panel), innervated in the C8 and T1 segments (except for the flexor carpi radialis, mostly innervated in C7). And finally, stimulating from an intermediate rostro-caudal level (around C7 level) yielded a recruitment almost purely restricted to the triceps muscle (Fig. 4b, middle panel), innervated from C7–T1.

Moreover, as illustrated by the mean muscular activation levels of Fig. 4c and the maximal selectivity indexes of Fig. 4d (see "Methods"), this rostro-caudal recruitment pattern was overall obtained with every animal involved in the study. Its high correlation with the segmental distribution of upper-limb motor nuclei is illustrated by Fig. 5c (dark gray circles).

We then analyzed the muscular recruitment induced by the medial electrode contacts (available in $n = 2$ animals). The results are presented in Fig. 5.

Compared to lateral, medial stimulation induced a recruitment pattern strongly biased toward caudally-innervated muscles (Fig. 5b, d). In particular, medial stimulation, even rostral, failed to recruit the deltoid and biceps muscles. Moreover, when computing the correlation between muscle recruitment patterns and motor nuclei rostro-caudal distributions (Fig. 5c), lateral stimulation outperformed medial stimulation for all muscles except for the flexor carpi radialis. The maximum level of muscle recruitment (Fig. 5d) and maximum muscle recruitment specificity (Fig. 5e) were also higher with lateral compared to medial stimulation.

Finally, we measured the latency of the muscle responses evoked in caudally-innervated muscles following EES delivered either from rostral or caudal electrodes. We found that rostral stimulation induced responses with latencies that were consistently higher (Fig. 5f, g and S2). Specifically, the observed differences ranged between 0.2 and 1.1 ms (25th and 75th percentiles, respectively) with a median of 0.3 ms. Considering the length separating the rostral and caudal electrodes (~2.5 cm), this means that recruiting a caudally-innervated muscle from a rostral electrode elicited responses with an additional latency that matches action potentials propagation velocities ranging from 23 m/s (25th percentile) to 122 m/s (75th percentile) with a median of 71 m/s, which is compatible with diameter-class Aβ–Aα fibers[30]. Furthermore, a symmetrical situation was observed for the biceps muscle in which caudally-induced responses had higher latencies than rostrally-induced responses (consistently across the 2 animals, and with delays comparable with those reported above, Fig. S2).

**Computational analysis of the Ia-induced motoneuronal recruitment**. The previous computational and experimental findings suggested a predominantly Ia-mediated monosynaptic recruitment of motoneurons. We thus sought to use our simulation environment to assess the plausibility of this predominance.

We first assessed the influence of several parameters characterizing the Ia-to-motoneuron synaptic connectivity on the Ia-mediated excitation of motoneurons. These were: the number of Ia-fibers ($n_{Ias}$), Ia-to-motoneurons connectivity ratio ($r_{connec}$), mean number of synapses per Ia-motoneuron pair ($a_{contact}$), and synaptic conductance ($g_{syn}$).

Expectedly, we found that the somatic excitatory post-synaptic potential (EPSP) induced in a motoneuron is determined by the product $n_{syn} \times g_{syn}$ between the total number of synapses it receives ($n_{syn}$) and the synaptic conductance ($g_{syn}$) (Fig. 6a). This

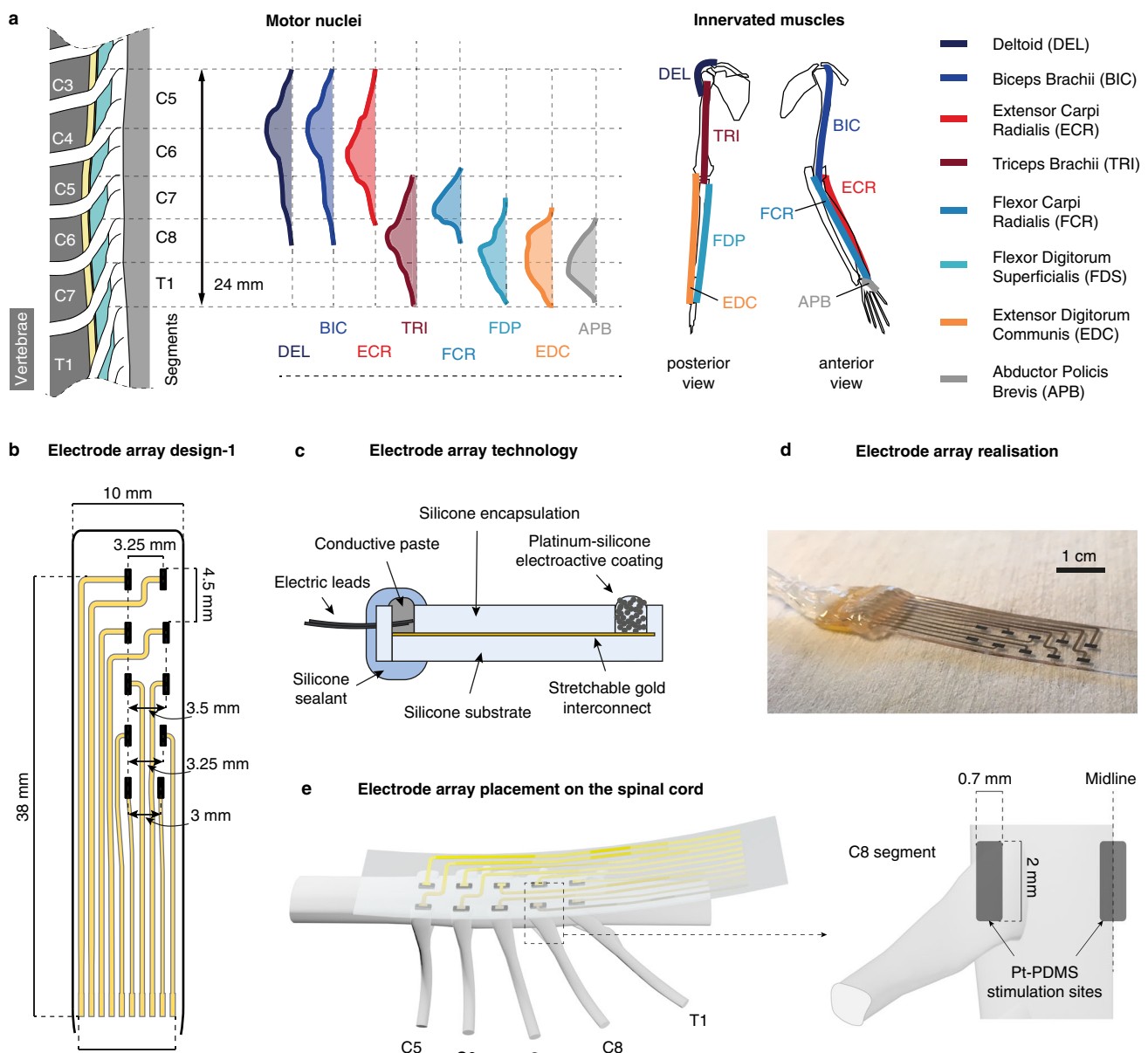

**Fig. 3 Organization of the monkey cervical spinal cord and soft electrode array tailored to the epidural space of the cervical spinal cord. a** Distribution of the motor nuclei of 8 upper-limb muscles in the monkey cervical spinal cord[27] and skeletal positions of these upper-limb muscles. **b** Layout of a custom electrode array with 5 lateral and 5 medial electrode contacts (design-1) tailored to the monkey cervical spinal cord. **c** Cross-section diagram of a soft electrode array. **d** Photograph of a fabricated soft electrode array. Scale bar: 1 cm. **e** Placement of the electrode array relative to the cervical spinal cord. Lateral electrode contacts were made to face individual dorsal roots while medial contacts were made to be along the midline of the dorsal columns. *Pt-PDMS:* platinum-polydimethylsiloxane.

suggested that the Ia-excitability of a motor nucleus innervated by a population of $n_{Ias}$ Ia-afferents can be characterized by the product $n_{Ias} \times r_{connec} \times a_{contact} \times g_{syn}$, since in this case, on average, $n_{syn} = n_{Ias} \times r_{connec} \times a_{contact}$ for each motoneuron.

Keeping constant the mean number of synapses per Ia-motoneuron pair ($a_{contact}$), we found that the number of Ia-afferents ($n_{Ias}$) that needs to be recruited to excite a given motor nucleus is determined by the product $r_{connec} \times g_{syn}$ (Fig. 6b). For instance, using $a_{contact} = 9.6^{19}$, $r_{connec} = 0.9^{21}$, and $g_{syn} = 5$ pS[37], we found that $56 \pm 3$ (mean ± standard deviation) Ia-fibers were required to induce the recruitment of 10% of their homonymous motoneurons, $93 \pm 7$ were required to induce the recruitment of 50%, and $175 \pm 9$ were required to induce the recruitment of 90%.

Next, again assuming a connectivity ratio ($r_{connec}$) of 0.9[21] and contact abundance ($a_{contact}$) of 9.6[19], and after estimating $n_{Ias}$ for the 8 upper-limb muscles retained in our study (see "Methods"), we determined for each of them the minimal $g_{syn}$ value enabling the recruitment of 100% of its motoneurons. The resulting $g_{syn}$ values ranged from 3.375 pS (triceps) to 28.5 pS (abductor pollicis brevis) (Supplementary Table 1) which are of the same order of magnitude (5 pS) and within the variability estimated experimentally by previous investigators[38]. Thus, our model was able to produce a purely Ia-mediated recruitment of motoneurons within a range of synaptic connectivity parameters coherent with experimental findings.

We then sought to evaluate whether a purely Ia-mediated recruitment is likely to occur during cervical EES. To that end, we

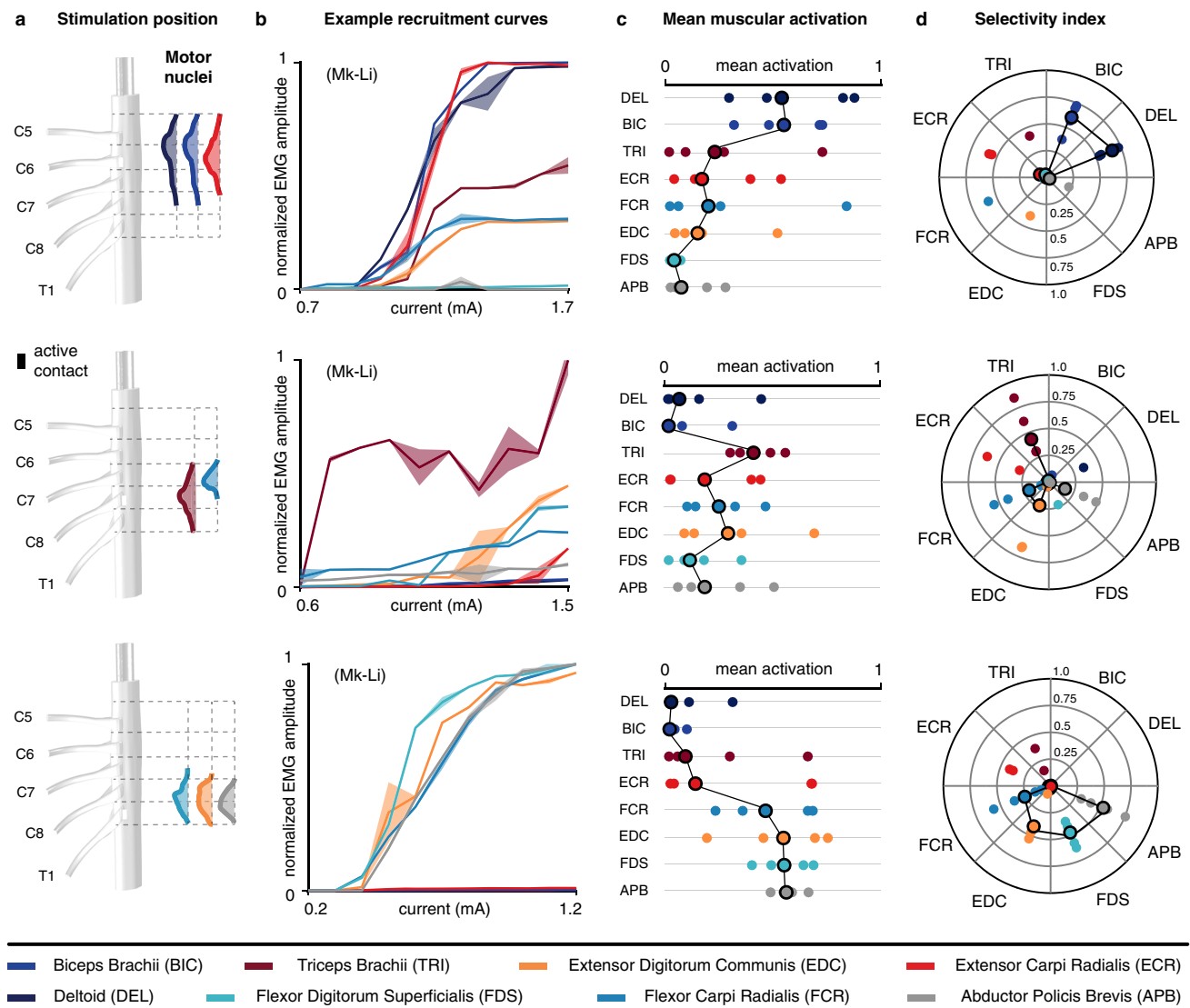

**Fig. 4 Muscular recruitment induced by laterally-positioned electrodes in the cervical spinal cord of monkeys. a** Approximate positions of the electrodes used to obtain the results in (**b**–**d**) and underlying motoneuronal distributions. Electrode contacts are magnified for better visualization (scale factor: 2). **b** Examples of muscular recruitment curves observed in monkey Mk-Li using one rostral, one intermediately rostral, and one caudal electrodes. Curves are made of 11 data points consisting of the mean and standard deviation of the normalized peak-to-peak EMG amplitude across four responses induced at the same stimulation current. **c** Mean muscular activations observed in 5 monkeys. One rostral, one intermediately rostral and one caudal electrodes were chosen for each animal, and the observed mean muscular activations (see "Methods") reported as individual bullets (for Mk-Li, the same active contacts as in (**b**) were used). **d** Maximal selectivity indexes (see "Methods") obtained for each muscle and each animal with the same electrodes as in (**c**). Circled bullets: medians across the five animals.

evaluated the monosynaptic recruitment of motoneurons resulting from the direct activation of Ia-fibers following lateral stimulation of the C6 root. We either used the $g_{syn}$ values of Supplementary Table 1 implying a uniform Ia-excitability across muscles (hypothesis H1), or a uniform average $g_{syn}$ value for all muscles, implying a higher excitability for muscles possessing higher numbers of Ia-fibers (hypothesis H2) (see "Methods"). Both hypotheses led to the trans-synaptic recruitment of the motor nuclei located in the targeted segment at stimulation amplitudes that were supra-threshold only for DR-fibers, i.e., for group-Ia and group-II fibers (Fig. 6d, f). Moreover, full (under H1) or almost full (under H2) monosynaptic recruitment could be reached before direct motor axonal recruitment began.

Finally, comparing these simulated recruitment curves with experimental recruitment curves did not allow to retain one hypothesis over the other (Fig. 6h). However, both implied that

the direct activation of Ia-afferents could induce the monosynaptic recruitment of a substantial number of motoneurons well before the direct recruitment of motor axons occurs. This provides a strong theoretical support to the hypothesis that cervical EES recruits motoneurons trans-synaptically (notably via Ia-fibers).

**Electrophysiological assessment of the trans-synaptic nature of the motoneuronal recruitment.** The previous results provide additional support to the hypothesis that the motoneuronal recruitment induced by lateralized EES is predominantly trans-synaptic and Ia-mediated. To gain experimental evidence, we conducted additional electrophysiological assessments. It is well established that repetitive stimulation of primary sensory afferents produces characteristic patterns of responses in the

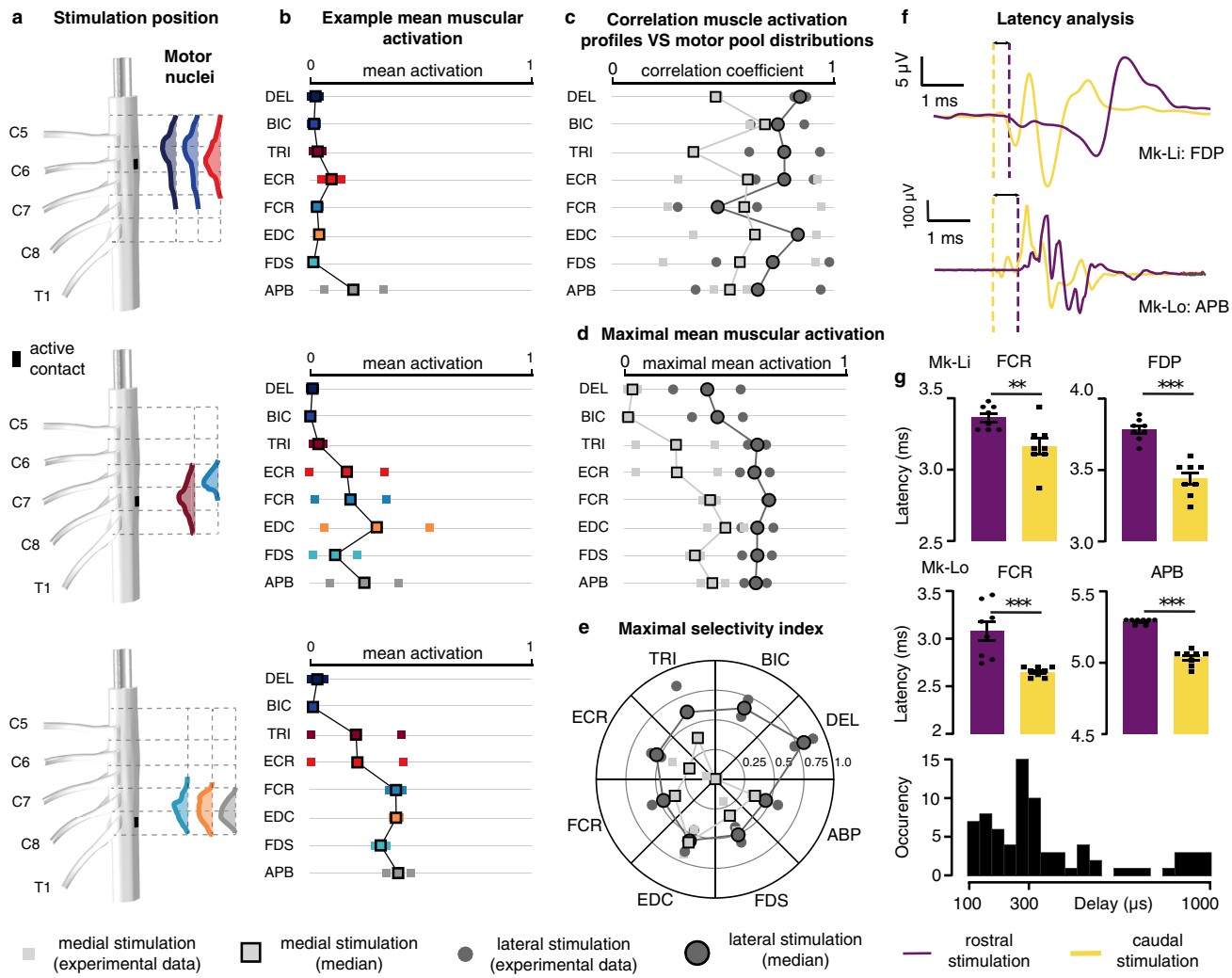

**Fig. 5 Comparison of muscular recruitment profiles induced by lateral and medial electrodes. a** Approximate positions of the medial electrodes used to obtain the results in (**b**) and underlying motoneuronal distributions. Electrode contacts are magnified for better visualization (scale factor = 2). **b** Mean muscular activations obtained with the medial electrodes at the same rostro-caudal levels than the lateral electrodes of Fig. 4c for the 2 monkeys implanted with the design-1 array (see "Methods"). DEL: deltoid, BIC: biceps brachii, TRI: triceps brachii, FCR: flexor carpi radialis, FDS: flexor digitorum superficialis, ECR: extensor carpi radialis, EDC: extensor digitorum communis, APB: abductor pollicis brevis. **c** Correlation coefficients between muscular recruitment profiles and motor nuclei distributions (see "Methods") for lateral and medial electrodes. **d** Maximal mean muscular activations achieved with lateral or with medial electrodes. **e** Maximal selectivity indexes (see "Methods") achieved with lateral or medial electrodes. In **b**-**e** data points (small squares/circles without borders) may be hidden by medians (large squares/circles with black borders). **f** Examples of muscular response latencies following stimulation from rostral medial or caudal medial electrodes. Top: muscular responses recorded in the flexor digitorum profundis of monkey Mk-Li. Bottom: muscular responses recorded in the abductor policis brevis of monkey Mk-Lo. Onsets of responses are indicated by the vertical dashed lines. **g** Statistical analysis of the differences in onset latencies between rostrally-induced responses and caudally-induced responses (two-sided unpaired t-tests, **p < 0.01; ***p < 0.001, see "Methods"). For each muscle, eight responses induced at amplitudes near motor threshold with one rostral (purple) and one caudal (yellow) electrodes were retained. Bottom histogram: distribution of the delay of rostrally-induced responses compared to caudally-induced responses.

homonymous muscles, such as frequency-dependent suppression[39] or alternation of different reflex responses[40]. We thus delivered supra-threshold stimuli using our spinal implants at multiple locations and frequencies of 10, 20, 50, and 100 Hz in four monkeys.

We observed various modulation modalities of the muscular responses in the recorded EMGs as consecutive stimuli were delivered: (1) attenuation of the responses' amplitude (Fig. 7b), (2) quasi-suppression of the responses (Fig. 7c), (3) alternation, often irregular, of two or three stereotypical responses (Fig. 7d), and (4) erratic adaptation to the pulse trains, wherein the first 5–10 responses didn't comply to the pattern respected by the subsequent responses of the train (Fig. 7e). Overall these behaviors were more likely to happen at high frequencies than

at low frequencies (Fig. 7h). Since adaptation of muscular responses should not occur following direct stimulation of motoneurons or motor axons in this range of stimulation frequencies, the previous modulations are all indicative of a mono- or polysynaptic motoneuronal recruitment. Absence of frequency-dependent modulation, such as reported in Fig. 7f was observed only rarely (Fig. 7h).

We then investigated whether the alternation patterns observed in EMG responses during high-frequency stimulation could indicate the involvement of central-pattern-generator-like spinal networks. For that, we examined the responses of agonist and antagonist muscles[41]. We found no cases in which agonist activity was correlated or anti-correlated with antagonist activity (Fig. 7g). However, we found that alternation patterns were more

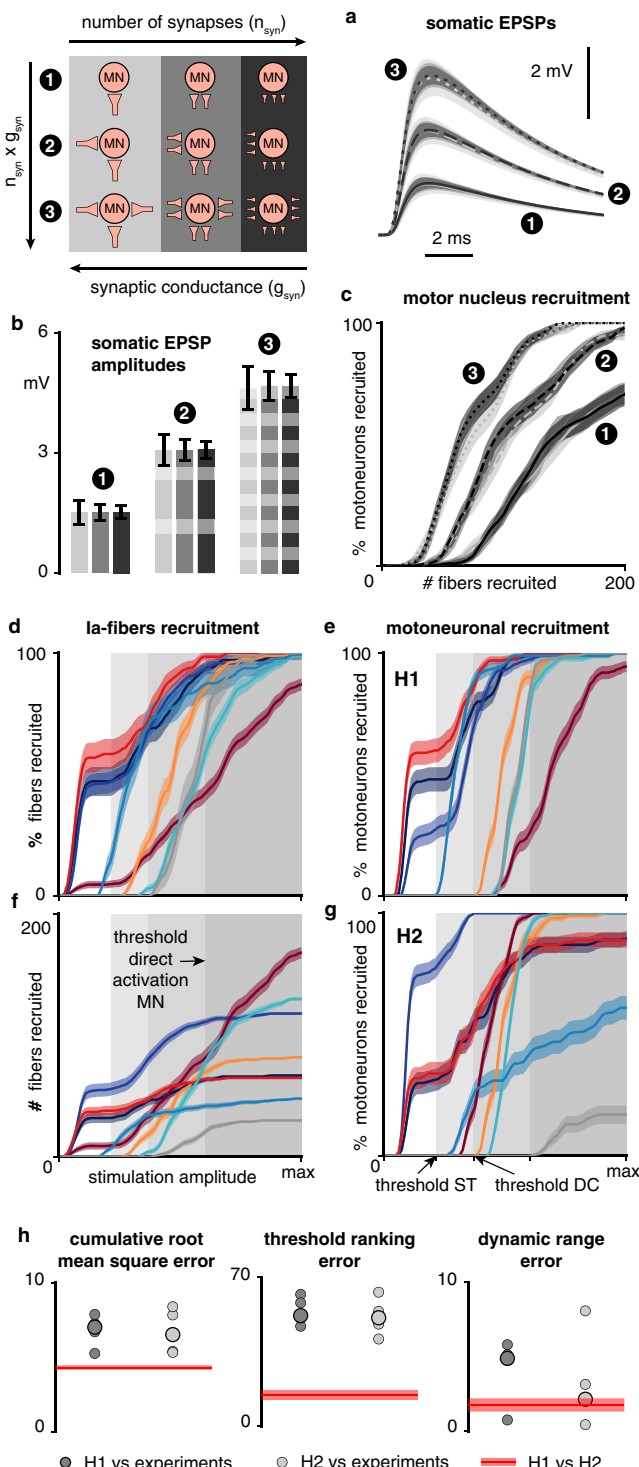

**Fig. 6 Computational analysis of the Ia-mediated recruitment of motoneurons. a** Time course of the somatic excitatory post-synaptic potential (EPSP) induced in a motoneuron model by various populations of synapses, for different population sizes $n_{syn}$ and synaptic conductances $g_{syn}$. Lines (solid, dashed and dotted) and filled areas represent the mean and standard deviation of the EPSPs obtained with 100 random synapse populations for each condition (9 conditions in total, see legend in the top left corner and "Methods"). **b** Maximal amplitudes of the EPSPs of (**a**). Bars and whiskers: mean ± standard deviation computed across the 100 synapse populations for each condition. **c** Recruitment of a motor nucleus as a function of the number of simultaneously activated fibers innervating this motor nucleus, for different connectivity ratios and synaptic conductances. Higher connectivity ratios are indicated by higher numbers of synapses in the legend. **d** Recruitment of Ia-fibers of specific muscles following electrical stimulation from a lateral contact at the C6 spinal level (Fig. 2a). **e** Monosynaptic recruitment of motoneurons resulting from the Ia-fiber recruitment shown in (**d**) using muscle-specific synaptic conductances (hypothesis H1, see Supplementary Table 1). **f** Same Ia-fiber recruitment as in (**d**) but represented in absolute numbers of recruited fibers. **g** Same as (**e**) but using a uniform synaptic conductance of 7.625 pS (hypothesis H2). **h** Comparison between experimental muscular recruitment curves and simulated motoneuronal recruitment curves with H1 or H2 (see "Methods"). Each bullet represents the comparison score for one animal (5 animals in total). Circled bullets indicate the medians across the 5 animals. Simulated Ia-fiber and motoneuron recruitment curves (**c–g**): curves are made of 80 data points (except for **c**, 40 data points) consisting in the mean and standard deviation of the recruitment computed across 10,000 bootstrapped populations (see "Methods"). Lines and filled areas represent the moving average over three consecutive data points.

the delivery of continuous EES at 50 Hz (typical stimulation frequency in therapeutic applications). We used a lateral rostral electrode targeting the C6–C7 roots and a lateral caudal electrode targeting the C8–T1 roots.

Continuous EES immediately and reliably evoked muscle responses in the upper-limb muscles (Fig. 8a, b). However, the amplitude of these stimulation-induced responses was modulated depending on the muscle and the movement phase considered. For example, responses in the triceps muscle were large during reaching and small during grasping and pulling (Fig. 8b). Accordingly, triceps' EMG energy was largely increased during reaching and only weakly during grasping and pulling (Fig. 8c).

The two contacts engaged distinct muscles (Fig. 8d). Consistent with its position, the C6–C7 contact mostly engaged the triceps. On the other hand, the C8–T1 contact mainly engaged the triceps and the flexor carpi radialis (FCR), which, again, was consistent with its position. However, this contact also engaged the biceps, which is more rostrally innervated (C5–C6 segments). Note that the biceps is a synergistic flexor muscle of the FCR, which may explain the emergence of the observed muscle responses (see "Discussion"). Overall the modulation was reduced during the grasping and pulling phase (Fig. 8d, right polar plot).

**Analysis of the recruitment of cervical motoneurons with epidural stimulation in humans**. We gained access to electro-physiological measurements acquired during clinical procedures in five human patients (see "Methods"). In three patients, a clinical paddle epidural array was implanted for the treatment of chronic neuropathic pain in the arms and hands, and thus approximately positioned between the spinal segments C6 and T1. In the two other patients, EES was delivered with a pen electrode at various rostro-caudal lateral positions during a spinal decompression procedure. Although the clinical procedures did not rigorously match the experimental procedures we performed

frequently elicited by rostral and intermediate contacts, while initial adaptation phenomena were more frequently induced by caudal contacts (Fig. 7i).

**Continuous stimulation of the cervical spinal cord during functional, three-dimensional arm movements**. To verify whether muscle activity and movement could be enhanced and modulated during a functional three-dimensional task, we trained one monkey (Mk-Sa) to freely reach, grasp and pull an object using a robotic framework that we designed for this purpose[42]. We investigated how the animal's arm EMGs were influenced by

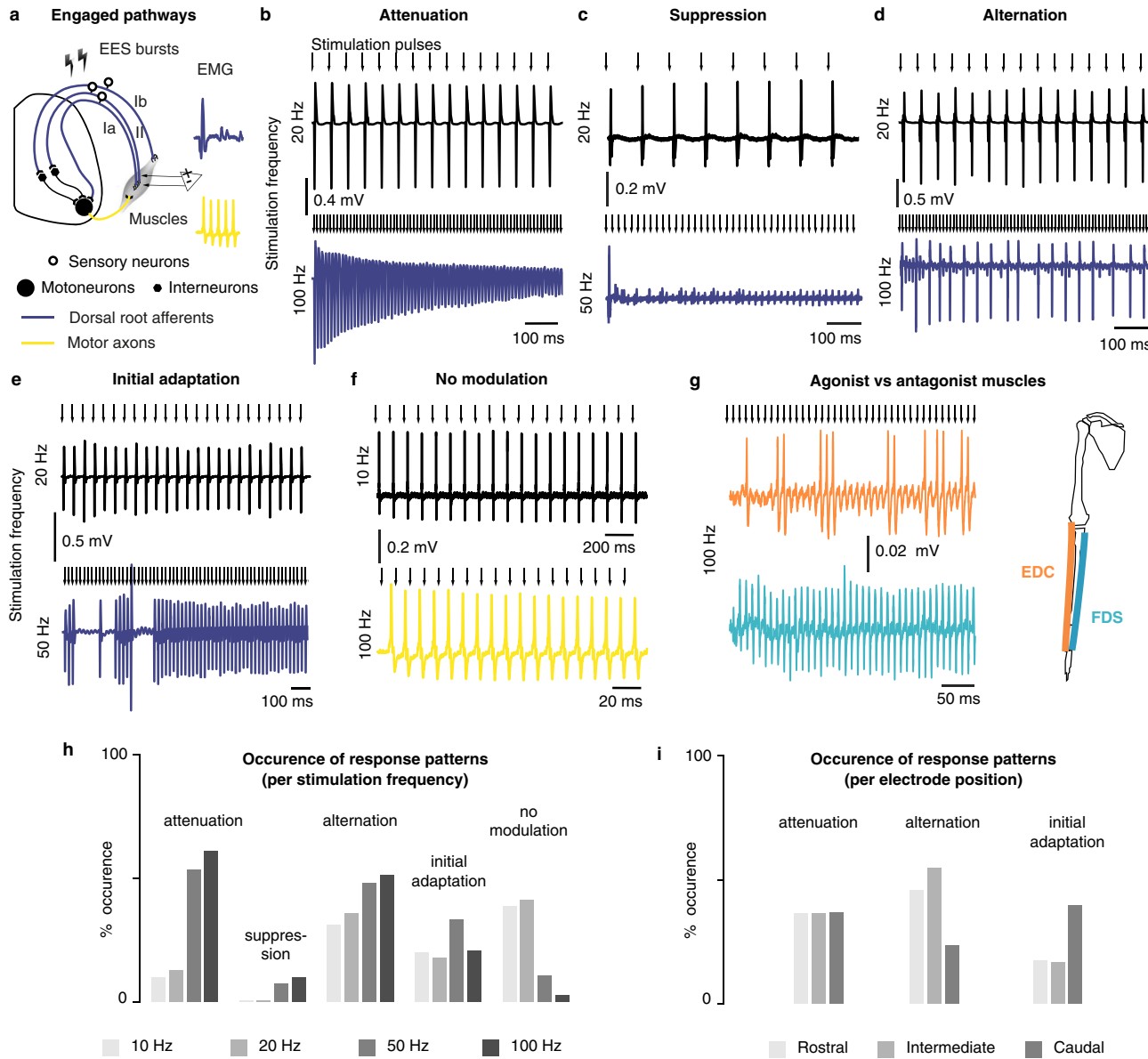

**Fig. 7 Patterns of muscular responses elicited during high-frequency stimulation of the cervical spinal cord of monkeys. a** Diagram of the presumed engaged pathways during high-frequency stimulation when muscular responses are modulated and unmodulated, respectively. **b–e** Examples of frequency-dependent modulation of muscular responses. In each panel, the top and bottom EMG traces were recorded in the same muscle and using the same stimulation amplitude (near motor threshold) but different frequencies. **f** Example of absence of frequency-dependent modulation. **g** Example of absence of correlation of frequency-dependent modulation between antagonist muscles (the top and bottom traces are simultaneous recordings of the extensor digitorum communis (EDC) and flexor digitorum superficialis (FDS) muscles of Mk-Lo during the same stimulation pulse train). **h** Frequency of occurrence of modulation patterns with respect to stimulation frequency. All the patterns recorded in all the muscles of the 4 animals in which high-frequency stimulation was tested were included in the analysis ($n = 80$ patterns at 10 Hz, $n = 39$ patterns at 20 Hz, $n = 75$ patterns at 50 Hz, $n = 72$ patterns at 100 Hz). **i** Same as (**h**), but with respect to electrode position ($n = 132$ patterns for rostral electrodes, $n = 66$ patterns for intermediate electrodes, and $n = 68$ patterns for caudal electrodes).

in monkeys, low-frequency (0.67–1 Hz) stimulation at increasing amplitudes from individual contacts was tested in all 5 patients, and supra-threshold stimulation at frequencies of 10, 20, 60, and 100 Hz was tested in the three patients implanted with a paddle array.

As observed in monkeys, the muscular recruitment profiles obtained in the 5 patients tended to reflect a segmental rostro-caudal innervation of the upper-limb muscles (Fig. 9c). However, the correlation was qualitatively less marked, and the stimulation specificity appeared poorer compared to monkeys. For instance, the deltoid and biceps could not be activated independently from

the triceps or from hand muscles, and reciprocally for the triceps. However, this limited measured performance might be due to the conditions of the clinical procedures as well as to the inappropriate dimensions of the paddle epidural array (see "Discussion").

Furthermore, high-frequency stimulation led to muscular responses modulated in a similar way than observed in monkeys. Specifically, we noted: (1) attenuation of the responses' amplitude, (2) alternation, often irregular, of two or three stereotypical responses, and (3) suppression of muscle responses within the first 3–5 responses (Fig. 9d). Also, as observed in monkeys,

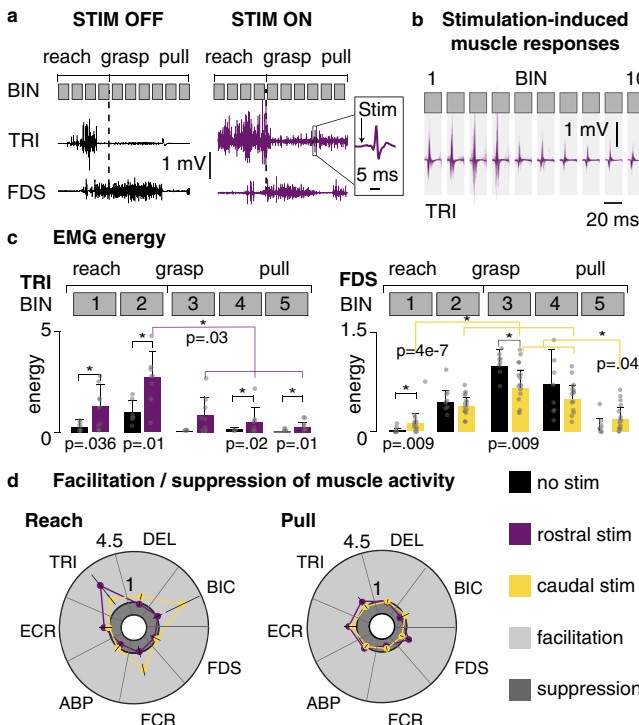

**Fig. 8 Effects of continuous stimulation during voluntary movement. a** Example of EMG activity during a reach, grasp and pull task. Left: without stimulation. Right: with stimulation. Top gray squares: division of time into 10 equal bins (used in **b**). Inset: typical stimulation-induced muscle response. TRI: triceps. FDS: flexor digitorum superficialis. **b** Overlay of stimulation-induced muscle responses in the triceps muscle during task execution. For multiple task executions (each divided in 10 bins as in **a**) and for each bin, the 20 ms windows of EMG data following each stimulation pulse occurring in the bin were extracted and overlaid (301 superimposed responses per bin). **c** EMG energy of the TRI (left) and FDS (right) muscles during task execution. Black: without stimulation. Purple: with rostral stimulation. Yellow: with caudal stimulation. Bars and whiskers: mean and standard deviation of the EMG energy across multiple task executions (rostral stim: 9 trials; caudal stim: 16 trials; baseline: 9 trials). For each trial, time was divided into 5 bins, and the EMG energy computed in each bin (see "Methods"). Statistics: Black: Wilcoxon Rank-Sum tests. Purple/yellow: Kruskal–Wallis tests analyzed post-hoc with Tukey–Kramer tests. Both: *$p < 0.05$ (see "Methods"). For TRI, energy is expressed in multiples of the baseline energy (i.e., without stimulation) during BIN 2. For FDS, energy is expressed in multiples of the baseline energy during BIN 3. **d** Mean facilitation indexes (dimensionless) of 7 upper-limb muscles (see "Methods"). Left: during reach. Right: during pull. Values greater than 1 indicate a facilitation effect; values smaller than 1 indicate suppression. DEL: deltoid, BIC: biceps, FCR: flexor carpi radialis, ECR: extensor carpi radialis, EDC: extensor digitorum communis, APB: abductor pollicis brevis.

modulation occurrences were overall more frequent at high frequencies (Fig. 9e). In fact, there were no instances of unmodulated responses following stimulation at 60 or 100 Hz.

## Discussion

In this study, we aimed to identify the main populations of nerve fibers recruited by EES applied to the cervical spinal cord, and to evaluate the influence of the anatomy of the cervical spinal cord on the ability of state-of-the-art epidural implants to engage specific upper-limb motor nuclei.

**Detailed modelling of EES of the cervical spinal cord.** The growing development of neuromodulation therapies has

accelerated the parallel development and use of computational models to guide the design of neurotechnologies[8,14,27,43,44]. The recent advent of large computational powers has enabled to increase the realism of the represented neurological systems, which might be critical to obtain accurate quantitative estimates directly usable in clinical applications. For instance, the explicit representation of the spinal roots, which has been missing in most computational models of spinal cord stimulation[10,13,14,16,27,45,46], should improve the accuracy of the performed simulations. In the present work, we elaborated algorithms to automatically build volumes representing the complex morphology of the roots based on the morphology and dimensions of the spinal cord and vertebral column. We could mesh these volumes with standard commercial software (COMSOL) and generate curvilinear coordinates following their longitudinal course, which allowed the local orientation of an anisotropic conductivity tensor aligned with virtual spinal root fibers. As a result, simulated currents inside the roots were effectively preferentially flowing along the direction of putative fibers. We also devised algorithms to generate realistic trajectories of nerve fibers and cells of various types in the spinal cord and in the spinal roots. In particular, we modelled dorsal root fibers with dorsal column projections and collateral branches projecting toward the grey matter[19–21,47], enabling a finer estimation of the segmental selectivity of the recruitment (see section "Stimulation specificity and lead design"). All our parametric models and corresponding software enabling the semi-automatic creation of hybrid neurophysical volume conductor models are freely distributed and hosted on public online repositories (https://bitbucket.org/ngreiner/fem_smc_ees/src/master/, https://bitbucket.org/ngreiner/biophy_smc_ees/src/master/).

Our hybrid model of cervical EES in monkeys led to an unexpected result: we found that the recruitment of Aβ-DR-fibers following lateral stimulation occurred at stimulation amplitudes only slightly higher than those necessary to recruit Aα-DR-fibers, albeit their average diameter was only two thirds that of the Aα-fibers. Comparatively, the difference in excitation threshold was much more pronounced for medial stimulation, closer to what is usually expected[33,34]. This result is likely due to the combination of (1) the close proximity between the lateral electrodes and the dorsal roots (Fig. 2a), which was a consequence of the fine reconstruction of the spinal cord model based on empirical morphological data, and (2) the small width of the electrodes. The latter were indeed approximately 700 μm wide, while the average distance between consecutive nodes of Ranvier of Aα-fibers and Aβ-fibers were 1400 μm and 900 μm, respectively. Thus, Aβ-fibers were more likely to possess a node directly below the electrode surface, which might have contributed to their greater excitability, albeit not as much as the larger diameters of the Aα-fibers. This effect might be less prominent in humans, where, given the larger size of the human spinal canal, the proximity between electrodes and dorsal roots should not be as close. However, the inability to selectively recruit the Aα dorsal root fibers with lateral electrodes would be detrimental to both sensory[48] and motor applications[49] of spinal cord stimulation, so we believe this point deserves further investigation.

**Trans-synaptic recruitment of motoneurons in the cervical spinal cord.** We have found that trains of electrical stimuli (10–100 Hz) modulated the stimulation-induced muscular responses in a frequency-dependent manner, suggesting the trans-synaptic nature of the underlying motoneuronal recruitment. Furthermore, we found that alternation patterns of muscle responses were more likely to occur with rostral (C4–C7 area) than with caudal (C8–T1) electrodes. Since propriospinal

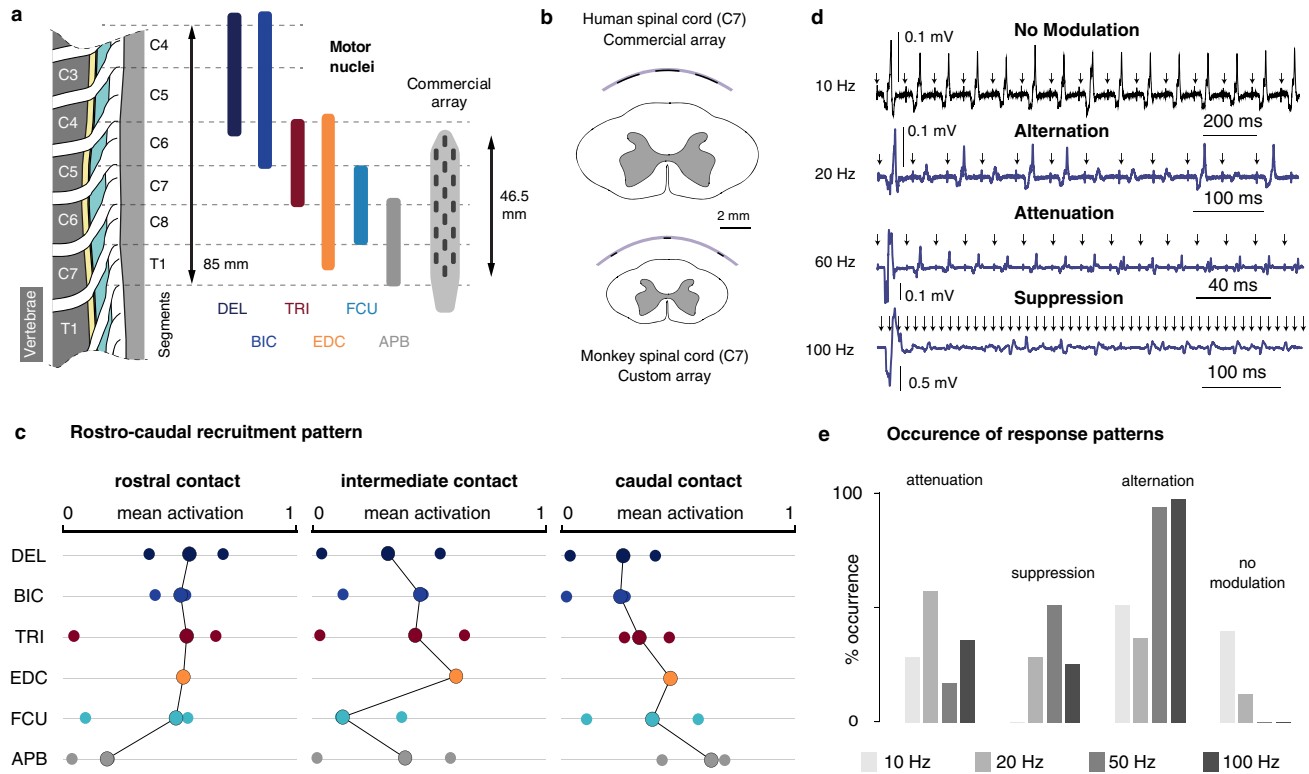

**Fig. 9 Muscular activity patterns evoked by EES of the cervical spinal cord in humans. a** Dimensions of the human cervical spinal segments, heuristic distribution of the motor nuclei of 6 upper-limb muscles[28], and sketch of the commercial paddle epidural electrode array used in 3 human patients. DEL: deltoid, BIC: biceps, TRI: triceps, EDC: extensor digitorum communis, FCU: flexor carpi ulnaris, APB: abductor pollicis brevis. **b** Comparison between the relative dimensions of the monkey cervical spinal cord and our custom implant, and the human cervical spinal cord and the commercial epidural implant of (**a**). **c** Mean muscular activations observed in 5 patients. One rostral, one caudal and one intermediate sites of stimulation (all lateral) were chosen for each patient, and the observed mean muscular activations (see "Methods") reported as individual bullets. For EDC, only 2 patients available. For FCU: only 3. Circled connected bullets: medians across the patients. **d** Examples of frequency-dependent modulation of muscular responses. The 4 EMG traces were obtained in the same muscle, using the same stimulation amplitude (near motor threshold) but different frequencies. Arrows: timestamps of stimulation pulses. **e** Frequency of occurrence of muscular response patterns during high-frequency stimulation. All the patterns recorded in all the muscles of the 3 subjects were included in the analysis ($n = 24$ patterns at 10 Hz, $n = 32$ patterns at 20 Hz, $n = 32$ patterns at 60 Hz, $n = 24$ patterns at 100 Hz).

circuits notably involved in the execution of skilled grasping are located in the C4–C6 segments[50,51], the emergence of alternation patterns exclusively with rostral electrodes might be explained by the engagement of these circuits, and again indicate the pre-motor nature of the elements directly activated by cervical EES. However, in the case of attenuation, alternation, or initial adap-tation of muscular responses, the fact that muscle activity is not fully suppressed prevents to completely discard the hypothesis that some direct recruitment of motor axons is occurring. Nonetheless, we believe that our results provide strong evidence that the evoked motor responses are predominantly of mono-/poly-synaptic origin, corroborating and extending previous findings[39].

Finally, the unlikelihood of direct motor axonal recruitment is also suggested by our simulation results. This would be attributable to the large dimensions of the monkey cervical spinal cord, which leads to relatively low electric potentials in the ventral roots compared to the dorsal roots and the dorsal columns. In turn, this suggests that direct motor axonal recruitment is even less likely in humans, whose spinal cords and vertebral canals are larger to those of monkeys, while fiber diameters are comparable in the two species and thus require similar potential distributions for external excitation. The observation that muscular responses induced by cervical EES in humans were also subject to frequency-dependent modulation at

least corroborates the hypothesis of a trans-synaptic recruitment of motoneurons also in humans.

**Stimulation specificity and lead design**. In our model, the detailed representation of the branching anatomy of dorsal root afferents allowed us to investigate their recruitment via their dorsal column projections. This information is important to determine optimal geometries and placements of epidural elec-trodes for maximizing the specificity with which motor nuclei can be modulated. Indeed, due to this branching anatomy, our simulations indicate that electrode mediolateral position is a key factor to achieve segmental selectivity (recruiting fibers of indi-vidual roots). With medial electrodes, dorsal column fibers are predicted to be more excitable than dorsal root fibers, which is due to the direct exposure of the large cervical dorsal columns to the generated currents[13,27]. Conversely, laterally-positioned electrodes should be able to recruit all the Ia-afferents of a sin-gle root at stimulation amplitudes that are subthreshold for every afferent coming from other roots. However, when increasing the stimulation amplitude, the recruitment of afferents of non-targeted roots is likely to occur via their dorsal column projec-tions. Thus, higher stimulation amplitudes may recruit fibers coming from roots far remote from the stimulation site, severely impacting the stimulation specificity.

Experimental findings were in line with our simulations. The recruitment profiles of arm and hand muscles induced by lateral contacts correlated well with the rostro-caudal distribution of their motor nuclei in the spinal cord. This is in accordance with a selective recruitment of individual dorsal roots under the assumptions that (1) Ia-afferents are distributed in the dorsal roots similarly to their homonymous motoneurons in the spinal segments, and (2) the motoneurons directly innervated by these afferents are predominantly homonymous[9,10,19,22,52]. Indeed, in this case, if fibers coming from different roots were recruited all at once, stimulation e.g. from rostral active sites would not necessarily induce recruitment of rostrally-innervated muscles.

By contrast, stimulation from medial contacts elicited markedly distinct muscular recruitment profiles. These profiles correlated less well with the spatial distributions of the motor nuclei, and lower levels of muscular activation were obtained compared to those obtained with lateral contacts. Moreover, the recruitment of caudally-innervated muscles induced by rostral contacts occurred with latencies that were significantly higher than those measured when stimulating with caudal contacts. The differences in latency were compatible with propagation delays through Aβ-Aα fibers, suggesting that rostrally-induced responses in caudally innervated muscles may result from the antidromic recruitment of dorsal column projections of caudal sensory afferents.

The validity of the two assumptions above is not firmly established. First, the distribution of Ia-afferents in the dorsal roots in monkeys and humans is unclear. Second, Ia-afferents are known to also form heteronymous connections[19,52]. Still, if heteronymous connections remain minority, it seems logical that Ia-afferents enter the spinal cord at the location where their homonymous motoneurons are located. On the same topic, the lack of quantitative information about heteronymous connections is the reason why we did not include them in our model.

Analysis of recruitment selectivity in the clinical data was limited by the nature of the clinical procedures, which were not experimentally-oriented. In particular, we could not assure that maximal EMG peak-to-peak amplitudes corresponded to maximal muscular activation levels. Therefore, the relative muscular activation levels of Fig. 9c should be regarded with caution. This notwithstanding, the data again tended to indicate a segmental rostro-caudal recruitment reflecting the distribution of motoneurons in the spinal cord, albeit not as markedly as in monkeys. However, comparing the dimensions of the clinical implant and the dimensions of the human cervical spinal cord[53] (Fig. 9b), it is likely that the electrode array was spanning only few spinal segments and that even the lateral electrodes were facing the dorsal columns. In fact, existing clinical implants are purposely designed to target dorsal column fibers[1]. We believe that new implants with lateralized electrodes are required to target individual dorsal roots and enable to engage specific motor nuclei.

Despite these limitations, the combination of our simulation and experimental results (1) indicate that cervical EES can engage upper-limb motor nuclei, (2) are in accordance with a predominantly trans-synaptic modulation of motoneurons, and (3) suggest that stimulation specificity is limited by the micro-anatomical organization of the dorsal roots. Specifically, the wide spatial separations between adjacent cervical dorsal roots (several millimeters), which is consistent across subjects, should allow robust selective recruitment of individual roots, but the intermingling of different fiber populations within these roots may limit the ability to modulate specific motor nuclei.

Similarly to EES of the lumbosacral spinal cord[54], the recruitment patterns that we obtained during single pulse experiments were explained by the rostro-caudal innervation of motor nuclei rather than by the recruitment of spinal circuits implementing muscle synergies[36]. Given the near perfect segmental separation between the biceps and triceps motor nuclei, a segmental modulation of motor nuclei seems sufficient to selectively promote elbow extension or flexion movements. However, given the overlapping innervation of forearm and hand muscles, selective facilitation of individual finger muscles is hardly conceivable with our type of spinal implant. By being closer to the targeted afferents, it may seem that subdural implants may improve the stimulation specificity, but a recent report[55] of the recruitment pattern induced by lateral subdural electrodes in monkeys indicated similar limitations to those outlined in the present work. Nevertheless, high-density electrode arrays and current steering techniques, potentially enabling the selective recruitment of structures as fine as individual dorsal rootlets, remain possible options to be explored to further improve the stimulation specificity.

**Impact of afferent-to-motoneuron connectivity**. The trans-synaptic nature of the engagement of motor nuclei during epidural stimulation implies that the connectivity between sensory afferents and motoneurons is of primary importance for the modulation exerted by EES. However, to date, even for the intensively studied Ia-to-motoneuron pathway[19–21,37,38,52,56], the properties characterizing the strength of the connection between Ia-afferents and motoneurons remain uncharacterized for muscles of monkeys and humans. For instance, while it is well-established that Ia-fibers can induce homonymous motoneuronal recruitment[57], variability in the Ia-to-motoneuron synaptic connectivity across muscles could influence this recruitment and make some muscles more prone to Ia-mediated activation than others.

Comparison between experimental data and simulation results obtained for different connectivity hypotheses did not allow to discriminate between the tested hypotheses. However, while we simulated the recruitment of motoneurons, the experimental data consisted in normalized EMG peak-to-peak amplitudes, which is only qualitatively related to the total number of recruited motoneurons, a priori disabling a direct comparison. We believe that the lack of thorough and detailed information on afferent-to-motoneuron connectivity currently limits the ability of computer simulations to produce reliable quantitative estimates of the modulation induced by EES and consequently also their use to design personalized clinical therapies.

**Cervical EES during voluntary movement**. We observed a segmental specificity in the recruitment of motoneurons following single pulses of low frequency cervical EES at rest. However, during movement execution, spinal circuits continuously receive supra-spinal, propriospinal, and natural sensory inputs which modify the integration of the artificial inputs induced by EES[12,18,54]. Thus, the direct activation of a given ensemble of pre-motor elements may modulate different motor nuclei during movement than at rest, in a task-dependent manner. This task- and phase-dependent modulation can actually improve the efficacy of EES. Indeed, EES should facilitate the activity of muscles naturally engaged during the execution of voluntary movements rather than imposing the recruitment of specific muscles[6,54]. These properties and therapeutic benefits have been demonstrated for lumbosacral EES, but remained unexplored for cervical EES.

We studied the modulation exerted by cervical EES during a typical reaching, grasping and pulling task performed by an intact monkey chronically implanted with our custom-made electrode array. Contrary to resting conditions, the modulation exerted by rostral and caudal contacts only partly reflected the rostro-caudal

innervation of the upper-limb muscles (Fig. 8d). Specifically, the activity of a rostrally-innervated muscle was consistently and significantly enhanced by the stimulation from the caudal contact. However, this muscle (biceps) was a flexor synergist of a caudally-innervated muscle (flexor carpi radialis, FCR) which was also engaged by the stimulation. This suggests that during the execution of a voluntary task, stimulating the primary afferents of upper-limb muscles (e.g., FCR) may enhance the motor activity of synergistic muscles, possibly via the contribution of heteronymous excitatory connections. Perhaps, at rest, these secondary pathways have only minor effects compared to homonymous monosynaptic pathways. An increase of coordinated excitatory inputs from circuits underlying muscle synergies, which are abundantly innervated by sensory afferents[58], might also be occurring. Finally, we found that the modulation of muscular activity was dependent on the movement phase and functionally-oriented (Fig. 8d), as is the case with lumbosacral EES[12,54].

**Potential clinical applications and limitations**. Our combined results suggest that, as in the case of the lower-limb[1–3,5,8], given the limited selectivity in the recruitment of specific upper-limb motor-nuclei, understanding the interplay of residual descending commands with propriospinal inputs, sensory feedback and the integrative properties of cervical spinal circuits will be key for the development of clinical EES protocols to restore arm and hand function. For example, while EES alone might not be sufficient to generate whole-arm functional movements in people with complete arm paralysis, clinical results in people with motor complete lower-limb paralysis suggest that descending pathways spared by the injury may still be able to use EES to produce complex functional movements[1,6,59]. On the contrary, the assistive potential of EES may be limited in subjects with significant residual motor control and sensory perception, notably by the emergence of pain or discomfort during stimulation. Such undesirable side effects may indeed occur at the stimulation intensities required to obtain relevant motor effects. However, subjects with incomplete motor paralysis and residual sensory function of the lower-limb did not report significant pain or discomfort at EES intensities that were necessary to obtain robust leg movements[8,11].

**Conclusions**. By combining computer simulations and electrophysiology in monkeys and humans, we have provided evidence that EES applied to the cervical spinal cord recruits motoneurons trans-synaptically via the direct excitation of sensory afferents and following their segmental distribution in the cervical spinal cord. Our results indicate that lateral contacts are necessary to achieve this segmental selectivity and that current human electrode arrays, with their medially-placed contacts and their short lengths, are unfit to achieve these results. They also show that the modulation exerted by EES during movement is movement-phase-dependent and likely promotes upper-limb muscle synergies. We believe that these combined results establish a pathway for the development of neurotechnologies for the restoration of arm movements in people with cervical spinal cord injury.

## Methods
**Volume conductor model of the cervical spinal cord**. The volume conductor model was implemented in Matlab (Matlab, The Mathworks, Inc.), using COMSOL Multiphysics v5.2a (COMSOL, Burlington MA) to assemble and mesh the geometry, generate the curvilinear coordinates, and compute the electric potential distributions using the finite element method. Computer code is available at https://bitbucket.org/ngreiner/fem_smc_ees/src/master/.

*Geometry*. Dimensions of the cervical segments were measured from $n = 2$ preserved spinal cords during dissections performed at the University of Fribourg, Switzerland. Preserved vertebral columns were cleared from connective tissue and

the spinous processes removed. The dura mater was cut longitudinally and retracted laterally to expose the spinal cord with care to avoid damaging the spinal roots. Spinal roots were labeled according to the vertebral level at which they exited the spine and spinal segments were delimited as portions of the spinal cord extending from the caudal-most rootlet of one root to the caudal-most rootlet of the root directly rostral to it (Fig. 1a). The segments' rostro-caudal lengths and coronal widths (at mid-segment-height) were measured independently by 3 experimenters and the average values were retained.

We reconstructed cross-sectional contours of the grey matter (GM) and white matter (WM) from a spinal cord atlas[60] and scaled them using the previous measurements. We formed 3D volumes interpolating these 2D contours using FreeCAD (https://www.freecadweb.org/).

We used OsiriX (Pixmeo SARL) to reconstruct a 3D volume representing the cervical vertebrae from CT-scan images (see imaging section for details). To do so, we first processed the acquired images to correct for the bending of the animal neck during the CT-scan acquisitions using custom Matlab routines. We then used Blender (Blender Foundation) to split the reconstructed volume into individual vertebrae, and MeshLab (Visual Computing Laboratory) to smooth these vertebrae. In this process, their relative positions were preserved. The GM and WM were positioned with respect to the vertebrae such that the C6 spinal root was perpendicular to the rostro-caudal axis of the spine.

An algorithm similar to ref. [61] was employed to build the volumes representing the roots. These exited the spine through the intervertebral foramina and had elongated elliptic cross-sections representing rootlets bundles[62] upon entering the spinal cord.

To build the volumes representing the dural sac, dura mater, and epidural tissue, we extracted the inner contours of the vertebrae which we smoothed and symmetrized using custom Matlab routines. We then scaled them to match the coronal and sagittal widths respectively of the dural sac, dura mater and epidural tissue (determined according to the local spinal cord dimensions), and interpolated them using FreeCAD similarly to the GM and WM. The thickness of the dura mater was set to 0.15 mm. The dimensions chosen for the dural sac were such that both the dural sac and dura mater were completely enclosed in the vertebral canal and did not overlap with the vertebrae.

The epidural electrode array was represented by metallic active contacts embedded in a large 3D strip (termed 'paddle') in contact with the dura. The dimensions and layout of this array were similar to those of one of the electrode arrays used in the in-vivo experiments (design-1, Fig. 3). It displayed two parallel columns of 5 active contacts, one along the midline of the spinal cord, and one shifted by ~3mm on the left side. By design, each row of active contacts was approximately at the rostro-caudal level of one spinal segment among C5 to T1. The active surface of the contacts had a geometric area of $1.0 \times 0.5$ mm$^2$.

Finally, a large cylinder representing the tissues surrounding the spine wrapped all the previous volumes and allowed to apply boundary conditions on its surface $\partial\Omega$.

*Physics*. Each represented tissue was assigned with an electrical conductivity tensor. We used values previously used in literature[61] for the GM, WM and spinal roots, CSF, epidural tissue, electrode contacts, bone and surrounding saline bath (wrapping cylinder). Isotropic conductivities of 0.03 S/m[46] and 1.0e−13 S/m (typical conductivity for silicone rubbers) were respectively assigned to the dura mater and the electrode paddle.

We used the Curvilinear Coordinates node of COMSOL to generate curvilinear coordinates in the WM and spinal roots and orient their anisotropic conductivity tensor. Specifically, we used the diffusion method with the inlet set on the top surface of the WM, the outlet set on the surfaces at the tip of each root and the bottom surface of the WM, and the wall set on the remaining lateral surfaces.

The capacitive and inductive effects of the materials were neglected, and the quasi-static approximation was employed to compute the electric potential distribution using electrical stimulation[63–65]. These assumptions lead to condensing Maxwell's equations into Laplace's equation $\nabla \cdot (\sigma \nabla V)(\mathbf{x}) = 0 \forall \mathbf{x} \in \Omega$ where $V(\mathbf{x})$ and $\sigma(\mathbf{x})$ are the electric potential and conductivity tensor at any point $\mathbf{x} \in \Omega$, and $\Omega$ denotes the interior of the volume conductor.

We modeled the delivery of a unitary electric current through the active surface $S$ of a contact by imposing $(\mathbf{j}_2(\mathbf{x}) - \mathbf{j}_1(\mathbf{x})) \cdot \mathbf{n}_{1,2} = 1 \forall \mathbf{x} \in S$, where $\mathbf{j}_1$, $\mathbf{j}_2$, and $\mathbf{n}_{1,2}$ are respectively the current densities on each side of the surface, and the normal vector to the surface pointing from side 1 to side 2. The resulting potential distributions (expressed in volts) were then divided by $A_S$ (the area of $S$) and by $10^3$, and by considering that they were expressed in millivolts instead of volts, they thus corresponded to a total injected current of 1 μA.

We assigned a zero-current flux condition at the outer boundary $\partial\Omega$ ($\nabla V(\mathbf{x}) \cdot \mathbf{n}(\mathbf{x}) = 0, \forall \mathbf{x} \in \partial\Omega$), and we inserted a grounded point ($V_{\text{ground}}$) in the ventral region of the wrapping cylinder, playing the role of a virtual return electrode placed there.

This boundary value problem was numerically solved using the finite element method. To this end, the geometry was discretized into a tridimensional mesh of approximately 10 million tetrahedral elements which was denser where high electric potential gradients were expected (near the electrode contacts). The equations' linearity allowed to estimate the electric potential distributions resulting from arbitrary amounts of injected current as scaled versions of the corresponding unitary distributions.

**Neurophysical models of neural entities**. The neurophysical models of nerve fibers and cells were implemented in Python 3.7 (The Python Software Foundation), using NEURON v7.5[32] to solve the membrane potential dynamics. Computer code is available at https://bitbucket.org/ngreiner/biophy_smc_ees/src/master/.

*α-motoneurons (α-MNs)*. α-MNs were modelled with a multi-compartment soma, a realistic dendritic tree, an explicit axon initial segment, and a myelinated axon. Dendritic trees were derived from digital reconstructions of cat spinal α-MNs established by Culheim and colleagues[66] freely available from the open-access library NeuroMorpho.org (cell references NMO_00604 to NMO_00609). These specify the geometry of dendritic trees as binary trees of frusta (tapered cylinders) originating at the soma. The axon initial segments comprised 3 linearly-connected identical cylindrical compartments[30]. These were prolonged by a myelinated axon compartmentalized according to the MRG model specifications[28]. The somata were modelled with multiple interconnected frusta following the developments of McIntyre and Grill[30]. For a given α-MN, the soma included one frustum for each dendrite stem, and one for the axon initial segment. The dimensions of the frusta were adjusted to preserve the total area of the soma, imposed by the α-MN diameter. The α-MN diameters were sampled uniformly in the range [44 μm, 71 μm][67]. Lengths and diameters of their dendritic compartments were linearly scaled accordingly. Axon diameters were linearly scaled to fall in the range [10 μm, 18 μm]. This last range was an arbitrary estimation for the class Aα fibers in monkeys, which we assumed to be intermediary between that of humans[68] and that of rats[69]. The dimensions of axon and initial segment compartments were derived from the axon diameters using piecewise linear interpolants established from the data of Supplementary Table 1 of ref. [28] and Supplementary Table 1 of ref. [30].

The somata centers of α-MNs were uniformly distributed in the ventro-lateral quarter of the GM ventral horn of their host segments. Their dendritic trees, selected uniformly at random among the 6 available templates, were rotated (at random) around their somata centers and partially verticalized to ensure they stayed in the GM or exited it only by short extents. Motor axon trajectories were generated as cubic splines running through the appropriate ventral spinal roots. They kept fixed relative positions along their paths in the spinal roots, reflecting the fact that nerve fibers do not jump from side to sides in nerve bundles.

The neurophysical properties of α-MN compartments were the same as in ref. [30]. In addition, the FLUT compartments of the motor axons comprised a fast potassium channel as in ref. [31].

Numbers of motoneurons and their rostro-caudal distributions for specific motor nuclei were extracted from ref. [23]. For the deltoid (data unavailable) we used the same distribution as for the biceps following observations made in humans[24].

*Group-Ia fibers/dorsal root proprioceptive Aα fibers*. Group-Ia fibers were modelled as composed of a dorsal root branch bifurcating into one ascending branch and one descending branch in the dorsal columns[19] and a series of collateral branches. Collateral branches originated from the dorsal column branches and projected toward the motor nuclei in the GM[20,21]. Each branch was compartmentalized according to the MRG model specifications[28]. The last node of Ranvier of the dorsal root branch was connected by its extremity to the extremities of the initial nodes of the ascending and descending branches. The initial nodes of the collateral branches were connected to the center of their branching node on the dorsal column branches. The diameters of the dorsal root branches were distributed log-normally with mean $\mu = 14$ μm and standard deviation $\sigma = 3$ μm. The diameters of the ascending and descending branches were derived by ponderation by $\sqrt{4/5}$ and $\sqrt{1/5}$, respectively. These factors ensured that ascending branches had diameters twice as large as descending branches[19], and that the total cross-sectional area was preserved upon bifurcation. The number of collaterals $N_{cols}$ for a given Ia-fiber was estimated based on the data reported by ref. [20] as $N_{cols} = \text{round}(0.75 \times L) \pm \eta$ where $L$ denotes the cumulated rostro-caudal extent of the dorsal columns branches (in mm), $\eta$ obeys a Poisson distribution with parameter $\lambda = 0.7$, and the sign ± denotes an equiprobably positive or negative deviation. Their diameters were log-normally distributed with mean $\mu = 2.5$ μm and standard deviation $\sigma = $ μm[21]. The dimensions of the compartments of the fiber branches were derived from the branch diameters using piecewise linear interpolants established from the data of Supplementary Table 1 of ref. [28].

The neurophysical properties of Ia-fiber compartments were the same as in ref. [28]. In addition, the FLUT compartments contained a fast potassium channel as described in ref. [31] which was adjusted to a resting potential of −80 mV instead of −70 mV.

Ia-fibers headcounts of the upper-limb muscles were derived from ref. [70]. First, the spindle headcount $sp_i^{mon}$ of muscle $i$ was derived from the spindle headcount of the homologous human muscle $sp_i^{hum}$ as $sp_i^{mon} = sp_i^{hum} \sqrt[3]{m^{mon}/m^{hum}}$ where $m^{mon}$ and $m^{hum}$ are typical masses of monkey specimens and human individuals respectively. The group-Ia fiber headcounts were obtained by assuming a 1:1 ratio between Ia-fibers and muscle spindles[68].

Their distribution in the spinal roots was assumed to be identical to that of their homonymous α-MNs. The resulting distributions, assuming $m^{mon} = 3.5$ kg and $m^{hum} = 70$ kg, are reported in Supplementary Table 1.

*Dorsal root proprioceptive Aβ fibers*. Class Aβ dorsal root proprioceptive fibers were modelled identically to group-Ia/Aα-DR-fibers, but their diameters were log-normally distributed with mean $\mu = 9$ μm and standard deviation $\sigma = 2$ μm.

*Dorsal columns fibers/dorsal spinocerebellar tract fibers/Lateral corticospinal tract fibers*. These fibers were modelled similarly to DR-Aβ-fibers but they didn't possess a dorsal root branch. They possessed a rostro-caudal branch running in the dorsal columns, dorsal spinocerebellar tract and lateral corticospinal tract respectively (see cross-section sketches of Fig. 1a). They also possessed a series of collateral branches. The dorsal branches of the dorsal column fibers were restricted to the outermost layer of the dorsal columns, where likelihood of recruitment is highest[71]. Lateral corticospinal tract fibers were meant to represent the large-diameter axons directly connecting to spinal motoneurons and originating from large Layer V cells in the primary motor cortex. Smaller diameter fibers were not represented. Dorsal spinocerebellar tract fibers represented Aβ sensory fibers originating from caudal spinal segments.

*Extracellular stimulation*. In the experiments, we used asymmetric biphasic stimulation pulses (described below). These were designed to minimize the effect of the anodic phase (employed uniquely for charge-balance). Therefore, in the simulations, we modelled 200 μs-long monophasic square pulses corresponding to the cathodic phases of the experimental biphasic pulses. This was chosen since the anodic phase, being 4 times smaller than the cathodic phase, had minimal (if any) influence on the emergence of action potentials during the simulations.

These monophasic pulses were simulated by transiently driving the batteries of NEURON's extracellular mechanism to appropriate voltages for each modelled compartment. These were obtained by multiplying the values computed with the FEM and interpolated at the appropriate positions in the volume conductor model by the desired stimulation amplitude. The rise and fall of the voltage transients were linear, lasting 2μs.

*Synapses*. Synapses contacting α-MNs were modelled as transient conductances inserted in the membrane of dendritic compartments. The temporal profile of these transients was described by the same function for every synapse, namely $\alpha(t) = (t - t_{onset})_+ \times \exp(\frac{t - t_{onset}}{\tau})$ where $t_{onset}$ is the time onset of the transient (synapse-dependent), $\tau$ is the time-to-peak of the transient (synapse-independent), $t$ denotes the running time of the simulation, and $(t - t_{onset})_+$ is null if $t < t_{onset}$ and is equal to $t - t_{onset}$ otherwise. $\tau$ was set equal to 0.2 ms[37].

For synapses supplied by Ia-fibers, the delay between electrical stimulation and the onset of the conductance transient was the sum of the action potential propagation delay through the Ia-fiber and a log-normally distributed stochastic jitter with mean 1.0 ms and standard deviation 0.5 ms accounting for synaptic transmission. Otherwise, the delay reduced to the previous stochastic jitter.

The synaptic currents were implemented as $i_{syn}(t) = g_{syn}\alpha(t)V(t)$ where $g_{syn}$ is the maximum of the synaptic conductance transient and $V(t)$ denotes the membrane potential at the synapse location.

Synapses were distributed on the dendritic trees of the contacted α-MNs using the electrotonic-distance-based distribution reported in Figure 16 of ref. [19].

*Somatic EPSP analysis*. We used a motoneuron model with somatic diameter of 55 μm, 10 dendrites and a total membrane area of ~475,000 μm². We performed $3 \times 3 \times 100$ simulations where each series of 100 simulations was characterized by a $(n_{syn}, g_{syn})$ pair of values. These series were grouped by 3, where each group was characterized by a constant $n_{syn} \times g_{syn}$ product. *Group 1: $n_{syn} \times g_{syn} = 500$ pS* ($n_{syn} = 50$, $g_{syn} = 10$ pS/$n_{syn} = 100$, $g_{syn} = 5$ pS/$n_{syn} = 150$, $g_{syn} = 3.3$ pS). *Group 2: $n_{syn} \times g_{syn} = 1000$ pS* ($n_{syn} = 100$, $g_{syn} = 10$ pS / $n_{syn} = 200$, $g_{syn} = 5$ pS/ $n_{syn} = 300$, $g_{syn} = 3.3$ pS). *Group 3: $n_{syn} \times g_{syn} = 1500$ pS* ($n_{syn} = 150$, $g_{syn} = 10$ pS/$n_{syn} = 300$, $g_{syn} = 5$ pS/$n_{syn} = 450$, $g_{syn} = 3.3$ pS).

*Ia-mediated monosynaptic recruitment of motoneurons*. We assessed the recruitment of a population of 100 motoneurons following the monosynaptic excitation provided by increasing numbers of Ia-fibers (fixed increment of 5 fibers). We assumed a fixed contact abundance of 9.6 synapses per Ia-motoneuron pair, and used $3 \times 3$ different $(r_{connec}, g_{syn})$ combinations. They were respectively characterized by a product $r_{connec} \times g_{syn}$ of 3 pS ($r_{connec} = 0.3, g_{syn} = 10$ pS/$r_{connec} = 0.6, g_{syn} = 5$ pS/$r_{connec} = 0.9, g_{syn} = 3.3$ pS), 5 pS ($r_{connec} = 0.3, g_{syn} = 15$ pS/$r_{connec} = 0.6, g_{syn} = 7.5$ pS/$r_{connec} = 0.9, g_{syn} = 5$ pS) or 6.75 pS ($r_{connec} = 0.3, g_{syn} = 22.5$ pS/$r_{connec} = 0.6, g_{syn} = 12.25$ pS/$r_{connec} = 0.9, g_{syn} = 7.5$ pS). The actual number of synapses of a given Ia-motoneuron pair was drawn from a Poisson distribution, and the motoneurons contacted by a given Ia-fiber were drawn uniformly at random.

To evaluate the muscle-specific $g_{syn}$ values of Supplementary Table 1, we assumed a connectivity ratio of 0.9[21], a contact abundance of 9.6[19] and we set $n_{Ias}$ appropriately for each muscle (see "Group-Ia fibers" section). The $g_{syn}$ values were determined via a binary search with a resolution of 0.125 pS. The uniform $g_{syn}$ value used under H2 was set to 7.625 pS, which is the value estimated for the extensor digitorum communis muscle, possessing an average number of Ia-fibers among the 8 studied muscles.

*Recruitment curves*. Figure 2. Populations of $N = 50$ nerve fibers or motoneurons were simulated for stimulation with multiple electrode contacts and stimulus amplitudes. A nerve fiber or motoneuron was considered recruited when an action

potential was elicited and traveled along its entire length. Threshold and saturation amplitudes of a population were defined as inducing recruitments respectively of 10% and 90% of the population (Fig. 2b, f). Standard deviations were obtained using a bootstrapping approach (see "Statistics" section).

Figure 6. Same as above but populations of $N = 100$ motoneurons were simulated for each muscle, and the sizes of the Ia-fiber populations were muscle-specific (see Supplementary Table 1).

*Selectivity indices.* The selectivity indices of Fig. 2h were computed as

$$\text{SI}_S^{\text{lat/med}} = \max_{\text{amp}} \left\{ R_S\left(\text{amp}, E_S^{\text{lat/med}}\right) - \frac{1}{N_{\text{seg}} - 1} \sum_{S \neq S} R_{S'}\left(\text{amp}, E_S^{\text{lat/med}}\right) \right\}$$

where $R_S(\text{amp}, E_S^{\text{lat/med}})$ denotes the recruitment level (between 0 and 1) of the population of Ia-fibers of segment $S$ at stimulation amplitude amp for either the lateral or medial electrode contact located at the rostro-caudal level of segment $S$, and $N_{\text{seg}}$ denotes the number of segments ($N_{\text{seg}} = 5$).

**Fabrication of custom spinal implants.** The implant manufacturing follows the Silicone-on-Silicon process[35]. Implants were prepared on 4" silicon wafers in a class 100 cleanroom environment by using a combination of microfabrication processes adapted to soft materials. The devices are fabricated by processing two identical membranes of silicone elastomer (Polydimethylsiloxane, PDMS, Sylgard 184, Dow Corning), both ~220 μm-thick. These serve as substrate and encapsulation layers that are subsequently covalently bonded together. A stretchable microcracked thin-film metallization (Cr–Au stack, 5–35 nm)[72] is deposited on the substrate by thermal evaporation through a PolyEthylene Terephthalate (PET) mask laminated on the silicone to define the interconnect pattern with conductor tracks leading to different electrodes (Fig. S1a). The encapsulation is laser micromachined to pattern through-holes that serve as vias to access the interconnect at locations corresponding to separate electrodes and wiring pads (Fig. S1b). After the PDMS substrate and encapsulation are processed, they are assembled by covalent bonding following oxygen plasma activation, with the interconnect sandwiched between the two layers and the holes in the encapsulation serving as vias to the exposed gold interconnect (Fig. S1c). Next, the vias over the electrodes are filled with a soft platinum-PDMS composite material[73] that provides both high charge injection capacity and mechanical compliance to stretching (Fig. S1d). The implants are then cut to shape and released from the silicon wafer (Fig. S1e). Finally, the vias over the wiring pads are used to make electrical contacts between discrete wires and the gold tracks, using a silver-based conductive paste (Epotek H27D) as soft solder. The connections are then mechanically stabilized by applying a room temperature vulcanization sealant (one component silicone sealant 734, Dow Corning) over the pads (Fig. S1f). Chronic stability, functionality, and biocompatibility are reported in ref. [35].

*Electrochemical impedance spectroscopy (EIS) of the electrode arrays.* EIS measurements were taken in vitro by immersing the array under test in a beaker containing Phosphate Buffered Saline solution (Gibco PBS, pH 7.4, 1X), along with a platinum wire as counter electrode and a Ag/AgCl reference electrode (Metrohm, El. Ag/AgCl DJ RN SC: KCl). In this 3-electrode configuration, electrochemical impedance spectra were acquired at room temperature using a Gamry Instruments Reference 600 potentiostat (100 mV amplitude, 1–Hz–1–MHz frequency).

In vivo EIS measurements of implanted electrodes were taken with the same equipment and in the same configuration as in vitro measurements. Two separate needle electrodes were inserted percutaneously in the skin on either side of the spine to serve as counter and reference electrodes.

**Experimental procedures**

*Animals involved in the study.* Four adult female and one male *Macaca Fascicularis* monkeys were involved in the study. Animal identification and information are summarized in Supplementary Table 2. All procedures were carried out in accordance to the Guide for Care and Use of Laboratory Animals[74] and the principle of the 3Rs[75]. Protocols were approved by local veterinary authorities of the Canton of Fribourg (authorizations reported in Supplementary Table 2 for each animal) including the ethical assessment by the local (cantonal) Survey Committee on Animal Experimentation and acceptance by the Federal Veterinary Office (BVET, Bern, Switzerland).

Monkeys were housed in collective rooms designed according to European guidelines (45 m³ for maximum 5 animals). They had free access to water and were not food deprived. Environmental enrichment was provided in the form of food puzzles, toys, tree branches and devices to climb and hide.

*Imaging.* We performed MRI and CT-scans on all animals involved in the study. All procedures were performed at the Cantonal Hospital of Fribourg, Switzerland. The animals were sedated with a combination of ketamine and medetomidine and transported to the imaging facilities where anatomical T1-weighted and T2-weighted images were acquired with a 3 T General Electric scanner at a resolution of 0.7 mm. CT-scan procedures were analog to MRI with a spatial resolution of 0.5 mm.

*Surgical procedures.* We performed two types of surgical procedures, terminal in n=3 animals, and acute tests under deep anesthesia in $n = 2$ animals. Both procedures were performed under full anesthesia induced with midazolam (0.1 mg/kg) and ketamine (10 mg/kg, intramuscular injection) and maintained under continuous intravenous infusion of propofol (5 ml/kg/h) and fentanyl (0.2–1.7 ml/kg/h) using standard aseptic techniques. Animals involved in the terminal procedures were injected with pentobarbital (60 mg/kg) and euthanized at the end of the experiments following the protocols described in the authorizations mentioned in Supplementary Table 2.

During the surgical procedures, the monkeys were implanted with bipolar stainless-steel electrodes to record electromyographic signals (sampling rate = 24 kHz) from the following upper-limb muscles: deltoid (DEL), biceps brachii (BIC), triceps brachii (TRI), flexor carpi radialis (FCR), flexor digitorum superficialis (FDS), extensor carpi radialis (ECR), extensor digitorum communis (EDC), and abductor pollicis brevis (APB). In one monkey (Mk-Ca), the flexor carpi ulnaris (FCU) and extensor carpi ulnaris (ECU) were recorded in place of FCR and ECR, respectively. Details on muscle implantation can be found in ref. [54].

Laminectomies were performed at the T1/T2 and C3/C4 junctions to provide access to the cord and allow insertion of the spinal implants. The custom-made spinal implant was inserted into the epidural space and pulled with the help of a custom-made polyamide inserter. Electrophysiological testing was performed intraoperatively to adjust the position of the electrodes. Specifically, we verified that a single pulse of stimulation delivered through an intermediately rostral electrode induced motor responses in the triceps muscle. Detailed protocols for the implantation and placement of the spinal implant are provided in ref. [54].

*Acute electrophysiology.* All acute electrophysiology was performed under propofol sedation (continuous intravenous injection, 5 ml/kg/h) that minimizes the impact of anesthesia on the responses to spinal cord stimulation[76]. Trains of biphasic electrical pulses were delivered at low frequency (0.67 Hz) through a single active site at a time. We used charge-balanced, asymmetric, cathodic-phase-first square pulses[77], with a cathodic phase of 200 μs and a balanced anodic phase of 800 μs and 4 times lower amplitude. Stimulus waveform influences the properties of neural recruitment[30,78]: we chose a waveform minimizing the influence of the anodic component while ensuring charge-balance, as recommended for safety reasons.

Within a train, square pulses were grouped by 4 using the same stimulation amplitude, and the amplitude was increased by fixed increments for 11 groups[54]. The first stimulation amplitude was chosen to be the lowest amplitude eliciting a response in any of the recorded muscles, while the last amplitude was chosen by the experimenters upon consensual judgement that some muscle was maximally recruited, either by inspection of the EMG responses or by direct observation of the movements induced in the limbs of the animals.

Additional trains of stimuli were delivered from multiple contacts (at least 2 per animal) at motor threshold stimulation amplitudes and at frequencies ranging from 10 to 100 Hz to test for frequency-dependent modulation of muscular responses[39].

*Electrophysiology during behavior.* We trained one animal (Mk-Sa) to perform an unconstrained reach, grasp, and pull task using a robotic framework designed for the purpose[42]. We then delivered continuous trains of biphasic electrical pulses (see previous paragraph) at 50 Hz while the monkey was performing the task. The stimulation amplitude was chosen to be the lowest amplitude eliciting a response in any of the recorded muscles, observed by inspection of the EMG signals. Using this minimal amplitude, we avoided to disrupt the execution of the task by the animal. We also recorded the animal performing the task in absence of stimulation to acquire reference signals for comparisons.

*Human data.* Anonymized clinical data were obtained from the Centre Hospitalier Universitaire Vaudois (CHUV, Lausanne, Switzerland). The data were obtained during standard clinical practice and under the CHUV's general ethical approval for clinical procedures and use of anonymized clinical data for scientific purposes. The Swiss federal institute of the use of human data for scientific research (Swiss Ethics) validated the use of the anonymized human dataset. In 3 patients suffering from upper-arm neuropathic pain, current-controlled electrical stimulation was delivered from individual contacts of a Medtronic Specify™ 5–6–5 interface at frequencies ranging from 1 Hz to 100 Hz and amplitudes ranging from 0 to 5 mA to test for correct surgical placement of the implant.

In 2 patients requiring surgical laminectomies for spinal decompression, current-controlled stimulation at a frequency of 1 Hz and amplitudes ranging from 100 μA to 800 μA was delivered with a manual probe from various positions of the epidural space.

In all 5 patients, percutaneous needles were placed in various upper-limb muscles and EMG signals were recorded (sampling rate = 10 kHz) using a clinical monitoring interface (NIM Eclypse). Exported data were anonymized and provided to the authors with no reference to patient identity.

**Data analysis**

*Muscle recruitment and recruitment curves.* From the electromyographic recordings of low-frequency stimulation protocols (0.67 Hz), we extracted 50 ms-long snippets of data following each stimulation pulse. For each data snippet, we measured the peak-to-peak amplitude of the recorded signal, P2P(snippet), except for purely

noisy signals, for which a value of 0 V was retained (a signal was considered purely noisy when its maximal amplitude was smaller than 3 times the standard deviation of the recording channel baseline signal). Then, for each animal ($A$), electrode contact ($E$), muscle ($M$) and stimulation amplitude (amp), we computed the mean and standard deviation across the 4 P2P(snippet) values corresponding to the configuration ($A, E, M$, amp), respectively noted $\text{P2P}^A_{E,M}(\text{amp})$ and $\text{sP2P}^A_{E,M}(\text{amp})$. We defined the muscle recruitment $R^A_{E,M}(\text{amp})$ and associated standard deviation $\text{sR}^A_{E,M}(\text{amp})$ as

$$R^A_{E,M}(\text{amp}) = \frac{\text{P2P}^A_{E,M}(\text{amp})}{\max\limits_{E',\text{amp}'}\left\{\text{P2P}^A_{E',M}(\text{amp}')\right\}} \quad \text{sR}^A_{E,M}(\text{amp}) = \frac{\text{sP2P}^A_{E,M}(\text{amp})}{\max\limits_{E',\text{amp}'}\left\{\text{P2P}^A_{E',M}(\text{amp}')\right\}}$$

and thereby obtained normalized muscular recruitment curves (as represented in Fig. 4b).

*Mean muscle activation and selectivity index.* For a given animal ($A$) and electrode active contact ($E$), the mean activation of muscle $M$ was estimated as

$$\overline{R^A_{E,M}} = \frac{1}{N_{\text{amps}}} \sum_{\text{amp} \in \text{DR}} R^A_{E,M}(\text{amp})$$

where DR denotes the dynamic range of stimulation amplitudes of electrode contact $E$ and was defined as the range from threshold amplitude (defined as inducing a muscular recruitment higher than 10% in at least one muscle) to saturation amplitude (defined either as inducing a muscular recruitment higher than 90% in every muscle, or as the highest amplitude used). The selectivity index of muscle $M$ was computed as

$$\text{SI}^A_{E,M} = \max\limits_{\text{amp} \in \text{DR}}\left\{R^A_{E,M}(\text{amp}) - \frac{1}{N_{\text{musc}} - 1}\sum\limits_{M' \neq M} R^A_{E,M'}(\text{amp})\right\}$$

*Correlation between rostro-caudal muscular recruitment profile and motor nucleus distribution.* For each animal ($A$) implanted with the design-2 array, and each muscle ($M$), we computed the correlation coefficient between rostro-caudal muscular recruitment profile and motor nucleus distribution as:

$$\text{CC}^{A,S}_M = \frac{\left(R^A_{E^S_1,M}, R^A_{E^S_2,M}, R^A_{E^S_3,M}, R^A_{E^S_4,M}, R^A_{E^S_5,M}\right) \cdot \left(p_{1,M}, p_{2,M}, p_{3,M}, p_{4,M}, p_{5,M}\right)}{\left\| R^A_{E^S_1,M}, R^A_{E^S_2,M}, R^A_{E^S_3,M}, R^A_{E^S_4,M}, R^A_{E^S_5,M}\right\| \times \left\| p_{1,M}, p_{2,M}, p_{3,M}, p_{4,M}, p_{5,M}\right\|}$$

where $E^S_1, E^S_2, E^S_3, E^S_4, E^S_5$ denote the electrode contacts at the C5, C6, C7, C8, and T1 levels for each side ($S$) lateral or medial, respectively; and $p_{1,M}, p_{2,M}, p_{3,M}, p_{4,M}, p_{5,M}$ are the proportions of motoneurons of muscle $M$ in each of C5, C6, C7, C8, and T1.

*Comparison between simulated and experimental recruitment curves. Representative subset of experimental recruitment curves:* for each animal ($n = 5$), we selected 3 recruitment curves respectively obtained with a rostral, a caudal, and an intermediately-rostral lateral active contact.

*Simulated recruitment curves:* the simulated motoneuronal recruitment curves corresponding to the previous 3 active contacts and obtained with each of the two synaptic conductance hypotheses (H1 and H2) were retained for comparison.

*Dynamic range:* here, the saturation amplitude of the simulated and experimental recruitment curves was defined as the amplitude at which any motor pool/muscle was recruited > 90%, incremented by two amplitude steps.

*Cumulative root mean square error (CRMSE):* we restricted the simulated and experimental recruitment curves to their respective dynamic ranges and resampled them over a normalized amplitude vector. The root mean square error (RMSE) between the simulated curve with hypothesis $H$, electrode $E$ and muscle $M$, and the corresponding experimental curve for animal $A$ was estimated as

$$\text{RMSE}_{H,A}(E,M) = \sqrt{\frac{1}{N_{\text{amps}}} \sum\limits_{\text{amp} \in \text{DR}} \left(R^H_{E,M}(\text{amp}) - R^A_{E,M}(\text{amp})\right)^2}$$

The CRME between the simulated recruitment curves with hypothesis $H$ and the experimental curves of animal $A$ was defined as

$$\text{CRMSE}_{H,A} = \sum\limits_{E,M} \text{RMSE}_{H,A}(E,M)$$

*Threshold ranking error (TRE):* we extracted from the simulated and experimental recruitment curves the recruitment order of the different muscles in the form of lists $[M_1, M_2, \dots, M_8]^{A/H}_E$ where each $M_i$ denotes a distinct muscle. We then compared the simulated recruitment order under hypothesis $H$ with the experimental recruitment order obtained with animal $A$ by counting the number of inversions (in the sense of permutations) between the lists $[M_1, M_2, \dots, M_8]^H_E$ and $[M_1, M_2, \dots, M_8]^A_E$, and summing the obtained values for each of the 3-electrode contact positions.

*Dynamic range error (DRE):* we extracted from the simulated and experimental recruitment curves the normalized length of the dynamic range $L^{A/H}_E$ as the length of the absolute dynamic range divided by its first amplitude. The dynamic range error between the simulated recruitment curves under hypothesis $H$ and the

experimental recruitment curves of animal $A$ was computed as

$$\text{DRE}_{H,A} = \sum\limits_E \left(\frac{L^A_E}{L^H_E} + \frac{L^H_E}{L^A_E} - 2\right)$$

The two sets of simulated recruitment curves were compared with respect to the same 3 metrics. Standard deviations were obtained using a bootstrapping approach (see Statistics section).

*Analysis of frequency-dependent modulation of muscular responses.* The recordings of high-frequency (10, 20, 50, and 100 Hz) stimulation protocols from all animals, muscles and electrode contacts were visually inspected and characterized according to 5 criteria: "attenuation", "suppression", "alternation", "initial adaptation", and "no modulation". "Attenuation" and "suppression" applied to those patterns of muscular responses where the amplitude of subsequent responses was progressively reduced and abruptly canceled, respectively (Fig. 7b, c). 'Alternation' applied to patterns where subsequent responses alternated between low amplitude and high amplitude[40] (Fig. 7d). 'Initial adaptation' applied to patterns in which the evoked responses became steady only after several (5–20) initial pulses (Fig. 7e). Finally, 'no modulation' applied to patterns in which no observable modulation or change in the evoked responses occurred (Fig. 7f). Multiple criteria could apply to a single pattern, except for the 'no modulation' criteria.

*Analysis of latencies of muscular responses.* Latencies of evoked muscular responses were measured as the duration between onset of stimulation and onset of evoked EMG waveform. For every animal and every muscle where this was possible, the difference in latency between the responses evoked by a rostral and a caudal stimulation site was computed (candidate muscles were such that a detectable response could be evoked from both a rostral and a caudal site). In this case, for both the rostral and caudal sites, the latencies of 8 individual responses evoked at stimulation amplitudes near motor threshold were retained to analyze the difference.

*Modulation of muscle activity during behavior.* To compute the EMG energies during task execution, the EMG signals were split in 5 contiguous temporal bins: the first 2 corresponding to the reach phase, and the last 3 to the pull phase. The transitions between reach and pull phases were determined by visual inspection of video recordings of the experiments.

For a trial involving stimulation with electrode contact $E$, the EMG energy of muscle $M$ during bin $B$ was computed as:

$$\xi^{E,B}_M = \sum\limits_{i=1}^N \left|\text{EMG}^{E,B}_M(i)\right|^2 \times \text{dt}$$

where $\text{dt} = T/N$ with $N$ the number of samples and $T$ the duration of $B$, and where $\text{EMG}^{E,B}_M(i)$ is the $i$-th sample of the EMG signal of muscle $M$ during $B$.

Next, in order to assess whether stimulation caused a suppression or a facilitation effect, we computed a *mean facilitation index* (MFI), estimated as the ratio between EMG energy during stimulation and baseline EMG energy (E = ∅). The mean facilitation index of muscle $M$ for electrode contact $E$ during the reach ($R$) and pull ($P$) phases were respectively computed as:

$$\text{MFI}^{E,R}_M = \frac{\frac{1}{N_E}\sum\limits_{i=1}^{N_E}\left(\xi^{E,B_1}_M + \xi^{E,B_2}_M\right)_i}{\frac{1}{N_\emptyset}\sum\limits_{j=1}^{N_\emptyset}\left(\xi^{\emptyset,B_1}_M + \xi^{\emptyset,B_2}_M\right)_j} = \frac{\xi^{E,R}_M}{\xi^{\emptyset,R}_M}$$

$$\text{MFI}^{E,P}_M = \frac{\frac{1}{N_E}\sum\limits_{i=1}^{N_E}\left(\xi^{E,B_3}_M + \xi^{E,B_4}_M + \xi^{E,B_5}_M\right)_i}{\frac{1}{N_\emptyset}\sum\limits_{j=1}^{N_\emptyset}\left(\xi^{\emptyset,B_3}_M + \xi^{\emptyset,B_4}_M + \xi^{\emptyset,B_5}_M\right)_j} = \frac{\xi^{E,P}_M}{\xi^{\emptyset,P}_M}$$

where $N_E$ and $N_\emptyset$ are the numbers of repeated task executions with stimulation with electrode $E$ and without stimulation, respectively, and the subscripts $i$ and $j$ denote individual task executions.

Variance of the MFI was estimated by propagating variances of each variable using the differential formulation:

$$\text{SEM}(\text{MFI}^{E,X}_M) = \sqrt{\left(\frac{\text{SEM}\left(\xi^{E,X}_M\right)}{\xi^{\emptyset,X}_M}\right)^2 + \left(\frac{\text{SEM}\left(\xi^{\emptyset,X}_M\right) * \xi^{E,X}_M}{\left(\xi^{\emptyset,X}_M\right)^2}\right)^2}$$

for $X = R$ or $X = P$.

*Human data.* We re-structured the clinical data provided by the CHUV to analyze it following the same framework used for the monkey data.

## Statistics

*Bootstrapping.* For each simulated recruitment curve, the initial population of N simulated nerve fibers or motoneurons was resampled with replacement to obtain K=10,000 fictive populations of N individuals. The means and standard deviations across these fictive populations were used to construct the recruitment curves of Figs. 2

and 6, the threshold and saturation amplitudes and the selectivity indices of Fig. 2, and the comparisons between the two sets of simulated recruitment curves of Fig. 6h.

*Analysis of latencies of muscular responses*. For each comparison between rostrally-evoked responses and caudally-evoked responses of Fig. 5g and Fig. S2, the performed statistical tests were unpaired two-sided *t*-tests with null-hypothesis the equality of the means of the two groups of data points.

*Modulation of muscle activity during behavior*. We performed two types of statistical tests to compare the obtained distributions of EMG energies. First, we compared the distributions {with stimulation} and {without stimulation} of each of the 5 temporal bins using Wilcoxon Rank-Sum tests. Second, we compared the distributions {with stimulation} across the 5 bins using a Kruskal–Wallis test analyzed post-hoc with Tukey–Kramer tests.

**Reporting summary**. Further information on research design is available in the Nature Research Reporting Summary linked to this article.

## Data availability

All data supporting the findings of this study are provided within the paper and its Supplementary information. The experimental data used in comparisons with simulation results are included in the public repository holding the computer code necessary to reproduce the simulations of the study (https://bitbucket.org/ngreiner/fem_smc_ees/src/master/). All additional information will be made available upon reasonable request to the authors. Source data are provided with this paper.

## Code availability

All the routines and data necessary to reproduce the computational model and all figures related to simulations in this manuscript are accessible from the following public repository: https://bitbucket.org/ngreiner/greiner_et_al_2020/src/master/.

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

## Acknowledgements

The authors would like to acknowledge the financial support from the Wyss Center grant (WCP 008 to G.C. and M.C.), the Bertarelli Foundation (Catalyst Fund Grant to MC) an Ambizione Fellowship (n°167912 to M.C.), an industrial grant support from GTX medical to M.C. and G.C., and the European Union's Horizon 2020 research and innovation program under the Marie Skłodowska-Curie grant agreement n°665667 to G.S. The authors would like to thank Prof. Eric Rouiller for his support as director of the Platform of Translational Neuroscience of the University of Fribourg; Dr. Eric Schmidlin for kindly providing us with the animals used in the terminal procedures and for his support for the anesthesia; Jacques Maillard and Laurent Bossy for their meticulous work on the care provided daily to the animals; and M.D. Etienne Pralong for providing the clinical data.

## Author contributions

N.G. and M.C conceived the work. N.G. created and implemented the computational model. N.G. and S.B. performed the anatomical analysis. B.B., N.G., M.C., N.J., G.C., and J.B. performed the acute experiments. S.C., M.K., B.B., and M.C. performed the chronic experiments in Mk-Sa. J.B., M.C., G.C., N.G., B.B., G.S., F.F., and S.L. designed the epidural arrays. G.S., F.F., and S.L. manufactured and tested the epidural arrays. G.C. and J.B. performed the chronic implantations in Mk-Sa. N.G., B.B., H.L., and N.J. analyzed the data and created the figures. M.C. and G.C. supervised the work. N.G. and M.C. wrote the manuscript and all the authors contributed to its editing.

## Competing interests

G.C., J.B., and S.L. are shareholders and founders of GTX medical, a company producing spinal cord stimulation technologies. G.C., J.B., M.C., B.B., and S.L. are inventors of multiple patent applications and granted patents covering parts of this work. All other authors declare no competing interests.

## Additional information

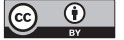

