## [Peer Review File · Nature Communications]

Reviewers' Comments:

Reviewer #1:

Remarks to the Author:

Grenier and colleagues investigate the use of cervical epidural electrical stimulation (ESS) for reanimating upper-limb muscles following paralysis. ESS in the lumbosacral spinal cord has shown promise for restoring lower-limb function following spinal cord, so there is increasing interest in its use also for the upper-limb. The rodent is not a good model for human upper-limb neuroanatomy, therefore this work uses computational modelling, non-human primate experiments as well as data from humans implanted with ESS electrodes for pain. The computational modelling appears very detailed, however I am not an expert on these techniques so my ability to review these sections is limited. The experimental work appears to have been conducted to a high standard. There are several key findings to the paper: (1) dorsal ESS does not activate motoneurons directly, but instead activates them transynaptically, likely via Ia afferents, (2) stimulation of lateral vs. medial sites confers greater muscle selectivity, because this targets the afferents before they branch in the rostro-caudal direction, (3) an electrode array with multiple lateral stimulation sites can reliably activate different motoneuron pools, (4) a variety of temporal response patterns is seen (especially for high stimulation frequencies) and this is true for both non-human primates and humans. Of these findings, I think (1) is not particularly novel or surprising, although the detailed (and open-source) modelling of the cervical cord is a useful addition to the field, (2) is non-obvious and demonstrated using both modelling and experimental work, but it is hard to tell from the data presented whether the selectivity is sufficient for this technique to restore useful, functional movements, (3) is a useful contribution to the neurotechnology field but I would have liked to see better characterization of the long-term stability and biocompatibility of the implant and (4) is an interesting observation but lacks mechanistic understanding. Overall, the manuscript is well written and presented, the open-source model will benefit the community, and the addition of human data helps the translational potential. However, I remain to be convinced that the advances would not be more suitable to a more specialized journal. In particular, there are several important aspects which should be explored in order for this to be a significant advance over the existing literature:

1. There is very little analysis of whether any of the movements produced by ESS are functional, especially since there does not appear to be fine-grained selectivity between muscles innervated from the same level (e.g. EDC, FDS and APB in Fig. 4). Kinematics would be useful here, alongside EMG data, or at least some effort to demonstrate that the EMG patterns correspond to functional muscle synergies.
2. Related to 1 - It is stated that in the introduction that "intraspinal microstimulation, which also engages motoneurons via pre-synaptic pathways, have reported low reproducibility and limited specificity of arm muscle recruitment when applied to the cervical spinal cord of monkeys, raising questions on the applicability of this technology to the upper-limb". However I do not see any comparison that convinces me that the selectivity obtained by ESS is better than that produced by intraspinal stimulation.
3. The experiments are conducted under anesthesia/sedation, however experiments in two animals used chronic implants. Do the authors have any data to show that similar responses can be obtained in awake animals?
4. What was the long-term stability of effects elicited by chronic implants (e.g. thresholds, selectivity etc.)? Was any histology performed to show that the spinal cord was not damaged by either the stimulation or the implant itself?
5. The variety of temporal patterns following high-frequency stimulation (Figure 7) is interesting, but this section is very descriptive without any consideration of possible mechanisms. For example, are there any systematic differences between proximal/distal muscles, or rostral/caudal stimulation sites? Are the 'alternation' patterns indicative of CPG-like behavior, or related to tremor mechanisms - this could perhaps be addressed by looking at synchrony between agonist/antagonist muscles. What is the difference between 'attenuation' and 'suppression', and do these reflect distinct neuronal mechanisms?

6. The human data is nice to have, but there have been previous demonstrations of motor effects of cervical ESS in humans (e.g. Lu et al. *Neurorehabil Neural Repair* 2016). Unfortunately, it was not possible to replicate the key result that lateral stimulation was more selective than medial stimulation. It is stated that the lack of 'maximal EMG amplitudes' meant that selectivity could not be assessed, but there would seem to be ways around this. For example, could the responses be normalized by maximal voluntary contractions (MVCs). Alternatively the authors could compare selectivity in terms of the thresholds for recruiting different muscles, or simply use the un-normalized responses for comparison between medial/lateral stimulation. In the absence of such comparisons, I am not convinced that this section really adds much to the scientific story.

Minor comments:

1. The authors should compare their results to the recent work of Kato et al. (*J Neural Eng* 2020) who have performed a similar study in non-human primates using subdural electrodes.
2. I found the axis labelling 'normalized amplitude' in Figure 2 confusing – I think this refers to stimulation intensity, but I initially read this as the amplitude of the response to stimulation. 'Normalized stimulus intensity' or similar might be better.
3. I found the colors in Figure 2 a bit too similar, hindering interpretation of this figure.

Reviewer #2:

Remarks to the Author:

This is a combined modeling and experimental study (primarily with monkeys, but also some supporting human data) of the mechanisms and efficacy of epidural spinal cord stimulation with the cervical and thoracic segments of the spinal cord to evoke arm movement. The work follows earlier studies in the lumbosacral cord that established the activation of afferent fibers in the dorsal roots leading to transsynaptic activation of motor neurons as the main mechanism. The design and execution of the study is quite sound. I found little of substance to object to, except as noted below. The transsynaptic mechanism is clearly confirmed here again, through both components of the study. The more interesting (and critical) question for this study is that of the selectivity of control of the arm compared to the leg. To be as useful as for the leg, spinal stimulation must be even more selective. The authors state, "Nonetheless, the distribution of sensory afferents in the posterior roots can be exploited to steer the modulation exerted by EES towards specific motor nuclei." I assumed at first they might be referring to current steering ("field shaping") using multi-contact stimulation to improve selectivity. Beyond testing medial and lateral stimulus locations, there wasn't any steering involved. Although figure 4-6 explored this question a bit, I found it hard to get much of a functional sense of what this level of control might permit. It would have been useful to know something about what arm movements were produced. Whether it was possible to produce anything like a grasp, whether there was any hope of independent arm and hand control. How did the selectivity compare to earlier studies that used peripheral nerve stimulation? A further question that is not addressed anywhere is that of the potential effect of anesthesia on excitability as well as selectivity.

Surprisingly, the larger diameter of the $A\alpha$ -DR-fibers does not seem to confer them a substantially lower excitation threshold compared to $A\beta$ -DR-fibers, as could have been expected

Are the curves completely on top of each other? The very similar colors makes this figure hard to interpret.

Fig 4c:

Color is redundant for muscles. Use it for monkeys? I thought at first that there were only 4 symbols for each muscle. The circle around the median is confusing and rather unnecessary, as it's obvious, particularly with the line connecting the medians.

Fig 4d :

Same concerns.

Fig 5c:

It looks suspiciously like one medial monkey worked quite well (column of small squares at ~ 0.9) while the other(s) did not. There should be only 2 medial monkeys. Why are there 3 sets of small gray squares? In a several cases (Bic, Tri, APB) there appear to be only a single medial point.

Thus, our model was able to produce a purely Ia-mediated recruitment of motoneurons within a range of synaptic connectivity parameters coherent with experimental findings.

These estimates themselves span nearly an order of magnitude. Is that also representative of the experimental data, or just the mean?

Moreover, full (under H1) or almost full (under H2) monosynaptic recruitment could be reached before direct motor axonal recruitment began.

However, isn't the main issue that of muscle selectivity? Full monosynaptic recruitment is not particularly useful if it is not selective.

or 0 V when the signal was purely noisy

What does this mean? How is a "pure noise" signal determined?

Minor

correlation between muscle recruitment patterns and motor pool rostrocaudal distributions (Figure 5d),

This is 5c

strongly biased towards caudally-innervated muscles (Figure 5a,e).

5b,d?

Fix the rest of the Fig 5 panel refs as well...

Fig 6: "Recruitment curves: curves are made of 80 data points..."

I'm not sure why there is a reference to recruitment curves at the bottom of the legend.

Fig 7: d-g Incorrect descriptions?

Reviewer #3:

Remarks to the Author:

This is generally a very thorough study, although the results are not especially surprising and could largely be expected, derived from long known anatomy and connectivity. The significant added value is the careful parameter-ization of the models used. Generally, this description of the models is excellent but there are a few details that seem to be missing:

1. Were different size motoneurons included in the simulation -i.e., consistent with the different input resistances of small and large (slow and fast) motoneurons and the size principle. This is not discussed, suggesting a single dimension MN model was used. Indeed this seems to be the case in one section. This is in contrast the the Poisson drawn synapse distributions that are used to manage variability. The rationale for not considering motoneuron size which is an important known variable, and clearly involved in intact motor control, is not elaborated at all, and needs to be explained. This choice seems to limit the usefulness of some measures deriving from the model. It is odd, given the detail elsewhere (e.g., synapse numbers).

2. The stimulation used in the simulation phases seems to be monophasic from the (very brief) simulation description. However, all useful stimulation is biphasic for charge balance(as was used in the experimental component of the study). The duration patterns of phases can be varied as parameters (McIntyre and Grill) to alter selectivity and effect. If pulses in simulation were not

monophasic this needs to be stated and better explained.

3. Muscle selectivity measures are elaborated in the methods but as far as I can tell this data is very briefly and oddly presented, simply as a brief call-out to Figure 4d, and is little interpreted, if at all. These selectivity data for muscle are important and should be presented in more detail and carefully unpacked for the reader. The best individual muscle selectivity possible is a key for fractionated control. If these stimulations only permit group effects of a myotome, this should be clearly stated. This issue is never discussed or presented with any real clarity. The way that muscle selectivity indices are used and presented is currently unclear, and potentially obfuscating, which I do not believe is the authors intent.

4. The authors describe the spinocerebellar tract (SCT) but should distinguish if they mean dorsal or ventral SCT. Further, they should explain why reticulospinal, rubrospinal and vestibulospinal tracts in the lateral funiculi were omitted from simulations. The authors uses posterior/anterior and dorsal/ventral root terminology interchangeably. Likely they should state the equivalence once, pick one terminology and stick with it.

5. It would be really helpful in Discussion to elaborate what -if anything- is surprising in the data and simulations. I believe the study needed to be done, and is generally very well done, but to me is largely confirmatory of the reasonable expectations of any physiologist. Perhaps the lack of difference between A alpha and beta fiber thresholds in simulation is a surprise?

Line numbers would be helpful in review. Some places the authors use 'columns' when I think they mean 'segments' e.g., on page 4.

COMPUTATIONAL AND EXPERIMENTAL ANALYSIS OF THE RECRUITMENT OF UPPER-LIMB MOTONEURONS WITH EPIDURAL ELECTRICAL STIMULATION OF THE PRIMATE CERVICAL SPINAL CORD

NOTE TO ALL THE REVIEWERS

AUTHORS' ANSWERS: report the answers to the reviewer's question.

ACTION IN THE MANUSCRIPT: report the actions taken in the manuscript (in the revised text as **bold red characters**).

TO ALL REVIEWERS: we wish to thank the reviewers for having provided a solid and useful review of our results. We would like to highlight the fact that we found all comments fair with no biased requests or comments. To answers the reviewers' comments, we added a substantial amount of new data. In consequence, we modified the author list accordingly to reflect the contribution of people that participated in the acquisition and analysis of the new data.

Reviewer #1

Grenier and colleagues investigate the use of cervical epidural electrical stimulation (ESS) for reanimating upper-limb muscles following paralysis. ESS in the lumbosacral spinal cord has shown promise for restoring lower-limb function following spinal cord, so there is increasing interest in its use also for the upper-limb. The rodent is not a good model for human upper-limb neuroanatomy, therefore this work uses computational modelling, non-human primate experiments as well as data from humans implanted with ESS electrodes for pain. The computational modelling appears very detailed, however I am not an expert on these techniques so my ability to review these sections is limited. The experimental work appears to have been conducted to a high standard. There are several key findings to the paper: (1) dorsal ESS does not activate motoneurons directly, but instead activates them transynaptically, likely via Ia afferents, (2) stimulation of lateral vs. medial sites confers greater muscle selectivity, because this targets the afferents before they branch in the rostro-caudal direction, (3) an electrode array with multiple lateral stimulation sites can reliably activate different motoneuron pools, (4) a variety of temporal response patterns is seen (especially for high stimulation frequencies) and this is true for both non-human primates and humans. Of these findings, I think (1) is not particularly novel or surprising, although the detailed (and open-source) modelling of the cervical cord is a useful addition to the field, (2) is non-obvious and demonstrated using both modelling and experimental work, but it is hard to tell from the data presented whether the selectivity is sufficient for this technique to restore useful, functional movements, (3) is a useful contribution to the neurotechnology field but I would have liked to see better characterization of the long-term stability and biocompatibility of the implant and (4) is an interesting observation but lacks mechanistic understanding.

Overall, the manuscript is well written and presented, the open-source model will benefit the community, and the addition of human data helps the translational potential. However, I remain to be convinced that the advances would not be more suitable to a more specialized journal. In particular, there are several important aspects which should be explored in order for this to be a significant advance over the existing literature:

AUTHORS' ANSWERS: we thank the reviewer for the appreciation of our work. We found his/her comments fair and honest and we strived to address them at best in this new version of the manuscript. We believe that the comments substantially improved the quality and impact of our work. In this revision **we complemented our results with a new dataset in a chronic intact animal** showing ability to recruit segmental circuits as well as synergistic muscles. We

also included **new analyses in humans**. We believe these new data and analyses addressed the reviewer's concerns while maintaining the main mechanistic goal of our manuscript.

REVIEWER 1 | Comment 1: There is very little analysis of whether any of the movements produced by EES are functional, especially since there does not appear to be fine-grained selectivity between muscles innervated from the same level (e.g. EDC, FDS and APB in Fig. 4). Kinematics would be useful here, alongside EMG data, or at least some effort to demonstrate that the EMG patterns correspond to functional muscle synergies.

AUTHORS' ANSWERS: we agree with the reviewer, and similarly to the others, that we failed at emphasizing this part of our results. We would like to take the occasion to expand on the concept of selectivity and how it does apply to EES.

In the lumbosacral spinal cord, we demonstrated that EES engaged dorsal afferents and thus muscles with a segmental recruitment pattern via mono- and poly-synaptic excitatory pathways. The fact that lumbar EES acts through a pre-motor gateway implies that it does not need to be highly selective to sustain functional movements. Indeed, the goal of EES is rather to convey excitatory inputs to affected segments of the spinal cord. Then, during movement, spinal reflexes and residual volitional control shape this extrinsic excitatory activity into functional movements (Capogrosso 2013 Journal of Neurosci, Capogrosso 2018 Nat Protocols).

Being able to show some level of selectivity allowed to design devices that can shape the neuromodulatory effects of EES towards the region of interests (e.g. promoting hip flexion vs leg extension) thus being more effective than stimulating the entire cord continuously (Wenger 2016 Nat Med, Capogrosso 2016 Nature , Capogrosso 2018, Nat Protoc. Wagner 2018 Nature).

Following this line of thought we wanted to verify what type of selectivity could be achieved for the upper-limb, given the radically different functional and anatomical organization of the upper cervical cord. This entails the demonstration that:

- a) Motoneurons are recruited trans-synaptically also in the cervical spinal cord.
- b) One could use the dorsal roots to produce recruitment patterns that correlate with the segmental organization of the motor nuclei.
- c) During movement different contacts engage different muscles and the recruitment patterns are modulated by reflexes and descending control.

While in the previous version of the manuscript we explored points a) and b), we agree with the reviewer that without demonstrating point c) we would be missing an important step in providing evidence that this technology could be used to sustain complex movements after injury of the cervical spinal cord.

Indeed, reflexes and spinal circuit modulation may not be so prominent in the cervical spinal cord thus hindering applicability of EES to restore arm control.

To address these questions, we changed the text in multiple sections adding discussion points and interpretation of the data. Unfortunately, we don't have video or kinematic recordings of the movement executed during the pulse-train experiments. In fact, our goal was to observe attenuation properties and we did not tune amplitudes to levels necessary to observe movements but just muscle contractions. Therefore, to address the reviewer comment we added a new dataset in a behaving intact animal (Mk-Sa) that was chronically implanted with our interface (**New Figure 8**) which we believe should be more interesting and appropriate:

The rationale for using an intact animal was to demonstrate that stimulation could interact with descending control, similarly to what we did in the past for the leg (Capogrosso 2016 Nature). Mk-Sa was trained to execute unconstrained three-dimensional reaching, grasping and pulling arm movements using a robot interface (Barra 2019 Journal of Neur Eng). We then delivered continuous stimulation from a rostral and a caudal contact at motor-threshold amplitudes in order to observe the modulation of the EMGs without interfering with the execution of the task.

We found that:

- a) Muscle responses were elicited by each pulse of stimulation and their amplitude was modulated during movement similarly to what happens in leg muscles (Moraud and Capogrosso 2016 Neuron).
- b) Muscle activity was enhanced, but modulation effects were phase-dependent and differed substantially between arm and hand muscles.
- c) Rostral and caudal contacts engaged different sets of synergistic flexors and extensors muscles.

This data shows that in the upper arm EES enhanced the activity of synergistic muscles (triceps and wrist extensors, and hand flexors and biceps). However, the way this enhancement interacted with voluntary control and reflexes is complex and differs for rostral and caudal contacts (see also response to other comments). While the triceps was enhanced approximately uniformly during movement (albeit in a highly modulated fashion), the flexor digitorum superficialis (FDS) showed phases of substantial enhancement (during grasping and pulling) and phases of inhibition (just before grasping), that are likely the result of a dynamic modulation from cortical inputs to enable the monkey to execute object grasping when stimulation was continuously delivered.

This combined data shows that EES delivered through cervical lateral contacts engages arm muscles similarly to lumbosacral EES for leg muscles. However, the way stimulation of hand muscles interacts with voluntary control is complex and requires new investigations. For example, it may be necessary to define complex stimulation patterns that take into account these dynamics to maximize function. At the same time, it also indicates that residual descending control might be critical to obtain skilled hand function with EES.

ACTION IN THE MANUSCRIPT:

We modified the **TITLE**:

COMPUTATIONAL AND EXPERIMENTAL ANALYSIS OF THE RECRUITMENT OF UPPER-LIMB MOTONEURONS WITH EPIDURAL ELECTRICAL STIMULATION OF THE PRIMATE CERVICAL SPINAL CORD

We modified the **ABSTRACT**:

Here, we combined a realistic computational model of the cervical spinal cord with experiments in macaque monkeys to explore the mechanisms of **upper-limb motoneuron recruitment with EES** and characterize the selectivity of cervical interfaces. **We show** that lateral electrodes **produce a segmental recruitment** of arm motoneurons **mediated by** the direct **activation of sensory afferents, and that muscle responses to EES are modulated during movement**. Intraoperative recordings suggested similar properties in humans **at rest**. **These modelling and experimental results support** the design of neuro-technologies to improve arm and hand control in humans with quadriplegia.

We modified the **INTRODUCTION:**

[...] Electrophysiological experiments in monkeys and humans **revealed** that: **a) cervical motoneurons are recruited trans-synaptically; b) lateral electrodes recruit arm and hand muscles with a rostro-caudal order that reflects the segmental innervation of upper-limb motor nuclei in the cervical spinal cord; and c) EES modulates the activity of arm muscles during functional movements performed by monkeys, and this modulation is dependent on the muscle and phase of the movement.**

These model and experimental results provide a conceptual framework to design novel EES technologies to improve upper-limb motor control after cervical SCI.

We modified the **RESULTS:**

Paragraph Computational analysis of the primary targets of EES applied to the cervical spinal cord:

As expected, our results indicate that Ia-afferents running in the targeted root (C6) are recruited at significantly lower stimulation amplitudes than those running in the adjacent or more distal roots (C5, C7, C8 and T1, **Figure 2c**). **This means that near full and exclusive recruitment of individual roots can be achieved with individual monopolar lateral electrodes.** [...]

Paragraph Experimental recruitment of cervical motoneurons with epidural stimulation in macaque monkeys

Laterally-positioned electrodes.

We first analyzed the muscular recruitment induced by the lateral electrodes of the electrode arrays (available for the 5 animals). The results are presented in **Figure 4**.

The obtained muscular recruitment profiles are not indicative of a recruitment pattern based on muscle synergies⁴⁰. Instead, they clearly indicate a segmental, rostro-caudal recruitment pattern which reflects the distribution of the upper-limb motor nuclei in the spinal cord (Figure 3).

We added a **new Figure 8** (previous Figure 8 is now Figure 9) and described the results of the corresponding experiment in a

New paragraph Continuous stimulation of the cervical spinal cord during functional, three-dimensional arm movements:

To verify whether muscle activity and movement could be enhanced and modulated during a functional three-dimensional task, we trained one monkey (Mk-Sa) to freely reach, grasp and pull an object using a robotic framework that we designed for this purpose⁴⁶. We investigated how the animal's arm kinematics and EMGs were influenced by the delivery of continuous EES at 50 Hz (typical stimulation frequency in therapeutic applications). We used a lateral rostral electrode targeting the C6-C7 roots and a lateral caudal electrode targeting the C8-T1 roots.

Continuous EES immediately and reliably evoked muscle responses in the upper-limb muscles (Figure 8a,b). However, the amplitude of these responses was modulated depending on the muscle and the movement phase considered (Figure 8b). For example,

responses in the triceps muscle were enhanced during reaching and strongly suppressed during grasping and pulling.

The two contacts engaged distinct muscles (Figure 8d). Coherently with its position, the C6-C7 contact mostly engaged the triceps. On the other hand, the C8-T1 contact mainly engaged the triceps and the flexor carpi radialis (FCR), which, again, was consistent with its position. However, this contact also engaged the biceps, which is more rostrally innervated (C5-C6 segments). Note that the biceps is a synergistic flexor muscle of the FCR, which may explain the emergence of the observed muscle responses (see Discussion). Overall the modulation was reduced during the grasping and pulling phase (Figure 8d, right polar plot).

Finally, we performed a principal component analysis (PCA) on a selection of upper-limb kinematic parameters (see Methods) recorded while the monkey performed the task with or without stimulation. Albeit we tuned the stimulation amplitude to a level that did not impair the animal's ability to perform the task, our analysis revealed that EES did influence arm kinematics. The affected parameters were differentially influenced by the two electrodes (Figure 8e). The PCA shows that kinematics parameters moved towards different directions in the PC space under the effect of the two electrodes. Interestingly, the rostral contact (targeting the triceps) enhanced the peak velocity of the elbow angle, as expected from an electrode engaging elbow extensors. Furthermore, we also observed a positively monotonic relationship between the stimulation amplitude and the amplitude of the modulation of the kinematic parameters (Figure 8e).

We modified the **DISCUSSION**:

Paragraph Stimulation specificity and lead design:

[...] Specifically, the wide spatial separations between adjacent dorsal roots at the cervical level (several millimeters), which is consistent across subjects, should allow robust selective recruitment of individual roots, but the intermingling of different fiber populations within these roots may limit the ability to engage specific motor nuclei.

Similarly to EES of the lumbosacral spinal cord⁵⁸, the recruitment patterns that we obtained during single pulse experiments were explained by the rostro-caudal innervation of motor nuclei rather than by the recruitment of spinal circuits implementing muscle synergies⁴⁰. Given the near perfect segmental separation between the biceps and triceps motor nuclei, a segmental recruitment pattern seems sufficient to selectively promote elbow extension or flexion movements. However, given the overlapping innervation of forearm and hand muscles, selective facilitation of individual finger muscles cannot be achieved with our type of implant. By being closer to the targeted afferents, it may seem that subdural implants may improve the recruitment selectivity, but a recent report⁵⁹ of the recruitment pattern induced by lateral subdural electrodes in monkeys indicated similar limitations to those outlined in the present work. Nevertheless, high-density electrode arrays and current steering techniques, potentially enabling the selective recruitment of structures as fine as individual dorsal rootlets, remain possible options to be explored to further improve the stimulation specificity.

New paragraph **Cervical EES during voluntary movement:**

We observed a segmental specificity in the recruitment of motoneurons following single pulses of low frequency cervical EES at rest. However, during movement execution, spinal circuits continuously receive supra-spinal, proprio-spinal and natural sensory inputs which modify the integration of the artificial inputs induced by EES^{12,18,58}. Thus, the direct activation of the same pre-motor elements may modulate different motor nuclei than during resting conditions depending on the specific motor task and movement phase. This task- and phase-dependent modulation can actually improve efficacy of EES. Indeed, EES should facilitate the activity of muscles naturally engaged during the execution of voluntary movements rather than imposing the recruitment of specific muscles^{6,58}. These properties and therapeutic benefits have been demonstrated for lumbosacral EES, but remained unexplored for cervical EES.

We studied the modulation exerted by cervical EES during a typical reaching, grasping and pulling task performed by an intact monkey chronically implanted with our custom-made electrode array. Contrary to resting conditions, the modulation exerted by rostral and caudal contacts only partly reflected the rostro-caudal innervation of the upper-limb muscles (Figure 8d). Specifically, the activity of a rostrally-innervated muscle was consistently and significantly enhanced by the stimulation from the caudal contact. However, this muscle (biceps) was a flexor synergist of a caudally-innervated muscle (flexor carpi radialis, FCR) which was also engaged by the stimulation. This suggests that during the execution of a voluntary task, stimulating the primary afferents of upper-limb muscles (e.g. FCR) may enhance the motor activity of synergistic muscles, possibly via the contribution of heteronymous excitatory connections. Perhaps, at rest, these secondary pathways may have only secondary effects compared to strong monosynaptic components. An increase of coordinated excitatory inputs from circuits underlying muscle synergies, which are abundantly innervated by sensory afferents⁶², might also be occurring. Finally, we found that the modulation of muscular activity was dependent on the movement phase (Figure 8d), as is the case with lumbosacral EES^{12,58}.

These combined results suggest that, as in the case of the lower limb^{1-3,5,8}, understanding the interplay of residual descending commands with proprio-spinal inputs, sensory feedback and the integrative properties of cervical spinal circuits will be key for the development of EES protocols that successfully ameliorate arm and hand deficits in people with spinal cord injury.

Paragraph Conclusions:

By combining computer simulations and electrophysiology in monkeys and humans, we have provided evidence that EES applied to the cervical spinal cord recruits motoneurons trans-synaptically via the direct excitation of sensory afferents and following their segmental innervation in the cervical spinal cord. Our results indicate that lateral contacts are necessary to achieve this segmental selectivity and that current human electrode arrays are unfit to achieve these results. They also show that the modulation of muscle activity exerted by EES during movement is movement-phase-dependent and likely promotes upper-limb muscle synergies. We believe that these combined results establish a pathway for the development of neuro-technologies for the restoration of arm movements in people with cervical spinal cord injury.

REVIEWER 1 | Comment 2: Related to 1 - It is stated that in the introduction that “intraspinal microstimulation, which also engages motoneurons via pre-synaptic pathways, have reported low reproducibility and limited specificity of arm muscle recruitment when applied to the cervical spinal cord of monkeys, raising questions on the applicability of this technology to the upper-limb”. However I do not see any comparison that convinces me that the selectivity obtained by ESS is better than that produced by intraspinal stimulation.

AUTHORS' ANSWERS: we apologize for this to the reviewer. Indeed, this is a bad wording from our side. We agree with the reviewer that intraspinal microstimulation shows similar specificity to what reported in our study. Indeed, the words “this technology” were actually referring to spinal cord stimulation in general: in the sense that spinal cord stimulation for the upper limb could be challenging. However, we realize that the sentence was badly phrased and misleading. Therefore, we removed it.

ACTION IN THE MANUSCRIPT: upon reviewer's suggestion we removed the sentence from the introduction.

REVIEWER 1 | Comment 3: The experiments are conducted under anesthesia/sedation, however experiments in two animals used chronic implants. Do the authors have any data to show that similar responses can be obtained in awake animals?

AUTHORS' ANSWERS: we addressed this comment with the new figure 8 and results described in the answers to comment 1. Data added are from Mk-Sa: the only animal in our authorization that performed chronic implanted experiments while being intact (it is 1 out of the 2 animals having had a survival surgery). Other animals that are not reported here were chronically implanted with our device. These animals were subject to a spinal cord injury and their data will be included in future work detailing the impact of our technology on motor performances after SCI.

Additionally, we also added details on the use of propofol as a reliable anaesthetic agent to perform electrophysiology.

ACTION IN THE MANUSCRIPT:

We modified the **METHODS** paragraph *Acute electrophysiology*:

All acute electrophysiology was performed under propofol sedation (continuous intravenous injection, 5 ml/kg/h) that minimizes the impact of anesthesia on the responses to spinal cord stimulation⁷⁹. Trains of biphasic [...]

REVIEWER 1 | Comment 4: What was the long-term stability of effects elicited by chronic implants (e.g. thresholds, selectivity etc.)? Was any histology performed to show that the spinal cord was not damaged by either the stimulation or the implant itself?

AUTHORS' ANSWERS: for this specific question we invite the reviewer to refer to our recent publication which details data regarding the technology we used and that we referenced in our manuscript (Schiavone et al., *Advanced Materials*, 2020). Additionally, the new data in the animal reported here was obtained at weeks 3 and 4 after implantation. The animal did not show any deficits or damage to the cord and was chronically implanted for 7 weeks. The animal was then explanted because of an infection on the EMG leads that were unfortunately connected to the same pedestal than the EES electrode. This forced us to explant everything. In

consequence in the subsequent animals we changed protocol and implanted the leads only after SCI. For this reason only Mk-Sa was recorded while being intact and was thus suitable for the content of this manuscript.

ACTION IN THE MANUSCRIPT: to clarify this point we explicitly refer to Schiavone et al. for data regarding stability of the interface.

We modified the **RESULTS** paragraph **Design of cervical spinal implants:**

The second design comprised a single, laterally-positioned column of seven stimulation contacts, and one medial stimulation contact spanning three short-circuited active sites (**Figure S1 g and h**). **Details on the fabrication and stability of the implants have been reported previously³⁹.**

We modified the **METHODS** paragraph **Fabrication of custom spinal implants:**

The connections are then mechanically stabilized by applying a room temperature vulcanization sealant (one component silicone sealant 734, Dow Corning) over the pads (**Fig. S1 f**). **Chronic stability, functionality and biocompatibility are reported in ref.³⁹.**

REVIEWER 1 | Comment 5: The variety of temporal patterns following high-frequency stimulation (Figure 7) is interesting, but this section is very descriptive without any consideration of possible mechanisms. For example, are there any systematic differences between proximal/distal muscles, or rostral/caudal stimulation sites? Are the 'alternation' patterns indicative of CPG-like behavior, or related to tremor mechanisms – this could perhaps be addressed by looking at synchrony between agonist/antagonist muscles. What is the difference between 'attenuation' and 'suppression', and do these reflect distinct neuronal mechanisms?

AUTHORS' ANSWERS: we thank the reviewer for raising these interesting points. When we analysed and performed these experiments, our aim was mostly to reproduce previous results and classical electrophysiological signatures of frequency-dependent suppression of sensory afferent inputs (see Sharpe et al. 2014). We thought that all these patterns clearly show that motoneurons are not activated directly, which was our initial goal. However, we agree with the reviewer that these responses are intriguing and never reported before (at least for the cervical level) hence deserved further analysis. Therefore, we followed the reviewer suggestion and performed a new series of analyses.

New results are reported in **Figure 7**.

We could not find alternation of agonist-antagonist muscles which could point to a "CPG"-like network (an example is given in **Figure 7g**), however we did find significant differences in how muscles respond when they are recruited from caudal contacts vs rostral contacts (**Figure 7i**). Specifically, alternation was more likely to happen with rostral and intermediate contacts while initial adaptation was more likely to occur for caudal contacts.

These differences may reflect electrophysiological signatures of the existence of proprio-spinal circuits implementing hand and arm function, possibly even fine grasping. Such circuits are hypothesized to be located at the C4-C6 spinal segments (Kinoshita et al. 2012), thus consistent with our findings of different behaviours for C4-C6 against C7-T1 root stimulation.

We included these new analyses in the paper.

We understand that these are only initial insights into the mechanisms of these responses, therefore we will investigate these mechanisms by extending our model in future works focusing specifically on these questions.

ACTION IN THE MANUSCRIPT:

We modified the **RESULTS** paragraph **Electrophysiological assessment of the trans-synaptic nature of the motoneuronal recruitment:**

[...] Absence of frequency-dependent modulation such as reported in **Figure 7f** was observed only rarely (**Figure 7h**).

We then investigated whether the alternation patterns observed in EMG responses during high-frequency stimulation could indicate the involvement of central-pattern-generator-like spinal networks. For that, we examined the responses of agonist and antagonist muscles⁴⁵. We found no cases in which agonist activity was correlated or anti-correlated with antagonist activity (Figure 7g). However, we found that alternation patterns were more frequently elicited by rostral and intermediate contacts, while initial adaptation phenomena were more frequently induced by caudal contacts (Figure 7i)

We modified the **DISCUSSION** paragraph **Trans-synaptic recruitment of motoneurons in the cervical spinal cord:**

We have found that trains of electrical stimuli (10-100 Hz) modulated the stimulation-induced muscular responses in a frequency-dependent manner, suggesting the trans-synaptic nature of the underlying motoneuronal recruitment. **Furthermore, we found that alternation patterns of muscle responses were more likely to occur with rostral (C4-C7 area) than with caudal (C8-T1) electrodes. Since proprio-spinal circuits notably involved in the execution of skilled grasping are located in the C4-C6 segments^{54,55}, the emergence of alternation patterns exclusively with rostral electrodes might be explained by the engagement of these circuits, and again indicate the pre-motor nature of the elements directly activated by cervical EES.**

REVIEWER 1 | Comment 6: The human data is nice to have, but there have been previous demonstrations of motor effects of cervical ESS in humans (e.g. Lu et al. Neurorehabil Neural Repair 2016). Unfortunately, it was not possible to replicate the key result that lateral stimulation was more selective than medial stimulation. It is stated that the lack of 'maximal EMG amplitudes' meant that selectivity could not be assessed, but there would seem to be ways around this. For example, could the responses normalized by maximal voluntary contractions (MVCs). Alternatively the authors could compare selectivity in terms of the thresholds for recruiting different muscles, or simply use the un-normalized responses for comparison between medial/lateral stimulation. In the absence of such comparisons, I am not convinced that this section really adds much to the scientific story.

AUTHORS' ANSWERS: the reviewer also in this case raises interesting points that pushed us to rethink at our data. Before presenting our new analysis, we would like to explain why we did not expand on analysis in the first place. These data were not acquired in a clinical trial. Instead, we received anonymized data from neuromonitoring procedures that are routinely executed during surgeries of the spine at CHUV. The goal of these monitoring procedures is usually to

avoid neural damage to certain structures and identify roots from myotomal innervations (see Schirmer et al 2011). Therefore, we did not have control on the number of muscles monitored, or the acquisition of MVC neither on the amplitude values explored in the procedures. We judged that without being able to normalize muscle responses, we could not produce robust analyses of inter-muscle variability because we were not able to reference a muscle to the other.

Therefore, to respond to the reviewer comment, we proceeded in 2 ways:

New analysis on previous data:

We decided to look at the data by normalizing the EMG recordings of each muscle by the maximal peak to peak value observed for that muscle. We thus reconstructed normalized recruitment curves at increasing current amplitudes from the available data. Even if this analysis is limited by the fact that the normalization is not guaranteed to be done with the MVC amplitude, a pattern could still be observed in the data. Caudal muscles seemed to be activated only by caudal lateral contacts with the Medtronic 5-6-5, providing evidence that a similar specificity than obtained in monkeys could be expected in humans. But also confirming that the shape and size of currently available electrode arrays are limiting the specificity that can be obtained compared to monkeys.

New data:

We reported the reviewer's concern to our clinical collaborator, Dr. Etienne Pralong, that provided us with the initial set of data. Dr. Pralong identified a new set of neuromonitoring data that was obtained during complex spinal fixation procedures in which the cervical dorsal bones were removed in 2 patients and roots stimulated independently with a manual probe. These new data, not obtained with the 5-6-5, were merged with the previous data to have a comprehensive analysis examining muscle recruitment along the rostrocaudal axis in 5 subjects. The results of this analysis are reported in the new **Figure 9**.

ACTION IN THE MANUSCRIPT:

We modified the **RESULTS:**

Paragraph Analysis of the recruitment of cervical motoneurons with epidural stimulation in humans:

We gained access to electrophysiological measurements acquired during clinical procedures in **five** human patients (see **Methods**). In **three** patients, a **clinical** paddle **epidural** array was **implanted** for the treatment of chronic neuropathic pain in the arms and hands, and thus approximately positioned between the spinal segments C6 and T1. **In the two other patients, EES was delivered with a pen electrode at various rostro-caudal lateral positions during a spinal decompression procedure.** Although the clinical procedures did not rigorously match the experimental procedures we performed in monkeys, **low-frequency (0.67 – 1 Hz) stimulation at increasing amplitudes from individual contacts was tested in all five patients, and supra-threshold stimulation at frequencies of 10, 20, 60 and 100 Hz was tested in the three patients implanted with a paddle array.**

As observed in monkeys, the muscular recruitment profiles obtained in the five patients tended to reflect a segmental rostro-caudal innervation of the upper-limb muscles. However, the correlation was qualitatively less marked, and the stimulation

specificity appeared poorer compared to monkeys. For instance, the deltoid and biceps could not be activated independently from the triceps or from hand muscles, and reciprocally for the triceps. However, this limited measured performance might be due to the conditions of the clinical procedures as well as from the inappropriate dimensions of the paddle epidural array (see Discussion).

Furthermore, high-frequency stimulation led to muscular responses modulated in a similar way than observed in monkeys...

We modified the **DISCUSSION:**

Paragraph: Stimulation specificity and lead design

Analysis of recruitment selectivity in the clinical data was limited by the nature of the clinical procedures, which were not experimentally-oriented. **In particular, we could not assure that maximal EMG peak-to-peak amplitudes corresponded to maximal muscular activation levels. Therefore, the relative muscular activation levels of Figure 9c should be regarded with caution. This notwithstanding, the data again tended to indicate a segmental rostro-caudal recruitment reflecting the distribution of motoneurons in the spinal cord, albeit not as markedly as in monkeys** However...

We modified the **METHODS:**

Human data.

Anonymized clinical data were obtained from the Centre Hospitalier Universitaire Vaudois (CHUV, Lausanne, Switzerland). The data were obtained during standard clinical practice and under the CHUV's general ethical approval for clinical procedures and use of anonymized clinical data for scientific purposes. The Swiss federal institute of the use of human data for scientific research (Swiss Ethics) validated the use of the anonymized human dataset. **In 3 patients suffering from upper-arm neuropathic pain, current-controlled electrical stimulation was delivered from individual contacts of a Medtronic Specify™ 5-6-5 interface** at frequencies ranging from 1 Hz to 100 Hz and amplitudes ranging from 0 to 5 mA to test for correct surgical placement of **the implant. In 2 patients requiring surgical laminectomies for spinal decompression, current-controlled stimulation at a frequency of 1 Hz and amplitudes ranging from 100 μ A to 800 μ A was delivered with a manual probe from various positions of the epidural space.**

In all 5 patients, percutaneous needles were placed in various upper-limb muscles and EMG signals were recorded (sampling rate = 10 kHz) using a clinical monitoring interface. Exported data were anonymized and provided to the authors with no reference to patient identity.

REVIEWER 1 | Comment 7: The authors should compare their results to the recent work of Kato et al. (J Neural Eng 2020) who have performed a similar study in non-human primates using subdural electrodes.

AUTHORS' ANSWERS: we thank the reviewer for pointing this out, indeed the paper was published right at the same time when we submitted our manuscript. We took into account the

results obtained by Kato et al. and commented them in the revised version of the manuscript.

ACTION IN THE MANUSCRIPT:

We modified the **DISCUSSION** paragraph **Stimulation specificity and lead design:**

[...] By being closer to the targeted afferents, it may seem that subdural implants may improve the recruitment selectivity, but a recent report⁵⁹ of the recruitment pattern induced by lateral subdural electrodes in monkeys indicated similar limitations to those outlined in the present work. Nevertheless, high-density electrode arrays and current steering techniques, potentially enabling the selective recruitment of structures as fine as individual dorsal rootlets, remain possible options to be explored to further improve the stimulation specificity.

REVIEWER 1 | Comment 8: I found the axis labelling ‘normalized amplitude’ in Figure 2 confusing – I think this refers to stimulation intensity, but I initially read this as the amplitude of the response to stimulation. ‘Normalized stimulus intensity’ or similar might be better.

AUTHORS’ ANSWERS: we thank the reviewer for pointing this out. We agree that the phrase “normalized amplitude” was ambiguous. We modified it in the sense of the reviewer’s proposal.

ACTION IN THE MANUSCRIPT: we modified “normalized amplitude” to “stimulus amplitude”, reminding in the figure caption that this amplitude is a normalized one.

REVIEWER 1 | Comment 9: I found the colors in Figure 2 a bit too similar, hindering interpretation of this figure.

AUTHORS’ ANSWERS: we thank the reviewer for pointing this out. We agree that the colors were hardly distinguishable. We changed them as suggested.

ACTION IN THE MANUSCRIPT: we changed the colors to allow a better interpretation of the figure.

Reviewer #2

This is a combined modeling and experimental study (primarily with monkeys, but also some supporting human data) of the mechanisms and efficacy of epidural spinal cord stimulation with the cervical and thoracic segments of the spinal cord to evoke arm movement. The work follows earlier studies in the lumbosacral cord that established the activation of afferent fibers in the dorsal roots leading to transsynaptic activation of motor neurons as the main mechanism. The design and execution of the study is quite sound. I found little of substance to object to, except as noted below. The transsynaptic mechanism is clearly confirmed here again, through both components of the study. The more interesting (and critical) question for this study is that of the selectivity of control of the arm compared to the leg. To be as useful as for the leg, spinal stimulation must be even more selective.

AUTHORS' ANSWERS: we thank the reviewer for the appreciation of our study. We found his/her supportive comments very useful to increase impact of our work and tried to address them by including **new data in both monkeys and humans**.

REVIEWER 2 | Comment 1: The authors state, “Nonetheless, the distribution of sensory afferents in the posterior roots can be exploited to steer the modulation exerted by EES towards specific motor nuclei.” I assumed at first they might be referring to current steering (“field shaping”) using multi-contact stimulation to improve selectivity. Beyond testing medial and lateral stimulus locations, there wasn’t any steering involved. Although figure 4-6 explored this question a bit, I found it hard to get much of a functional sense of what this level of control might permit.

AUTHORS' ANSWERS: we apologize for this to the reviewer. Indeed, that specific wording was confusing. Our hypothesis is that the segmental distribution of the motoneurons and their monosynaptic connectivity with sensory afferents in the posterior roots, allow to steer the effects of EES to specific segments. For instance, recruiting afferents in the root C5, will convey excitatory inputs to all the motoneurons connected to C5 afferents. Contrary to what we expected, our model showed that we could engage single roots without the need of current steering. Indeed, single monopolar stimulation was enough to have a single root recruitment pattern (**Figure 2,4**). However, further specificity may be obtained by using subdural high-density contacts allowing current steering as the reviewer is suggesting. To address the reviewer concerns we modified the introduction and the discussion.

ACTION IN THE MANUSCRIPT:**We modified the INTRODUCTION:**

[...] Nonetheless, the distribution of sensory afferents in the **dorsal** roots can be exploited, **by targeting individual roots**, to **direct** the modulation exerted by EES towards specific motor nuclei. [...]

We modified the DISCUSSION paragraph Stimulation specificity and lead design:

[...] **By being closer to the targeted afferents, it may seem that subdural implants may improve the recruitment selectivity, but a recent report⁵⁹ of the recruitment pattern induced by lateral subdural electrodes in monkeys indicated similar limitations to those outlined in the present work. Nevertheless, high-density electrode arrays and current steering techniques, potentially enabling the selective recruitment of structures as fine as individual dorsal rootlets, remain possible options to be explored to further improve the stimulation specificity.**

REVIEWER 2 | Comment 2: It would have been useful to know something about what arm movements were produced. Whether it was possible to produce anything like a grasp, whether there was any hope of independent arm and hand control.

AUTHORS' ANSWERS: Reviewer 1 expressed a similar concern which we tried to address by adding new data and changed the manuscript in multiple sections.

Briefly, the purpose of EES is not to produce movement but to support the excitation of spinal segments that are required to perform a movement after spinal cord injury (Capogrosso 2018 Nat Protocol, Edgerton 2008). Specificity is useful because it allows to direct this excitation to the required segments thus allowing the use of stronger amplitudes without disrupting other circuits and producing stronger and more effective movements (Wenger 2016 Nat Med, Capogrosso 2016 Nature, Capogrosso 2018 Nature Protocols, Wagner 2018 Nature). However during movement overlapping recruitment of antagonist muscle is smoothed by reciprocal inhibition reflexes and residual descending control (the concept of *functional specificity* Moraud & Capogrosso 2016 Neuron, Capogrosso 2018 Nature Protocols).

To address the reviewer's concern, we included in the revised manuscript a new dataset (**New Figure 8**) in which we measured the enhancement of muscle activity during movement in an intact animal chronically implanted with our interface.

Sa was trained to execute unconstrained three-dimensional reaching, grasping and pulling arm movements using a robot task (Barra 2019 Journal of Neur Eng). We then delivered continuous stimulation from a rostral and a caudal contact at threshold in order to observe the modulation of the EMGs without interfering with the execution of the task.

We found that:

- a) Spinal reflexes were elicited by each pulse of stimulation and their amplitude was modulated during movement similarly to what happens in leg muscles (Moraud and Capogrosso 2016 Neuron).
- b) Muscle activity was enhanced, but modulation effects were phase-dependent and differed substantially between arm muscles and hand flexors.
- c) Rostral and caudal contacts engaged different sets of synergistic flexors and extensors muscles.

This data shows that in the upper arm EES enhanced the activity of synergistic muscles (Triceps and wrist extensors and hand flexors and biceps). However, the way this enhancement interacted with voluntary control and reflexes is complex and differs from rostral and caudal contacts (see also response to other comments). While triceps were enhanced approximately uniformly over movement albeit highly modulated, FDS showed regions of substantial enhancement and even regions of inhibition, that are likely the result of a dynamic modulation from cortical inputs to enable the monkey to execute object grasping when stimulation was continuously delivered.

This combined data shows that EES delivered to cervical lateral contacts engages arm muscles and similarly to EES for leg muscles. However, the way stimulation of hand muscles interacts with voluntary control is complex and requires new investigations. For example, it may be necessary to define complex stimulation patterns that take into account these dynamics to maximize function. At the same time, it also indicates that residual descending control might be critical to obtain skilled hand function with EES.

ACTION IN THE MANUSCRIPT:

We modified the **TITLE:**

COMPUTATIONAL AND EXPERIMENTAL ANALYSIS OF THE RECRUITMENT OF UPPER-LIMB MOTONEURONS WITH EPIDURAL ELECTRICAL STIMULATION OF THE PRIMATE CERVICAL SPINAL CORD

We modified the **ABSTRACT:**

Here, we combined a realistic computational model of the cervical spinal cord with experiments in macaque monkeys to explore the mechanisms of **upper-limb motoneuron recruitment with EES** and characterize the selectivity of cervical interfaces. **We show** that lateral electrodes **produce a segmental recruitment** of arm motoneurons **mediated by** the direct **activation of sensory afferents, and that muscle responses to EES are modulated during movement**. Intraoperative recordings suggested similar properties in humans **at rest**. **These modelling and experimental results support** the design of neuro-technologies to improve arm and hand control in humans with quadriplegia.

We modified the **INTRODUCTION:**

[...] Electrophysiological experiments in monkeys and humans **revealed** that: **a) cervical motoneurons are recruited trans-synaptically; b) lateral electrodes recruit arm and hand muscles with a rostro-caudal order that reflects the segmental innervation of upper-limb motor nuclei in the cervical spinal cord; and c) EES modulates the activity of arm muscles during functional movements performed by monkeys, and this modulation is dependent on the muscle and phase of the movement.**

These model and experimental results provide a conceptual framework to design novel EES technologies to improve upper-limb motor control after cervical SCI.

We modified the **RESULTS:**

Paragraph Computational analysis of the primary targets of EES applied to the cervical spinal cord:

As expected, our results indicate that Ia-afferents running in the targeted root (C6) are recruited at significantly lower stimulation amplitudes than those running in the adjacent or more distal roots (C5, C7, C8 and T1, **Figure 2c**). **This means that near full and exclusive recruitment of individual roots can be achieved with individual monopolar lateral electrodes.** [...]

Paragraph Experimental recruitment of cervical motoneurons with epidural stimulation in macaque monkeys

Laterally-positioned electrodes.

We first analyzed the muscular recruitment induced by the lateral electrodes of the electrode arrays (available for the 5 animals). The results are presented in **Figure 4**.

The obtained muscular recruitment profiles are not indicative of a recruitment pattern based on muscle synergies⁴⁰. Instead, they clearly indicate a segmental, rostro-caudal recruitment pattern which reflects the distribution of the upper-limb motor nuclei in the spinal cord (Figure 3).

We added a **new Figure 8** (previous Figure 8 is now Figure 9) and described the results of the corresponding experiment in a

New paragraph **Continuous stimulation of the cervical spinal cord during functional, three-dimensional arm movements:**

To verify whether muscle activity and movement could be enhanced and modulated during a functional three-dimensional task, we trained one monkey (Mk-Sa) to freely reach, grasp and pull an object using a robotic framework that we designed for this purpose⁴⁶. We investigated how the animal's arm kinematics and EMGs were influenced by the delivery of continuous EES at 50 Hz (typical stimulation frequency in therapeutic applications). We used a lateral rostral electrode targeting the C6-C7 roots and a lateral caudal electrode targeting the C8-T1 roots.

Continuous EES immediately and reliably evoked muscle responses in the upper-limb muscles (Figure 8a,b). However, the amplitude of these responses was modulated depending on the muscle and the movement phase considered (Figure 8b). For example, responses in the triceps muscle were enhanced during reaching and strongly suppressed during grasping and pulling.

The two contacts engaged distinct muscles (Figure 8d). Coherently with its position, the C6-C7 contact mostly engaged the triceps. On the other hand, the C8-T1 contact mainly engaged the triceps and the flexor carpi radialis (FCR), which, again, was consistent with its position. However, this contact also engaged the biceps, which is more rostrally innervated (C5-C6 segments). Note that the biceps is a synergistic flexor muscle of the FCR, which may explain the emergence of the observed muscle responses (see Discussion). Overall the modulation was reduced during the grasping and pulling phase (Figure 8d, right polar plot).

Finally, we performed a principal component analysis (PCA) on a selection of upper-limb kinematic parameters (see Methods) recorded while the monkey performed the task with or without stimulation. Albeit we tuned the stimulation amplitude to a level that did not impair the animal's ability to perform the task, our analysis revealed that EES did influence arm kinematics. The affected parameters were differentially influenced by the two electrodes (Figure 8e). The PCA shows that kinematics parameters moved towards different directions in the PC space under the effect of the two electrodes. Interestingly, the rostral contact (targeting the triceps) enhanced the peak velocity of the elbow angle, as expected from an electrode engaging elbow extensors. Furthermore, we also observed a positively monotonic relationship between the stimulation amplitude and the amplitude of the modulation of the kinematic parameters (Figure 8e).

We modified the **DISCUSSION:**

Paragraph **Stimulation specificity and lead design:**

[...] Specifically, the wide spatial separations between adjacent dorsal roots at the cervical level (several millimeters), which is consistent across subjects, should allow robust selective recruitment of individual roots, but the intermingling of different fiber populations within these roots may limit the ability to engage specific motor nuclei.

Similarly to EES of the lumbosacral spinal cord⁵⁸, the recruitment patterns that we obtained during single pulse experiments were explained by the rostro-caudal innervation of motor nuclei rather than by the recruitment of spinal circuits implementing muscle synergies⁴⁰. Given the near perfect segmental separation between the biceps and triceps motor nuclei, a segmental recruitment pattern seems sufficient to selectively promote elbow extension or flexion movements. However, given the overlapping innervation of forearm and hand muscles, selective facilitation of individual finger muscles cannot be achieved with our type of implant. By being closer to the targeted afferents, it may seem that subdural implants may improve the recruitment selectivity, but a recent report⁵⁹ of the recruitment pattern induced by lateral subdural electrodes in monkeys indicated similar limitations to those outlined in the present work. Nevertheless, high-density electrode arrays and current steering techniques, potentially enabling the selective recruitment of structures as fine as individual dorsal rootlets, remain possible options to be explored to further improve the stimulation specificity.

New paragraph **Cervical EES during voluntary movement:**

We observed a segmental specificity in the recruitment of motoneurons following single pulses of low frequency cervical EES at rest. However, during movement execution, spinal circuits continuously receive supra-spinal, proprio-spinal and natural sensory inputs which modify the integration of the artificial inputs induced by EES^{12,18,58}. Thus, the direct activation of the same pre-motor elements may modulate different motor nuclei than during resting conditions depending on the specific motor task and movement phase. This task- and phase-dependent modulation can actually improve efficacy of EES. Indeed, EES should facilitate the activity of muscles naturally engaged during the execution of voluntary movements rather than imposing the recruitment of specific muscles^{6,58}. These properties and therapeutic benefits have been demonstrated for lumbosacral EES, but remained unexplored for cervical EES.

We studied the modulation exerted by cervical EES during a typical reaching, grasping and pulling task performed by an intact monkey chronically implanted with our custom-made electrode array. Contrary to resting conditions, the modulation exerted by rostral and caudal contacts only partly reflected the rostro-caudal innervation of the upper-limb muscles (Figure 8d). Specifically, the activity of a rostrally-innervated muscle was consistently and significantly enhanced by the stimulation from the caudal contact. However, this muscle (biceps) was a flexor synergist of a caudally-innervated muscle (flexor carpi radialis, FCR) which was also engaged by the stimulation. This suggests that during the execution of a voluntary task, stimulating the primary afferents of upper-limb muscles (e.g. FCR) may enhance the motor activity of synergistic muscles, possibly via the contribution of heteronymous excitatory connections. Perhaps, at rest, these secondary pathways may have only secondary effects compared to strong monosynaptic components. An increase of coordinated excitatory inputs from circuits underlying muscle synergies, which are abundantly innervated by sensory afferents⁶², might also be occurring. Finally, we found that the modulation of muscular activity was dependent on the movement phase (Figure 8d), as is the case with lumbosacral EES^{12,58}.

These combined results suggest that, as in the case of the lower limb^{1-3,5,8}, understanding the interplay of residual descending commands with proprio-spinal inputs, sensory feedback and the integrative properties of cervical spinal circuits will be

key for the development of EES protocols that successfully ameliorate arm and hand deficits in people with spinal cord injury.

Paragraph Conclusions:

By combining computer simulations and electrophysiology in monkeys and humans, we have provided evidence that EES applied to the cervical spinal cord recruits motoneurons trans-synaptically via the direct excitation of sensory afferents and following their segmental innervation in the cervical spinal cord. Our results indicate that lateral contacts are necessary to achieve this segmental selectivity and that current human electrode arrays are unfit to achieve these results. They also show that the modulation of muscle activity exerted by EES during movement is movement-phase-dependent and likely promotes upper-limb muscle synergies. We believe that these combined results establish a pathway for the development of neuro-technologies for the restoration of arm movements in people with cervical spinal cord injury.

REVIEWER 2 | Comment 3: How did the selectivity compare to earlier studies that used peripheral nerve stimulation?

AUTHORS' ANSWERS: The selectivity obtained with EES in the forearm and hand area is markedly lower than that obtainable with peripheral nerve interfaces (See Ledbetter et al 2013 J Neurophys or Brill et al Journal of Neural Eng. 2018). Perhaps only comparable in the upper arm and the sharp separation between elbow flexor/extensors. However, as we specified above, muscle specificity during single pulse experiments is not a direct goal for spinal cord stimulation.

ACTION IN THE MANUSCRIPT:

We modified the **DISCUSSION** paragraph **Stimulation specificity and lead design:**

[...] However, given the overlapping innervation of forearm and hand muscles, selective facilitation of individual finger muscles cannot be achieved with our type of implant. [...]

REVIEWER 2 | Comment 4: A further question that is not addressed anywhere is that of the potential effect of anesthesia on excitability as well as selectivity.

AUTHORS' ANSWERS: We thank the reviewer for pointing this out, indeed it is a very relevant question. We used propofol because it is a very stable and reliable agent for electrophysiological studies, contrary to isoflurane or sevoflurane for example. Impact of propofol during spinal cord stimulation were recently reported in a study from the group of Mushawar (Toosi et al JNE 2019) that we now reported.

ACTION IN THE MANUSCRIPT:

We modified the **METHODS** paragraph *Acute Electrophysiology*:

All acute electrophysiology was performed under propofol sedation (continuous intravenous injection, 5 ml/kg/h) that minimizes the impact of anesthesia on the responses to spinal cord stimulation⁷⁹. Trains of biphasic [...]

REVIEWER 2 | Comment 5: Surprisingly, the larger diameter of the $A\alpha$ -DR-fibers does not seem to confer them a substantially lower excitation threshold compared to $A\beta$ -DR-fibers, as could have been expected. Are the curves completely on top of each other? The very similar colors makes this figure hard to interpret.

AUTHORS' ANSWERS: we thank the reviewer for his/her comment, which was also brought up by reviewer #1. The curves are indeed almost superimposed. Albeit that is particularly true in comparison of other fiber types and only for lateral stimulation. When the electrode is further away from the root, this effect seems less evident and the separation in threshold between the two fiber types becomes significant: threshold of $A\beta$ -DR-fibers is almost twice as $A\alpha$ (**Figure 2e**). We think that the proximity to the fibers and the dimensions of the electrode might explain the small difference in recruitment threshold obtained with a lateral electrode. To address the reviewer comment, we expanded this point in the discussion and modified the **RESULTS** section. Furthermore, we agree that the colors were very similar, hindering the interpretation of the figure, therefore we changed them.

ACTION IN THE MANUSCRIPT: we changed the colors of the new **Figure 2**

We modified the **RESULTS:**

Paragraph Laterally-positioned electrodes.

Surprisingly, **for lateral contacts**, the larger diameter of the $A\alpha$ -DR-fibers compared to $A\beta$ -DR-fibers does not seem to confer them a substantially lower excitation threshold, as could be expected^{37,38} (**see Discussion**).

Paragraph Medially-positioned electrodes.

[...] while that for motor axons to be as high as ~9 times this value (**Figure 2f**). **In addition, contrary to lateral stimulation, the threshold of $A\beta$ -DR-fibers to medial stimulation is significantly higher than that of $A\alpha$ -DR-fibers (almost twice as high), as could be expected for an electrode position appreciably far from the fibers^{37,38}.**

We modified the **DISCUSSION** paragraph **Detailed modelling of EES of the cervical spinal cord:**

[...] **Our hybrid model of cervical EES in monkeys led to an unexpected result: we found that the recruitment of $A\beta$ -DR-fibers following lateral stimulation occurred at stimulation amplitudes only slightly higher than those necessary to recruit $A\alpha$ -DR-fibers, albeit their average diameter was only two thirds that of the $A\alpha$ -fibers. Comparatively, the difference in excitation threshold was much more pronounced for medial stimulation, closer to what is usually expected^{37,38}. This result is likely due to the combination of 1) the close proximity between the lateral electrodes and the dorsal roots (Figure 2a), which was a consequence of the fine reconstruction of the spinal cord model based on empirical morphological data, and 2) the small width of the electrodes. The latter were indeed approximately 700 μm wide, while the average distance between consecutive nodes of Ranvier of $A\alpha$ -fibers and $A\beta$ -fibers were respectively 1400 μm and 900 μm . Thus, $A\beta$ -fibers were more likely to possess a node directly below the electrode surface, which might have given them a surplus of excitability compared to $A\alpha$ -fibers, albeit not as large as the surplus coming from the larger diameters of the $A\alpha$ -fibers. This effect might be less prominent in humans, where, given the larger size of the human spinal**

canal, the proximity between electrodes and dorsal roots should not be as close. However, the inability to selectively recruit the A α dorsal root fibers with lateral electrodes would be detrimental to both sensory⁵² and motor applications⁵³ of spinal cord stimulation, so we believe this point deserves further investigations.

REVIEWER 2 | Comment 6: Fig 4c: Color is redundant for muscles. Use it for monkeys? I thought at first that there were only 4 symbols for each muscle. The circle around the median is confusing and rather unnecessary, as it's obvious, particularly with the line connecting the medians. Fig 4d : Same concerns.

AUTHORS' ANSWERS: we tried to follow the reviewer suggestion by switching colors or symbols but the figure became even more confusing. We really think that coding monkeys instead of muscles makes the visualization extremely complicated. We understand and agree that there is a lot of colors but we could not find another way to summarize all this information. Finally, also the black circle around the median is important given that the median is one of the data points. We also think that it guides the eyesight on one point distinguished from the rest. Therefore, we did not change this figure.

REVIEWER 2 | Comment 7: Fig 5c: It looks suspiciously like one medial monkey worked quite well (column of small squares at ~ 0.9) while the other(s) did not. There should be only 2 medial monkeys. Why are there 3 sets of small gray squares? In a several cases (Bic, Tri, APB) there appear to be only a single medial point.

AUTHORS' ANSWERS: the reviewer is correct in identifying that one monkey is better than the other for more caudal muscles. However, that is not true for Deltoid, Biceps and Triceps. This in fact is clear also from **Figure 4b**. In fact, in our monkeys the median recruitment seems to be biased towards more caudally innervated muscles as we reported in the manuscript. For the most caudal muscles this is not so dissimilar than the lateral stimulation. Albeit, we would like to point out that thresholds are higher and hence mean activation is significantly lower (**Figure 5d**) for the lateral contacts.

We also would like to thank the reviewer for his notification about the squares representing the data points. We inspected carefully the figure and we found that for those muscles where it seemed that only 1 point was present, the data point squares (small, without borders) were hidden by the median point square (large, with black borders). This is true except for the APB muscle in panels **c,d,e**: we found that the computer script generating the figure missed 1 data point for this muscle. We apologize for this mistake, which we now corrected (this did not affect the conclusions drawn from the figure and results).

ACTION IN THE MANUSCRIPT: we modified **Figure 5c** and inserted a notification about the hidden squares in the figure caption.

REVIEWER 2 | Comment 8: Thus, our model was able to produce a purely Ia-mediated recruitment of motoneurons within a range of synaptic connectivity parameters coherent with experimental findings. These estimates themselves span nearly an order of magnitude. Is that also representative of the experimental data, or just the mean?

AUTHORS' ANSWERS: we thank the reviewer for this interesting question. From (Finkle and Redman 1983), it can be understood that:

- individual Ia synaptic boutons induce an average somatic peak potential of 100 μ V (ranging from 75 μ V to 200 μ V).

- for a somatic synaptic bouton, this would correspond to a peak synaptic conductance of 5nS.

- for a distal dendritic bouton, the number of ion channels, and thus the peak synaptic conductance, are believed to be 10 times higher.

Therefore, in a single motoneuron, the conductance of the Ia synapses may vary by the same factor 10 which we have found for the Ia synaptic conductance of different muscles.

Indeed, we are not arguing that there actually exists a factor 10 between the Ia synaptic conductances of different muscles. What we are saying is: if different muscles possess different numbers of Ia fibers (which is likely), and if the motor pools of these muscles are equally excitable by Ia fibers, then there must be a difference between their respective synaptic conductance (all other things being equal, such as the connectivity ratio and the contact abundance). But it may very well be that different muscles are in fact differently excitable by Ia fibers, or that adjustments of other parameters than the synaptic conductance (such as the connectivity ratio and the contact abundance) are responsible for an even Ia excitability. Unfortunately, our data did not permit to adjudicate on this matter. We changed the section to make this point clearer.

ACTION IN THE MANUSCRIPT:

We modified the **RESULTS** paragraph **Computational analysis of the Ia-induced motoneuronal recruitment**:

The resulting g_{syn} values ranged from 3.375 pS (triceps) to 28.5 pS (abductor pollicis brevis) (**Supplementary Table 1**) which are of the same order of magnitude (**5 pS**) and **within the variability** estimated experimentally by previous investigators⁴⁰.

REVIEWER 2 | Comment 9: Moreover, full (under H1) or almost full (under H2) monosynaptic recruitment could be reached before direct motor axonal recruitment began. However, isn't the main issue that of muscle selectivity? Full monosynaptic recruitment is not particularly useful if it is not selective.

AUTHORS' ANSWERS: please allow us to expand on this theoretical finding. We have found that a pure monosynaptic recruitment is predicted for a wide range of synaptic connectivity parameters between Ia-fibers and motoneurons. This is true even under the H2 hypothesis that assumes a low synaptic conductance for all muscles. This finding constitutes a strong argument supporting the hypothesis that EES allows to modulate motoneurons trans-synaptically (via the direct recruitment of sensory afferents) without directly recruiting motor axons. In fact, EES seems able to achieve full motoneuron recruitment trans-synaptically without recruiting ventral axons.

This is important because in our view, the direct recruitment of motor axons should be avoided when using EES: if occurring, it would make it very difficult to selectively engage specific muscles or muscle synergies because spinal reflex mechanisms and descending control that

modulate this output (as we reported in **New Figure 9**) would not be able to modulate activation of motor axons, thus leading to strong co-contractions and muscle fatigue. Targeting directly motor axons would be more efficient with other type of technologies such as intra-fascicular peripheral nerve electrodes.

ACTION IN THE MANUSCRIPT:

We modified the **RESULTS** paragraph **Computational analysis of the Ia-induced motoneuronal recruitment:**

Moreover, full (under H1) or almost full (under H2) monosynaptic recruitment could be reached before direct motor axonal recruitment began.

Finally, comparing these simulated recruitment curves with experimental recruitment curves did not allow to retain one hypothesis over the other (Figure 6h). However, both implied that the direct activation of Ia-afferents could induce the monosynaptic recruitment of a substantial number of motoneurons well before the direct recruitment of motor-axons occurs. This provides a strong theoretical support to the hypothesis that cervical EES recruits motoneurons trans-synaptically (notably via Ia-fibers).

REVIEWER 2 | Comment 10: or 0 V when the signal was purely noisy. What does this mean? How is a “pure noise” signal determined?

AUTHORS' ANSWERS: We apologize, the reviewer is correct that this sentence was vague. A 50 ms-long snippet of EMG data putatively embedding a compound muscular action potential was considered purely noisy when its maximal amplitude did not exceed 3 times the standard deviation of the last 15 ms of the data snippet. This time window always only contained the baseline signal recorded by the concerned electrode, free of any muscular electrical activity.

ACTION IN THE MANUSCRIPT:

We modified the **METHODS** paragraph:

Muscle recruitment and recruitment curves.

From the electromyographic recordings of low-frequency stimulation protocols (0.67 Hz), we extracted 50 ms-long snippets of data following each stimulation pulse. For each data snippet, we measured the peak-to-peak amplitude of the recorded signal, $P2P(snippet)$, **except for purely noisy signals, for which a value of 0 V was retained (a signal was considered purely noisy when its maximal amplitude was smaller than 3 times the standard deviation of the recording channel baseline signal).**

REVIEWER 2 | Comment 11: Minor correlation between muscle recruitment patterns and motor pool rostrocaudal distributions (Figure 5d), This is 5c

AUTHORS' ANSWERS: we thank the reviewer for noticing this. The reference was indeed

erroneous.

ACTION IN THE MANUSCRIPT: we corrected the reference.

REVIEWER 2 | Comment 12: strongly biased towards caudally-innervated muscles (Figure 5a,e). 5b,d?

AUTHORS' ANSWERS: we thank the reviewer for noticing this. The references were indeed erroneous.

ACTION IN THE MANUSCRIPT: we corrected the references.

REVIEWER 2 | Comment 12: Fix the rest of the Fig 5 panel refs as well...

AUTHORS' ANSWERS: we thank the reviewer for his suggestion. There remained a last erroneous reference.

ACTION IN THE MANUSCRIPT: we corrected the last erroneous reference.

REVIEWER 2 | Comment 13: Fig 6: "Recruitment curves: curves are made of 80 data points..." I'm not sure why there is a reference to recruitment curves at the bottom of the legend.

AUTHORS' ANSWERS: this refers to the simulated recruitment of Ia-fibers and motoneurons at increasing amplitude in **Figure 6c,d,e,f,g**. We modified the figure legend to make it clearer.

ACTION IN THE MANUSCRIPT:

We modified the legend of **Figure 6:**

Simulated Ia-fiber and motoneuron recruitment curves (panels c,d,e,f,g): curves are made of 80 data points [...]

REVIEWER 2 | Comment 14: Fig 7: d-g Incorrect descriptions?

AUTHORS' ANSWERS: we thank the reviewer for noticing this.

ACTION IN THE MANUSCRIPT: we corrected the legend of **Figure 7** also according to the new data.

Figure 7. Patterns of muscular responses elicited during high-frequency stimulation of the cervical spinal cord of monkeys. **a** Diagram of the presumed engaged pathways during high-frequency stimulation when muscular responses are modulated and unmodulated respectively. **b, c, d, e** Examples of frequency-dependent modulation of muscular responses. In each panel, the top and bottom EMG traces were recorded in the same muscle and using the same stimulation amplitude (near motor threshold) but different frequencies. **f** Example of absence of frequency-dependent modulation.

g Example of absence of correlation of frequency-dependent modulation between antagonist muscles (the top and bottom traces are simultaneous recordings of the EDC and FDS muscles of Mk-Lo during the same stimulation pulse train). **h Frequency of occurrence of modulation patterns with respect to stimulation frequency.** All the patterns recorded in all the muscles of the 4 animals in which high-frequency stimulation was tested were included in the analysis (n=80 patterns at 10 Hz, n=39 patterns at 20 Hz, n=75 patterns at 50 Hz, n=72 patterns at 100 Hz). **i Same as h, but with respect to electrode position (n=132 patterns for rostral electrodes, n=66 patterns for intermediate electrodes, and n=68 patterns for caudal electrodes).**

Reviewer #3

This is generally a very thorough study, although the results are not especially surprising and could largely be expected, derived from long known anatomy and connectivity. The significant added value is the careful parameter-ization of the models used. Generally, this description of the models is excellent but there are a few details that seem to be missing:

AUTHORS' ANSWERS: we thank the reviewer for the appreciation of our work. We strived to be precise and provide enough details for our work to be reproducible and a tool for the community. We agree that our results are in line with expectations from anatomy. Nevertheless, we believe it was important to carefully demonstrate such results with a combined modelling/experimental approach to provide arguments sensitive to a large scientific audience given the recent hype on this type of approach and perhaps too fast adoption in clinical applications.

REVIEWER 3 | Comment 1: Were different size motoneurons included in the simulation -i.e., consistent with the different input resistances of small and large (slow and fast) motoneurons and the size principle. This is not discussed, suggesting a single dimension MN model was used. Indeed this seems to be the case in one section. This is in contrast the the Poisson drawn synapse distributions that are used to manage variability. The rationale for not considering motoneuron size which is an important known variable, and clearly involved in intact motor control, is not elaborated at all, and needs to be explained. This choice seems to limit the usefulness of some measures deriving from the model. It is odd, given the detail elsewhere (e.g., synapse numbers).

AUTHORS' ANSWERS: we fully agree with the reviewer's concerns regarding the influence of the motoneuron size on simulation predictions. In fact, we did consider different sizes in our simulations. Specifically, the sizes of the motoneurons and their dendritic trees were randomized. We described this in the Methods sections but perhaps this was not clear and so we now tried to clarify this point.

Only the analysis of the relationship between number/strength of synapses and EPSP amplitude (Figure 6a,b) was performed with a single motoneuron model, and thus a single motoneuron size. All the other analyses (Figure 2b,f, Figure 6c,e,g,h) were performed with populations of motoneurons of randomized sizes and dendritic trees.

ACTION IN THE MANUSCRIPT:

We modified the **RESULTS:**

Paragraph Realistic model of the cervical spinal cord:

The trajectory models were based on documented morphological analyses^{21,25,33}, and numerical instances were generated at random using stochastic parameters elaborated from these analyses. **The diameters of the fibers and motoneurons were also randomized**, and motoneurons included dendritic arborizations represented by binary trees of tapered cylinders **also chosen at random (see Methods)**. Their electrical behavior in response to extracellular stimulation was emulated with neurophysical compartmental models^{32,34,35} using NEURON v7.5³⁶.

REVIEWER 3 | Comment 2: The stimulation used in the simulation phases seems to be monophasic from the (very brief) simulation description. However, all useful stimulation is biphasic for charge balance(as was used in the experimental component of the study). The

duration patterns of phases can be varied as parameters (McIntyre and Grill) to alter selectivity and effect. If pulses in simulation were not monophasic this needs to be stated and better explained.

AUTHORS' ANSWERS: we thank the reviewer for pointing out this important lack of information. In our experiments, we employed asymmetric, charge-balanced, biphasic stimulation pulses, with the cathodic phase first. The cathodic phase always lasted 200 μ s-long, and in order to minimize the influence of the anodic phase, it was 4 times as long as the cathodic phase, with an amplitude divided by 4 (Merrill 2005).

Therefore, in our simulations, we only modelled the cathodic component, as is common practice in neurophysical modelling (McIntyre et al. 2002, Capogrosso 2013, Rattay 2000).

We added new explanation for this in the manuscript and added (Merrill 2005) and (Grill 1997) to the references.

ACTION IN THE MANUSCRIPT:

We modified the **METHODS:**

Paragraph *Extracellular stimulation:*

In the experiments, we used asymmetric biphasic stimulation pulses (described below). These were designed to minimize the effect of the anodic phase (employed uniquely for charge-balance). Therefore, in the simulations, we modelled 200 μ s-long monophasic square pulses corresponding to the cathodic phases of the experimental biphasic pulses. This was chosen since the anodic phase, being 4 times smaller than the cathodic phase, had minimal (if any) influence on the emergence of action potentials during the simulations.

These monophasic pulses were simulated by transiently driving the batteries of NEURON's extracellular mechanism to appropriate voltages for each modelled compartment. These were obtained by multiplying the values computed with the FEM and interpolated at the appropriate positions in the volume conductor model by the desired stimulation amplitude. The rise and fall of the voltage transients were linear, lasting 2 μ s.

Paragraph *Acute Electrophysiology:*

All acute electrophysiology was performed under propofol sedation (continuous intravenous injection, 5 ml/kg/h) that minimizes the impact of anesthesia on the responses to spinal cord stimulation⁷⁹. Trains of biphasic electrical pulses were delivered at low frequency (0.67 Hz) through a single active site at a time. We used charge-balanced, asymmetric, cathodic-phase-first square pulses⁸⁰, with a cathodic phase of 200 μ s and a balanced anodic phase of 800 μ s and 4 times lower amplitude. Stimulus waveform influences the properties of neural recruitment^{34,81}: we chose a waveform minimizing the influence of the anodic component while ensuring charge-balance, as recommended for safety reasons.

REVIEWER 3 | Comment 3: Muscle selectivity measures are elaborated in the methods but as far as I can tell this data is very briefly and oddly presented, simply as a brief call-out to Figure 4d, and is little interpreted, if at all. These selectivity data for muscle are important and should be presented in more detail and carefully unpacked for the reader. The best individual muscle selectivity possible is a key for fractionated control. If these stimulations only permit group effects of a myotome, this should be clearly stated. This issue is never discussed or presented with any real clarity. The way that muscle selectivity indices are used and presented is currently unclear, and potentially obfuscating, which I do not believe is the authors intent.

AUTHORS' ANSWERS: we thank the reviewer for pointing this out. This comment is also in line with the other reviewers and we made several changes throughout the entire manuscript to more correctly address the selectivity problem.

Briefly we believe that our combined data shows that, at rest, recruitment of the dorsal roots does not produce synergistic movements. Instead the muscle recruitment curves from lateral electrodes are explained by the rostro-caudal distribution of myotomal innervation. While it is possible to engage upper arm motor nuclei independently, this is not the case for the forearm and hand muscles. However, this is not the purpose of EES, which is rather to direct the artificial excitation to spinal segments naturally active during task execution (Capogrosso 2016, Capogrosso 2018, Wagner 2018).

To complement these observations and relative discussion we:

- 1) added new text explaining and discussing the selectivity at rest with single pulses;
- 2) added new analyses showing evidence that bursts of stimulation engaged proprio-spinal circuits segmentally localized;
- 3) added a new dataset in a behaving chronically implanted animal showing that EES seems to engage synergistic muscles during movement perhaps because of the effects of heteronymous afferent connections or spinal circuits that are more prominent during awake conditions;
- 4) added new data in humans showing evidence of similar rostro-caudal recruitment properties.

We modified the **TITLE:**

COMPUTATIONAL AND EXPERIMENTAL ANALYSIS OF THE RECRUITMENT OF UPPER-LIMB MOTONEURONS WITH EPIDURAL ELECTRICAL STIMULATION OF THE PRIMATE CERVICAL SPINAL CORD

We modified the **ABSTRACT:**

Here, we combined a realistic computational model of the cervical spinal cord with experiments in macaque monkeys to explore the mechanisms of **upper-limb motoneuron recruitment with EES** and characterize the selectivity of cervical interfaces. **We show** that lateral electrodes **produce a segmental recruitment** of arm motoneurons **mediated by** the direct **activation of sensory afferents, and that muscle responses to EES are modulated during movement**. Intraoperative recordings suggested similar properties in humans **at rest**. **These modelling and experimental results support** the design of neuro-technologies to improve arm and hand control in humans with quadriplegia.

We modified the **INTRODUCTION:**

[...] Electrophysiological experiments in monkeys and humans **revealed** that: **a) cervical motoneurons are recruited trans-synaptically; b) lateral electrodes recruit arm and hand muscles with a rostro-caudal order that reflects the segmental innervation of upper-limb motor nuclei in the cervical spinal cord; and c) EES modulates the activity of arm muscles during functional movements performed by monkeys, and this modulation is dependent on the muscle and phase of the movement.**

These model and experimental results provide a conceptual framework to design novel EES technologies to improve upper-limb motor control after cervical SCI.

We modified the **RESULTS**:

Paragraph Computational analysis of the primary targets of EES applied to the cervical spinal cord:

As expected, our results indicate that Ia-afferents running in the targeted root (C6) are recruited at significantly lower stimulation amplitudes than those running in the adjacent or more distal roots (C5, C7, C8 and T1, **Figure 2c**). **This means that near full and exclusive recruitment of individual roots can be achieved with individual monopolar lateral electrodes.** [...]

Paragraph Experimental recruitment of cervical motoneurons with epidural stimulation in macaque monkeys

Laterally-positioned electrodes.

We first analyzed the muscular recruitment induced by the lateral electrodes of the electrode arrays (available for the 5 animals). The results are presented in **Figure 4**.

The obtained muscular recruitment profiles are not indicative of a recruitment pattern based on muscle synergies⁴⁰. Instead, they clearly indicate a segmental, rostro-caudal recruitment pattern which reflects the distribution of the upper-limb motor nuclei in the spinal cord (Figure 3).

We added a **new Figure 8** (previous Figure 8 is now Figure 9) and described the results of the corresponding experiment in a

New paragraph Continuous stimulation of the cervical spinal cord during functional, three-dimensional arm movements:

To verify whether muscle activity and movement could be enhanced and modulated during a functional three-dimensional task, we trained one monkey (Mk-Sa) to freely reach, grasp and pull an object using a robotic framework that we designed for this purpose⁴⁶. We investigated how the animal's arm kinematics and EMGs were influenced by the delivery of continuous EES at 50 Hz (typical stimulation frequency in therapeutic applications). We used a lateral rostral electrode targeting the C6-C7 roots and a lateral caudal electrode targeting the C8-T1 roots.

Continuous EES immediately and reliably evoked muscle responses in the upper-limb muscles (Figure 8a,b). However, the amplitude of these responses was modulated depending on the muscle and the movement phase considered (Figure 8b). For example, responses in the triceps muscle were enhanced during reaching and strongly suppressed during grasping and pulling.

The two contacts engaged distinct muscles (Figure 8d). Coherently with its position, the C6-C7 contact mostly engaged the triceps. On the other hand, the C8-T1 contact mainly engaged the triceps and the flexor carpi radialis (FCR), which, again, was consistent with its position. However, this contact also engaged the biceps, which is more rostrally innervated (C5-C6 segments). Note that the biceps is a synergistic flexor muscle of the FCR, which may explain the emergence of the observed muscle responses (see Discussion). Overall the modulation was reduced during the grasping and pulling phase (Figure 8d, right polar plot).

Finally, we performed a principal component analysis (PCA) on a selection of upper-limb kinematic parameters (see Methods) recorded while the monkey performed the task with or without stimulation. Albeit we tuned the stimulation amplitude to a level that did not impair the animal's ability to perform the task, our analysis revealed that EES did influence arm kinematics. The affected parameters were differentially influenced by the two electrodes (Figure 8e). The PCA shows that kinematics parameters moved towards different directions in the PC space under the effect of the two electrodes. Interestingly, the rostral contact (targeting the triceps) enhanced the peak velocity of the elbow angle, as expected from an electrode engaging elbow extensors. Furthermore, we also observed a positively monotonic relationship between the stimulation amplitude and the amplitude of the modulation of the kinematic parameters (Figure 8e).

We modified the **DISCUSSION**:

Paragraph Stimulation specificity and lead design:

[...] Specifically, the wide spatial separations between adjacent dorsal roots at the cervical level (several millimeters), which is consistent across subjects, should allow robust selective recruitment of individual roots, but the intermingling of different fiber populations within these roots may limit the ability to engage specific motor nuclei.

Similarly to EES of the lumbosacral spinal cord⁵⁸, the recruitment patterns that we obtained during single pulse experiments were explained by the rostro-caudal innervation of motor nuclei rather than by the recruitment of spinal circuits implementing muscle synergies⁴⁰. Given the near perfect segmental separation between the biceps and triceps motor nuclei, a segmental recruitment pattern seems sufficient to selectively promote elbow extension or flexion movements. However, given the overlapping innervation of forearm and hand muscles, selective facilitation of individual finger muscles cannot be achieved with our type of implant. By being closer to the targeted afferents, it may seem that subdural implants may improve the recruitment selectivity, but a recent report⁵⁹ of the recruitment pattern induced by lateral subdural electrodes in monkeys indicated similar limitations to those outlined in the present work. Nevertheless, high-density electrode arrays and current steering techniques, potentially enabling the selective recruitment of structures as fine as individual dorsal rootlets, remain possible options to be explored to further improve the stimulation specificity.

New paragraph Cervical EES during voluntary movement:

We observed a segmental specificity in the recruitment of motoneurons following single pulses of low frequency cervical EES at rest. However, during movement execution, spinal circuits continuously receive supra-spinal, proprio-spinal and

natural sensory inputs which modify the integration of the artificial inputs induced by EES^{12,18,58}. Thus, the direct activation of the same pre-motor elements may modulate different motor nuclei than during resting conditions depending on the specific motor task and movement phase. This task- and phase-dependent modulation can actually improve efficacy of EES. Indeed, EES should facilitate the activity of muscles naturally engaged during the execution of voluntary movements rather than imposing the recruitment of specific muscles^{6,58}. These properties and therapeutic benefits have been demonstrated for lumbosacral EES, but remained unexplored for cervical EES.

We studied the modulation exerted by cervical EES during a typical reaching, grasping and pulling task performed by an intact monkey chronically implanted with our custom-made electrode array. Contrary to resting conditions, the modulation exerted by rostral and caudal contacts only partly reflected the rostro-caudal innervation of the upper-limb muscles (Figure 8d). Specifically, the activity of a rostrally-innervated muscle was consistently and significantly enhanced by the stimulation from the caudal contact. However, this muscle (biceps) was a flexor synergist of a caudally-innervated muscle (flexor carpi radialis, FCR) which was also engaged by the stimulation. This suggests that during the execution of a voluntary task, stimulating the primary afferents of upper-limb muscles (e.g. FCR) may enhance the motor activity of synergistic muscles, possibly via the contribution of heteronymous excitatory connections. Perhaps, at rest, these secondary pathways may have only secondary effects compared to strong monosynaptic components. An increase of coordinated excitatory inputs from circuits underlying muscle synergies, which are abundantly innervated by sensory afferents⁶², might also be occurring. Finally, we found that the modulation of muscular activity was dependent on the movement phase (Figure 8d), as is the case with lumbosacral EES^{12,58}.

These combined results suggest that, as in the case of the lower limb^{1-3,5,8}, understanding the interplay of residual descending commands with proprio-spinal inputs, sensory feedback and the integrative properties of cervical spinal circuits will be key for the development of EES protocols that successfully ameliorate arm and hand deficits in people with spinal cord injury.

Paragraph Conclusions:

By combining computer simulations and electrophysiology in monkeys and humans, we have provided evidence that EES applied to the cervical spinal cord recruits motoneurons trans-synaptically via the direct excitation of sensory afferents and following their segmental innervation in the cervical spinal cord. Our results indicate that lateral contacts are necessary to achieve this segmental selectivity and that current human electrode arrays are unfit to achieve these results. They also show that the modulation of muscle activity exerted by EES during movement is movement-phase-dependent and likely promotes upper-limb muscle synergies. We believe that these combined results establish a pathway for the development of neuro-technologies for the restoration of arm movements in people with cervical spinal cord injury.

REVIEWER 3 | Comment 4: The authors describe the spinocerebellar tract (SCT) but should distinguish if they mean dorsal or ventral SCT. Further, they should explain why reticulospinal, rubrospinal and vestibulospinal tracts in the lateral funiculi were omitted from simulations. The authors uses posterior/anterior and dorsal/ventral root terminology interchangeably. Likely they should state the equivalence once, pick one terminology and stick with it.

AUTHORS' ANSWERS: we apologize for the lack of clarity. Regarding the spino-cerebellar tract (SCT), we were referring to the dorsal SCT. Regarding the cortico-spinal tract (CST), we were referring to the lateral CST. The dorsal rubrospinal tract and the dorsal reticulospinal tract were not explicitly represented, but they can be thought of being embedded into the lateral CST since they are small and adjacent to it. The other tracts were not represented because their depth in the spinal cord and their distance from the stimulation electrodes are such that their recruitment could be predicted to be null or negligible on the basis of previous studies without the need for further simulations (Coburn and Sin 1985, Lempka et al. 2015).

Regarding the terminologies dorsal/ventral VS posterior/anterior, we now consistently use dorsal/ventral.

ACTION IN THE MANUSCRIPT:

We modified the **RESULTS:**

Paragraph Realistic model of the cervical spinal cord:

[...] We then elaborated realistic tridimensional trajectory models for group-Ia afferent fibers ($A\alpha$ diameter class), group-II afferent fibers ($A\beta$ diameter class), motoneurons with their efferent axons ($A\alpha$), **dorsal** spinocerebellar tract fibers (ST, $A\beta$), **lateral** corticospinal tract fibers (CST, $A\beta$) and dorsal column fibers (DC, $A\beta$) (**Figure 1c,d**). **Fibers from other tracts were not represented due to their depth in the spinal cord and their distance from the stimulation electrodes: a negligible recruitment could be predicted on the basis of previous studies^{30,31}.** [...]

We modified the **METHODS:**

Paragraph Dorsal columns fibers / Dorsal spinocerebellar tract fibers / Lateral corticospinal tract fibers:

These fibers were modelled similarly to DR- $A\beta$ -fibers but they didn't possess a dorsal root branch. They possessed a rostro-caudal branch running in the dorsal columns, **dorsal** spinocerebellar tract and **lateral** corticospinal tract respectively. They also possessed a series of collateral branches. The dorsal branches of the dorsal column fibers were restricted to the outermost layer of the dorsal columns, where likelihood of recruitment is highest⁷⁸. **Lateral corticospinal** tract fibers were meant to represent the large-diameter axons directly connecting to spinal motoneurons and originating from large Layer V cells in the primary motor cortex. Smaller diameter fibers were not represented. **Dorsal spinocerebellar** tract fibers represented $A\beta$ sensory fibers originating from caudal spinal segments.

REVIEWER 3 | Comment 5: It would be really helpful in Discussion to elaborate what -if anything- is surprising in the data and simulations. I believe the study needed to be done, and is generally very well done, but to me is largely confirmatory of the reasonable expectations of any physiologist. Perhaps the lack of difference between A alpha and beta fiber thresholds in simulation is a surprise?

AUTHORS' ANSWERS: we agree with the reviewer that we should have done a better work at emphasizing the novel aspects in the work. We now have substantially changed the results and discussion sections in multiple parts to highlight the novel insights. In addition to what is already

reported above, we also expanded our results on the $A\alpha/A\beta$ recruitment as the reviewer suggested.

ACTION IN THE MANUSCRIPT:

We modified the **RESULTS:**

Paragraph *Laterally-positioned electrodes.*

Surprisingly, **for lateral electrodes**, the larger diameter of the $A\alpha$ -DR-fibers compared to $A\beta$ -DR-fibers does not seem to confer them a substantially lower excitation threshold, as could be expected^{37,38} (**see Discussion**).

Paragraph *Medially-positioned electrodes.*

[...] while that for motor axons to be as high as ~9 times this value (**Figure 2f**). **In addition, contrary to lateral stimulation, the threshold of $A\beta$ -DR-fibers to medial stimulation is significantly higher than that of $A\alpha$ -DR-fibers (almost twice as high), as could be expected for an electrode position appreciably far from the fibers^{37,38}.**

We modified the **DISCUSSION:**

Paragraph **Detailed modelling of EES of the cervical spinal cord:**

[...] **Our hybrid model of cervical EES in monkeys led to an unexpected result: we found that the recruitment of $A\beta$ -DR-fibers following lateral stimulation occurred at stimulation amplitudes only slightly higher than those necessary to recruit $A\alpha$ -DR-fibers, albeit their average diameter was only two thirds that of the $A\alpha$ -fibers. Comparatively, the difference in excitation threshold was much more pronounced for medial stimulation, closer to what is usually expected^{37,38}. This result is likely due to the combination of 1) the close proximity between the lateral electrodes and the dorsal roots (Figure 2a), which was a consequence of the fine reconstruction of the spinal cord model based on empirical morphological data, and 2) the small width of the electrodes. The latter were indeed approximately 700 μm wide, while the average distance between consecutive nodes of Ranvier of $A\alpha$ -fibers and $A\beta$ -fibers were respectively 1400 μm and 900 μm . Thus, $A\beta$ -fibers were more likely to possess a node directly below the electrode surface, which might have given them a surplus of excitability compared to $A\alpha$ -fibers, albeit not as large as the surplus coming from the larger diameters of the $A\alpha$ -fibers. This effect might be less prominent in humans, where, given the larger size of the human spinal canal, the proximity between electrodes and dorsal roots should not be as close. However, the inability to selectively recruit the $A\alpha$ dorsal root fibers with lateral electrodes would be detrimental to both sensory⁵¹ and motor applications⁵² of spinal cord stimulation, so we believe this point deserves further investigations.**

REVIEWER 3 | Comment 6: Line numbers would be helpful in review. Some places the authors use 'columns' when I think they mean 'segments' e.g., on page 4.

AUTHORS' ANSWERS: we apologize for not having used line numbers, we now have introduced line numbering in the new version of the manuscript.

Reviewers' Comments:

Reviewer #1:

Remarks to the Author:

Regarding the points in my original review:

1. There is very little analysis of whether any of the movements produced by ESS are functional, especially since there does not appear to be fine-grained selectivity between muscles innervated from the same level (e.g. EDC, FDS and APB in Fig. 4). Kinematics would be useful here, alongside EMG data, or at least some effort to demonstrate that the EMG patterns correspond to functional muscle synergies.

The authors have added new data from a single awake animal. To be honest I find this data somewhat underwhelming - the authors show that reaching movements are perturbed by stimulation. I have some specific questions related to this new analysis:

1a: Can the authors confirm that some consideration was paid to possible stimulus artefacts in the EMG traces - i.e. that these were removed before computing EMG energy? Given the time-scale in Fig. 8b it is hard to assess. In addition, it would be useful to know whether and with what latency are the EMG responses (in the case of facilitation) time-locked to the stimuli (it appears so in Fig. 8b). This would help reassure that the changes are not caused by some indirect effect (e.g. a painful sensation altering the way the animal moves).

1b: Did the statistical tests correct for multiple comparisons (over EMGs and bins)?

1c: Were any statistical analyses carried out on the altered kinematics? Ensuring statistics are robust is particularly important since this data is coming only from a single animal.

More generally, I am somewhat concerned that more robust movements cannot be demonstrated. The introduction speaks of 'engaging specific muscles with EES' and 'facilitating the execution of arm and hand function in people with cervical SCI', yet the results demonstrate only a relatively modest perturbation of an ongoing movement. The authors say that they "did not tune amplitudes to levels necessary to observe movements but just muscle contractions" in the anesthetized experiments, and in the awake experiments they "tuned the stimulation amplitude to a level that did not impair the animal's ability to perform the task". I am left wondering why they did not attempt to stimulate at higher intensities, at least to find out if functional movements could be produced. Was there some concern over causing pain/discomfort in the intact animals? Perhaps this would not be a problem for a therapy geared towards complete SCI, but since the authors talk about interactions with "residual volitional control" this seems equally a concern for incomplete SCI. Based on the data presented, I find it difficult to believe that ESS within the range delivered in this study would be clinically useful in either patient population. Moreover, if stimulation has to be tuned differently to yield larger effects, will such stimulation still have the necessary specificity? For these reasons, I feel the revision has not yet adequately addressed this original question.

2. Related to 1 - It is stated that in the introduction that "intraspinal microstimulation, which also engages motoneurons via pre-synaptic pathways, have reported low reproducibility and limited specificity of arm muscle recruitment when applied to the cervical spinal cord of monkeys, raising questions on the applicability of this technology to the upper-limb". However I do not see any comparison that convinces me that the selectivity obtained by ESS is better than that produced by intraspinal stimulation.

This has been addressed by removing this claim.

3. The experiments are conducted under anesthesia/sedation, however experiments in two animals used chronic implants. Do the authors have any data to show that similar responses can be obtained in awake animals?

This has been partially addressed by the new data in the awake animal. It is not clear that the responses in the awake animal are exactly similar to those under sedation, which again rather limits the conclusions that can be drawn (e.g. about selectivity) from these data. Nevertheless I agree that "understanding the interplay of residual descending commands with proprio-spinal inputs, sensory feedback and the integrative properties of cervical spinal circuits will be key for the development of EES protocols."

4. What was the long-term stability of effects elicited by chronic implants (e.g. thresholds, selectivity etc.)? Was any histology performed to show that the spinal cord was not damaged by either the stimulation or the implant itself?

I did not see any response to my question "Was any histology performed..."

5. The variety of temporal patterns following high-frequency stimulation (Figure 7) is interesting, but this section is very descriptive without any consideration of possible mechanisms. For example, are there any systematic differences between proximal/distal muscles, or rostral/caudal stimulation sites? Are the 'alternation' patterns indicative of CPG-like behavior, or related to tremor mechanisms – this could perhaps be addressed by looking at synchrony between agonist/antagonist muscles. What is the difference between 'attenuation' and 'suppression', and do these reflect distinct neuronal mechanisms?

This point has been addressed satisfactorily. The finding of a rostral/caudal difference is interesting and points to a possible mechanistic explanation.

6. The human data is nice to have, but there have been previous demonstrations of motor effects of cervical ESS in humans (e.g. Lu et al. *Neurorehabil Neural Repair* 2016). Unfortunately, it was not possible to replicate the key result that lateral stimulation was more selective than medial stimulation. It is stated that the lack of 'maximal EMG amplitudes' meant that selectivity could not be assessed, but there would seem to be ways around this. For example, could the responses be normalized by maximal voluntary contractions (MVCs). Alternatively the authors could compare selectivity in terms of the thresholds for recruiting different muscles, or simply use the un-normalized responses for comparison between medial/lateral stimulation. In the absence of such comparisons, I am not convinced that this section really adds much to the scientific story.

The new data that has been added still does not replicate the key result that lateral stimulation was more selective than medial stimulation, but the relevance of this section has been improved. However, it is claimed that "As observed in monkeys, the muscular recruitment profiles obtained in the five patients tended to reflect a segmental rostro-caudal innervation of the upper-limb muscles". What statistical test was performed to assess this?

The remaining minor points have been addressed satisfactorily.

Reviewer #2:

Remarks to the Author:

The authors have done an impressive job of responding to my comments. Most of the remaining issues are in the new figure and associated text.

Comment 1: Might be worth a reference to the current steering technique.

Comment 2: "These model[ing] and experimental results provide"

"This means that near full and exclusive recruitment of individual roots can be achieved"

I see no evidence to support this statement. There would be ways to quantify the effects, but they've not been done. Short of this, you should stick to a less categorical statement.

Show the same set of muscles in 8 a and b, as they are to be compared directly.

"...responses in the triceps muscle were enhanced during reaching and strongly suppressed during grasping and pulling."

I don't see this at all. Although its activation during pulling is small generally, Tri is clearly more active throughout in b than in a. What am I missing? Is this just a description of the normal suppression of triceps during elbow flexion, rather than a description of the effect of stimulation?

The lower Tri trace in b is fairly confusing. It is not described in the text, and it makes a completely different point from all the other traces. If it is important, describe it and set it apart in the figure more clearly. If it is not, delete it.

"Coherently with its position" > Consistent with its position

Are the bins in c meant to align with the time axes of a and b? They don't quite, but it would be helpful if they did. C is described nowhere in the text. Add some comment.

Panel e adds very little beyond the muscle analysis. Is the rightmost panel intended to show the positive monotonic relation? If so, it is not clear how. There is no description of this in the legend and no direct reference in the text. I'd delete the entire analysis.

Rather than this sentence:

Thus, the direct activation of the same pre-motor elements may modulate different motor nuclei than during resting conditions depending on the specific motor task and movement phase.

I'd suggest:

Thus, the direct activation of a given pre-motor element may modulate different motor nuclei during movement than at rest, in a task-dependent manner.

and that current human electrode arrays[, with their medially-placed contacts,] are unfit to achieve these results.

"...might have given them a surplus of excitability..." Surplus means more than enough. I doubt this is what you mean. Perhaps, "given them greater sensitivity to the stimulation... albeit not as large as that of the larger diameter A α -fibers"?

Comment 5: "this point deserves further investigations." > Investigation

There are many words that are inappropriately hyphenated:

Proprio-spinal, trans-synaptic, neuro-technologies, downward-oriented, cross-sections, Ia-afferents, Ia-fibers, supra-threshold, motor-axons

Reviewer #3:

Remarks to the Author:

The revisions have handled all my comments sufficiently satisfactorily in my estimation.

COMPUTATIONAL AND EXPERIMENTAL ANALYSIS OF THE RECRUITMENT OF UPPER-LIMB MOTONEURONS WITH EPIDURAL ELECTRICAL STIMULATION OF THE PRIMATE CERVICAL SPINAL CORD

Reading guidelines (color code)

AUTHORS' ANSWERS: report the answers to the reviewer's question.

ACTION IN THE MANUSCRIPT: report the actions taken in the manuscript (in the revised text as **bold red characters**).

Reviewer #1

Regarding the points in my original review:

1. There is very little analysis of whether any of the movements produced by ESS are functional, especially since there does not appear to be fine-grained selectivity between muscles innervated from the same level (e.g. EDC, FDS and APB in Fig. 4). Kinematics would be useful here, alongside EMG data, or at least some effort to demonstrate that the EMG patterns correspond to functional muscle synergies.

The authors have added new data from a single awake animal. To be honest I find this data somewhat underwhelming - the authors show that reaching movements are perturbed by stimulation. I have some specific questions related to this new analysis:

REVIEWER 1 | Comment 1: 1a: Can the authors confirm that some consideration was paid to possible stimulus artefacts in the EMG traces – i.e. that these were removed before computing EMG energy? Given the time-scale in Fig. 8b it is hard to assess. In addition, it would be useful to know whether and with what latency are the EMG responses (in the case of facilitation) time-locked to the stimuli (it appears so in Fig. 8b). This would help reassure that the changes are not caused by some an indirect effect (e.g. a painful sensation altering the way the animal moves).

AUTHORS' ANSWERS: This is a valid point, in fact over the years we developed robust procedures to analyze and process EMG data during SCS in rats, monkeys and humans (Capogrosso 2013 JNS, Wenger et al 2015 Nature Medicine, Capogrosso 2016 Nature, Capogrosso 2018 Nature Protocols and Wagner 2018 Nature). As the reviewer correctly guessed stimulation artifacts were present almost in all traces. We removed these artifacts by blanking thanks to our synchronized recordings of EMG traces and stimulation trigger output (using TDT RZ2 system). In monkeys (and humans) this procedure is very easy from a signal processing standpoint because latencies between stimulation pulses and evoked responses ranges several milliseconds (2-10 ms in our case), depending on the distance from the stimulation site of each muscle (we sample EMGs at 24 kHz). In consequence we restricted our analysis only to blanked EMG signals and specifically to the 20ms following the stimulation pulse. We highlighted this temporal scale on the new version of the figure and zoomed response that we displayed for the Triceps muscle. New Figure 8b, shows evoked responses grouped by time of occurrence (the entire movement is divided in 10 bins) and overlapped. Thanks to the change proposed by the reviewer we hope that this figure is now much clearer and most of all that highlights the fact that evoked responses have a very consistent shapes throughout the movement and only vary in amplitude (which is an additional sanity check since stimulation amplitude was constant over time, hence artefact intensity did not change).

ACTION IN THE MANUSCRIPT: We significantly modified Figure 8

REVIEWER 1 | Comment 2: 1b: Did the statistical tests correct for multiple comparisons (over EMGs and bins)?

AUTHORS' ANSWERS: The reviewer raised another very important point. In fact, the analysis that we reported did not require multiple-comparison correction of the p-value because we did not test for differences between the bins. Tests were performed independently for each bin and only tested for stim-on vs stim-off distribution in each bin. However, upon suggestion with the reviewer we realized that it would be important to show that stimulation bins had a significantly higher values during reach than during pull. Therefore we introduced a new multi-group comparison test in the new Figure8 which is corrected for multi-group

ACTION IN THE MANUSCRIPT: We modified Figure8 and introduce multiple-comparison post-hoc corrected test

REVIEWER 1 | Comment 3: 1c: Were any statistical analyses carried out on the altered kinematics? Ensuring statistics are robust is particularly important since this data is coming only from a single animal.

AUTHORS' ANSWERS: The reviewer is correct. However, following up on comments from reviewer 2 as well as realizing that this data is moving the papers towards a direction that we did not intend we decided to remove the kinematic analysis. We felt that this was bringing the paper too much in the direction of demonstrating clinical efficacy which is beyond the scope of this manuscript. However, for transparency we attached here for the reviewer the results of our statistical analysis that we did not include in the new version of the manuscript.

ACTION IN THE MANUSCRIPT: Panel e has been removed. Attached statistical analysis on kinematic variables for the reviewer interpretation. We believe that the 2 contacts are clearly modulating different kinematic variables. Moreover the effects on kinematics seem to depend from stimulation position and intensity. This suggest that a tuning of stimulation parameters similar to what is done in the lumbosacral spinal cord is necessary to optimize potential clinical

benefits Capogrosso et al. 2018 Nature Protocols, Wagner et al 2018 Nature).

REVIEWER 1 | Comment 4: More generally, I am somewhat concerned that more robust movements cannot be demonstrated. The introduction speaks of ‘engaging specific muscles with EES’ and ‘facilitating the execution of arm and hand function in people with cervical SCI’, yet the results demonstrate only a relatively modest perturbation of an ongoing movement. The authors say that they “did not tune amplitudes to levels necessary to observe movements but just muscle contractions” in the anesthetized experiments, and in the awake experiments they “tuned the stimulation amplitude to a level that did not impair the animal’s ability to perform the task”. I am left wondering why they did not attempt to stimulate at higher intensities, at least to find out if functional movements could be produced. Was there some concern over causing pain/discomfort in the intact animals? Perhaps this would not a problem for a therapy geared towards complete SCI, but since the authors talk about interactions with “residual volitional control” this seems equally a concern for incomplete SCI. Based on the data presented, I find it difficult to believe that ESS within the range delivered in this study would be clinically useful in either patient population. Moreover, if stimulation has to be tuned differently to yield larger effects, will such stimulation still have the necessary specificity? For these reasons, I feel the revision has not yet adequately addressed this original question.

AUTHORS’ ANSWERS: Once again we thank the reviewer for his/her constructive criticism. We do agree that the data reported does not clearly demonstrate a potential clinical efficacy of EES.

However, while reviewing the manuscript we also realized that demonstrating such efficacy goes far beyond the goals of this manuscript. We have realized that this is partly our fault for the way we framed the goals of our study in the introduction. We now modified the introduction and the discussion to better state our goals and avoid misunderstandings.

We will introduce in this section changes to the introduction. Please refer to the answer to the next question to address the comments about potential clinical limitations, pain and discomfort.

Modification in the introduction.

Often in medicine, once the efficacy of certain therapies is demonstrated (e.g. DBS for Parkinson’s Disease) researchers work to understand the mechanisms of such therapy in order to further optimize it. Indeed, this has been the case also for EES. After the first experimental demonstration in animals and partly in humans, in the last 10 years a large corpus of literature has been published on the potential mechanisms underlying the assistive effects that EES had on the restoration of lower limb motor control (Rattay 2000, Minassian 2004, Gerasimenko 2006, Capogrosso 2013, Moraud 2016, Formento 2018).

We now identified the critical features that are necessary for EES to be effective:

- 1) EES primarily recruits dorsal root sensory fibers, via which it directly engages spinal motor circuits trans-synaptically;
- 2) targeting individual dorsal roots allows to some extent to restrict the exerted modulation to the motor nuclei located in the corresponding spinal segments;
- 3) thanks to this trans-synaptic recruitment during movement, the exerted modulation is movement phase dependent because it interacts with both residual descending control and spinal reflexes. This is critical to enhance voluntary motor control.

Ignoring that such progresses have been made and directly proceed with a clinical or pre-clinical demonstration of the efficacy of EES may lead to poor clinical outcomes (see Lu 2016 Neurorehabil Neural Repair). Hence, we thought that it was important to first demonstrate whether and to what extent these principles applied also to EES of the cervical spinal cord. We thought that these results could be used to understand how cervical EES technologies should be designed to maximize these properties in future studies testing efficacy and we believed that this would make efficacy demonstration more efficient.

Indeed, we found that current clinical leads are unfit to maximally exploit the critical characteristics of EES that made EES of the lumbosacral spinal cord so successful. And that leads must be placed laterally to obtain selective recruitment of single dorsal roots afferents. A new study testing directly for clinical efficacy of EES can now leverage the results of our study.

In fact, following up on the results of this manuscript we did perform a study in monkeys with SCI to verify that lateral EES of the cervical spinal circuits could support the execution of arm and hand movements after injury. We are still working to process and appropriately document the results of that follow up study. As the reviewer can imagine, experimental methods, subject population and goals of that study are far outside and beyond the scope of this manuscript.

Nevertheless, after discussing with the editor, we thought that we could provide the reviewer with a confidential video with preliminary results on the potential assistive effects of EES on the production of reaching and grasping after SCI. We hope that while demonstrating efficacy is outside the scope of this specific work, it may help addressing some of the questions that the reviewer raised on potential clinical benefits of EES of the cervical spinal cord. The videos shows the amelioration of reaching and grasping in two different monkey with arm paralysis (C6 SCI, and C7 lesion) Video can be found at the following link:

C6 video: <https://www.dropbox.com/s/0khvpu4bsevfkff/Mk-fullvideo-C6.avi?dl=0>

C7 video: <https://www.dropbox.com/s/ntb0t3e24x1k1ms/Mk-fullvideo-C7.avi?dl=0>

ACTION IN THE MANUSCRIPT: we modified the introduction to clarify the goals of our work.

INTRODUCTION:

Two decades of preclinical and clinical studies have demonstrated that the delivery of EES to the lumbosacral spinal cord can reactivate spinal sensorimotor circuits after SCI¹⁻⁹. Computational and experimental studies conducted in animal models and humans¹⁰⁻¹⁶ have brought evidence that EES applied over the lumbosacral spinal cord primarily engages large myelinated afferent fibers running in the dorsal roots and dorsal columns of the spinal cord. These fibers form synaptic connections with **lumbosacral** spinal interneurons and motoneurons, thereby constituting a gateway to the motor circuits controlling leg muscles^{17,18}.

Due to their branching morphology¹⁹⁻²¹, the artificial recruitment of these fibers supplies synaptic inputs to multiple spinal segments. This divergence of inputs may limit the ability to **modulate** specific **motor nuclei** with EES, which could be particularly detrimental to applications aiming at restoring arm and hand function. Nonetheless, the distribution of sensory afferents in the dorsal roots can be exploited, by targeting individual roots, to direct the modulation exerted by EES towards specific motor nuclei. Indeed, **evidence suggests that stimulating individual roots predominantly affects the motor nuclei located in the corresponding spinal segments, notably via group-Ia afferent fibers, which form monosynaptic excitatory connections with motoneurons^{10,16,22}. In the context of**

paraplegia and locomotion, this principle was used to define stimulation protocols that target individual lumbosacral dorsal roots independently, with timings corresponding to the natural dynamics of the segmental motoneuronal activity **underlying** walking. Such spatiotemporal patterns of targeted EES induced immediate mitigation of lower-limb motor deficits in both animals and humans^{3,8,9}. **Furthermore, the trans-synaptic nature of the modulation exerted by EES on lumbosacral motor nuclei was identified as a central factor in these functional outcomes. Trans-synaptic recruitment increased the activity of naturally active motor nuclei without inducing parasitic activity in motor nuclei naturally silent during movement¹⁸.**

Similarly to the lumbosacral circuits, the **cervical** circuits controlling upper-limb muscles **are organized with dorsal root afferent fibers innervating spinal motoneurons and interneurons, and with segmentally-clustered motor nuclei^{23,24}. This suggests that the above mechanisms underlying the capacity of EES to support neural activity of the lumbosacral spinal segments may translate to the cervical segments. If so, EES of the cervical spinal cord could also facilitate arm and hand function in people with cervical SCI²⁵.**

Here, **we investigated the mechanisms underlying the recruitment of arm and hand motoneurons with EES of the cervical spinal cord and we inferred the properties that epidural implants should have to direct the modulation exerted by EES towards specific upper-limb motor nuclei. To this end**, we implemented a detailed computational model capable of estimating the recruitment of various populations of cervical nerve fibers and neurons in response to single pulses of EES. Numerical simulations suggested that cervical EES primarily recruits large myelinated afferent fibers in the dorsal roots and in the dorsal columns and is unlikely to recruit motor axons directly. The model **also suggested** that the selective recruitment of individual dorsal roots (and thus, to a lesser extent, of upper-limb motor nuclei) **is achievable with monopolar epidural electrodes, but** contingent on the precise mediolateral and rostro-caudal **position of the electrodes**. Electrophysiological experiments in monkeys and humans revealed that: a) cervical **EES recruits upper-limb** motoneurons trans-synaptically; b) lateral electrodes **engage upper-limb motor nuclei** with a rostro-caudal order that reflects **their** segmental innervation in the cervical spinal cord; and c) **the modulation of upper-limb muscle activity exerted by EES in behaving monkeys is movement-phase dependent.**

These **modeling** and experimental results provide a conceptual framework to design novel EES technologies to improve upper-limb motor control after cervical SCI.

REVIEWER 1 | Comment 5: The experiments are conducted under anesthesia/sedation, however experiments in two animals used chronic implants. Do the authors have any data to show that similar responses can be obtained in awake animals?

This has been partially addressed by the new data in the awake animal. It is not clear that the responses in the awake animal are exactly similar to those under sedation, which again rather limits the conclusions that can be drawn (e.g. about selectivity) from these data. Nevertheless I agree that “understanding the interplay of residual descending commands with proprio-spinal inputs, sensory feedback and the integrative properties of cervical spinal circuits will be key for the development of EES protocols.”

AUTHORS' ANSWERS: We are happy that the reviewer share's our beliefs on the path that future investigations should take to optimize clinical efficacy of EES especially for the arm and hand.

Since we believe that previous points regarding the possible emergence of pain or discomfort are linked to this topic we modified the discussion to include these topics:

Pain and discomfort in our study.

First, before proceeding we would like to clarify that in the recruitment curves we used stimulation amplitudes that were way above the values used during the behavioral experiments. However, in the awake monkey we focused on using EES intensity levels that were slightly supra-threshold, because these were typical levels used in clinical trials (Angeli 2018, Gill 2018, Wagner 2018, Lu 2016). Indeed, these parameters were sufficient to produce EMG responses and increase muscle activity as could have been expected by previous work in humans.

Moreover, as the reviewer might imagine, when working with monkeys, especially in the framework of the swiss regulation that forbid water intake regulation, a slight change in the daily practice and habits lead to the animal stopping to work during an experimental session. In fact, in Switzerland, monkey's motivation is purely based on positive reinforcement with food treats while the animals have anyway access to food and unrestricted liquid during the day. EES does induce paresthesia and sensations on the limbs, increasing stimulation intensity in our case lead to the animal to stop working, but we don't know whether this was caused by pain, discomfort or just arising of new sensations that the animal never felt before. Indeed, monkeys tend to hide pain in front of investigators and other threats.

Subjects with incomplete SCI and pain during EES

Despite this, recent clinical trials using EES to support leg function were executed also on subjects with incomplete SCI and spared motor and sensory functions from the legs (Wagner 2018 Nature, Formento 2018 Nature Neuroscience). These studies did not report pain or discomfort in the patients at EES intensities that were necessary to sustain motor function. However, we do acknowledge that each patient might respond differently to EES and that even non-uncomfortable sensations induced by artificial muscle activation can be very uncomfortable for certain patients. Considering the results from these clinical trials we expect that at least a some of the patients with incomplete injuries will be responsive to EES. In consequence, we disagree with the reviewer that EES has potential to fail in this population. Finally, we would like to clarify that we never claimed in our manuscript that the intensities that we used in this study are those that should be used in clinical trials. Instead, we believe that subject-specific tuning of EES intensities will likely be required as we have previously proposed (Wagner 2018 Nature, Capogrosso 2018 Nature Protocols). We believe that the lead design that we proposed in this manuscript and the knowledge gained from our models will accelerate those tuning procedures.

Subjects with complete SCI and EES

Regarding the potential efficacy of EES in people with motor complete spinal injuries, we acknowledge that our study shows that EES is not specific enough to completely drive functional and skilled arm and hand movements. However, as we said in the previous response, that is not the goal of EES. Indeed, when applied to the lumbosacral spinal cord, leg muscle specificity of EES is not sufficient to control the 39 muscles in each leg to produced voluntary locomotion. Yet, patients with complete spinal cord injury regained the ability to move their limb and even walk with EES (Harkema 2011, Angeli 2014, Angeli 2018). We firmly believe that the efficacy of the assistive potential of EES stems from its trans-synaptic nature and on the ability of spared residual fibers (even if minimal) to exploit the artificial excitatory pre-synaptic signals to produce voluntary motor control (Formento 2018 Nature Neuroscience). Therefore, we disagree with the reviewer on the fact that the results from our study suggest that EES won't be

effective in people with motor complete paralysis. On the contrary, we believe that following the guidance provided in our study on how to engage specific cervical segments will maximize the likelihood that EES could work also in people with motor-complete spinal cord injuries and possibly other types of motor disorders.

ACTION IN THE MANUSCRIPT: To address the reviewer's comments on potential clinical application and limitations of EES in motor complete and incomplete patient population we modified the discussion and added a specific paragraph that discusses the potential issues.

New paragraph:

Potential clinical applications and limitations

Our combined results suggest that, as in the case of the lower-limb^{1-3,5,8}, given the limited selectivity in the recruitment of specific upper-limb motor-nuclei, understanding the interplay of residual descending commands with propriospinal inputs, sensory feedback and the integrative properties of cervical spinal circuits will be key for the development of clinical EES protocols to restore arm and hand function. For example, while EES alone might not be sufficient to generate whole-arm functional movements in people with complete arm paralysis, clinical results in people with motor complete lower-limb paralysis suggest that descending pathways spared by the injury may still be able to use EES to produce complex functional movements^{1,59,60}. On the contrary, the assistive potential of EES may be limited in subjects with significant residual motor control and sensory perception, notably by the emergence of pain or discomfort during stimulation. Such undesirable side effects may indeed occur at the stimulation intensities required to obtain relevant motor effects. However, subjects with incomplete motor paralysis and residual sensory function of the lower-limb did not report significant pain or discomfort at EES intensities that were necessary to obtain robust leg movements^{8,11}.

REVIEWER 1 | Comment 6: What was the long-term stability of effects elicited by chronic implants (e.g. thresholds, selectivity etc.)? Was any histology performed to show that the spinal cord was not damaged by either the stimulation or the implant itself?

I did not see any response to my question "Was any histology performed..."

AUTHORS' ANSWERS: we apologize for having forgotten to reply to this point. The answer is "no": unfortunately, no histology was performed. We would like to add a little bit of context. The EMG leads near the shoulder of Mk-Sa got infected after 6/8 weeks from implantation. In consequence of this event we had to explant the entire system because EMGs and SCS array were routed through the same pedestal. The risk was that infection would propagate to the spinal cord thus killing the animal. We then kept working with this animal after the study and this is why we don't have histological analysis with this array. However, like we said, we did not observe any behavioral or motor deficits after electrode implant and explant.

REVIEWER 1 | Comment 7: The human data is nice to have, but there have been previous demonstrations of motor effects of cervical ESS in humans (e.g. Lu et al. Neurorehabil Neural Repair 2016). Unfortunately, it was not possible to replicate the key result that lateral stimulation was more selective than medial stimulation. It is stated that the lack of 'maximal EMG amplitudes' meant that selectivity could not be assessed, but there would seem to be ways around this. For example, could the responses normalized by maximal voluntary contractions

(MVCs). Alternatively the authors could compare selectivity in terms of the thresholds for recruiting different muscles, or simply use the un-normalized responses for comparison between medial/lateral stimulation. In the absence of such comparisons, I am not convinced that this section really adds much to the scientific story.

The new data that has been added still does not replicate the key result that lateral stimulation was more selective than medial stimulation, but the relevance of this section has been improved. However, it is claimed that “As observed in monkeys, the muscular recruitment profiles obtained in the five patients tended to reflect a segmental rostro-caudal innervation of the upper-limb muscles”. What statistical test was performed to assess this?

AUTHORS' ANSWERS: We did not perform a statistical test because we believe that performing parametric statistical tests with only few points (5 in this case) is inappropriate use of statistics. That's why we normally adopt non-parametric statistical tests when appropriate. The minimum number of point necessary to obtain a p-value less than 0.05 is n=6 when using rank-based non-parametric tests. Therefore, non-parametric tests would be non-significant in consequence of the low number of point. We believe that this is a more correct use of statistics. Indeed, since we did not perform tests and the results were not as strong as in the monkey we used the wording “tended to reflect”.

Reviewer #2

The authors have done an impressive job of responding to my comments. Most of the remaining issues are in the new figure and associated text.

REVIEWER 2 | Comment 1: Might be worth a reference to the current steering technique.

AUTHORS' ANSWERS: we apologize to the reviewer, our answer to this interrogation during the previous revision remained unclear.

In this study, we did not employ any current steering technique. All stimulations were monopolar. When talking about “steering the effects of EES towards specific motor nuclei”, what we mean is that by stimulating individual dorsal roots (with monopolar stimulation), the effects of EES are somewhat restricted to (i.e. steered towards) specific motor nuclei (notably those located in the corresponding spinal segments).

ACTION IN THE MANUSCRIPT: we modified the introduction to address comments from reviewer 1 and also made that sentence clearer.

The model **also suggested** that the selective recruitment of individual dorsal roots (and thus, to a lesser extent, of upper-limb motor nuclei) **is achievable with monopolar epidural electrodes, but** contingent on the precise mediolateral and rostro-caudal **position of the electrodes.**

REVIEWER 2 | Comment 2: “These model[ing] and experimental results provide”

AUTHORS' ANSWERS: we thank the reviewer for his suggestion. It was our intention to designate the computational model itself, but we actually agree that it is better to mention the modeling results instead.

ACTION IN THE MANUSCRIPT: we implemented the reviewer's suggestion in the text.

REVIEWER 2 | Comment 3: “This means that near full and exclusive recruitment of individual roots can be achieved”

I see no evidence to support this statement. There would be ways to quantify the effects, but they’ve not been done. Short of this, you should stick to a less categorical statement.

AUTHORS’ ANSWERS: this statement appears in the paragraph **Computational analysis of the primary targets of EES applied to the cervical spinal cord** of the **Results** section, which is devoted to describing the results of the numerical simulations of the study.

In the specific case of the statement mentioned by the reviewer, this statement follows from the results presented in **Figure 2c**, which shows that the estimated (simulated) recruitment of the Ia-fibers of the C6-root following stimulation targeting the C6-root reaches 100% at stimulation amplitudes for which the estimated recruitment for the other roots is still 0%. This indeed corresponds to the full and exclusive recruitment of the C6-root.

However, the statement does not intend to state that such full and exclusive recruitment of individual roots can be achieved in-vivo, but that we obtained it in our simulations. We modified the sentence to clarify this point

ACTION IN THE MANUSCRIPT: we modified the discussion to improve clarity of that sentence.

As expected, our results indicate that Ia-afferents running in the targeted root (C6) are recruited at significantly lower stimulation amplitudes than those running in the adjacent or more distal roots (C5, C7, C8 and T1, **Figure 2c**). **In other words, our simulations suggest that** near full and exclusive recruitment of individual roots can be achieved with individual monopolar lateral electrodes.

REVIEWER 2 | Comment 4: Show the same set of muscles in 8 a and b, as they are to be compared directly.

AUTHORS’ ANSWERS: We thank the reviewer for this suggestion. We agree that showing the same set of muscles makes the comparison easier and we implemented this change.

ACTION IN THE MANUSCRIPT: We updated Figure 8.

REVIEWER 2 | Comment 5: “...responses in the triceps muscle were enhanced during reaching and strongly suppressed during grasping and pulling.”

I don’t see this at all. Although its activation during pulling is small generally, Tri is clearly more active throughout in b than in a. What am I missing? Is this just a description of the normal suppression of triceps during elbow flexion, rather than a description of the effect of stimulation?

AUTHORS’ ANSWERS: we apologize with the reviewer. Indeed the wording of this sentence was inaccurate.

What we observed is that stimulus-triggered responses in TRI were *large* during reaching and *small* during grasping and pulling, which is shown by the new Figure 8b. Furthermore, when looking at the energy of the EMG signals, we observed that this energy was largely increased during reaching and only very weakly during grasping and pulling.

We modified the text to express these findings more clearly.

ACTION IN THE MANUSCRIPT: We updated the results to address the reviewer comments.

For example, responses in the triceps muscle were **large** during reaching and **small** during grasping and pulling (**Figure 8b**). **Accordingly, triceps' EMG energy was largely increased during reaching and only weakly during grasping and pulling (Figure 8c).**

REVIEWER 2 | Comment 6: The lower Tri trace in b is fairly confusing. It is not described in the text, and it makes a completely different point from all the other traces. If it is important, describe it and set it apart in the figure more clearly. If it is not, delete it.

AUTHORS' ANSWERS: We thank the reviewer for this comment. We acknowledge that the discussed trace was poorly described in the text. Nonetheless we believe that this plot is important in order to give a sense of how stimulation effects are modulated during the movement in an awake animal. We therefore modified Figure 8 in order to clarify how this plot is derived from the raw data showed in (a). Moreover made sure we would reference to it in the text, in order to make our point.

ACTION IN THE MANUSCRIPT: See new version of Figure 8

REVIEWER 2 | Comment 7: “Coherently with its position” > Consistent with its position

AUTHORS' ANSWERS: we thank the reviewer for noticing this bad wording.

ACTION IN THE MANUSCRIPT: we implemented the reviewer's suggestion in text.

REVIEWER 2 | Comment 8: Are the bins in c meant to align with the time axes of a and b? They don't quite, but it would be helpful if they did. C is described nowhere in the text. Add some comment.

AUTHORS' ANSWERS: We apologize with the reviewer; indeed the organization of Figure 8 was confusing. We now changed it to better show how the bins relate to each other.

ACTION IN THE MANUSCRIPT: See new version of Figure 8

REVIEWER 2 | Comment 9: Panel e adds very little beyond the muscle analysis. Is the rightmost panel intended to show the positive monotonic relation? If so, it is not clear how. There is no description of this in the legend and no direct reference in the text. I'd delete the entire analysis.

AUTHORS' ANSWERS: We agree with the reviewer. We now felt that this section not only was adding little but potentially deviating the entire goal of the paper. Therefore, following your suggestion and a discussion with reviewer 1 we removed the kinematic analysis.

ACTION IN THE MANUSCRIPT: Panel e has been removed.

REVIEWER 2 | Comment 10: Rather than this sentence:
Thus, the direct activation of the same pre-motor elements may modulate different motor nuclei than during resting conditions depending on the specific motor task and movement phase.

I'd suggest:

Thus, the direct activation of a given pre-motor element may modulate different motor nuclei during movement than at rest, in a task-dependent manner.

AUTHORS' ANSWERS: we thank the reviewer for his suggestion. We implemented a slightly different version of the suggested sentence in the text.

ACTION IN THE MANUSCRIPT:

Thus, the direct activation of **a given ensemble of** pre-motor elements may modulate different motor nuclei during **movement than at rest, in a task-dependent manner.**

REVIEWER 2 | Comment 11: and that current human electrode arrays[, with their medially-placed contacts,] are unfit to achieve these results.

AUTHORS' ANSWERS: we thank the reviewer for his suggestion. We implemented a slightly different version of the suggested sentence in the text.

ACTION IN THE MANUSCRIPT:

Our results indicate that lateral contacts are necessary to achieve this segmental selectivity and that current human electrode arrays, **with their medially-placed contacts and their short lengths,** are unfit to achieve these results.

REVIEWER 2 | Comment 12: "...might have given them a surplus of excitability..." Surplus means more than enough. I doubt this is what you mean. Perhaps, "given them greater sensitivity to the stimulation... albeit not as large as that of the larger diameter A-alpha-fibers"?

AUTHORS' ANSWERS: we thank the reviewer for his notification. We implemented a slightly different version of the suggested sentence in the text.

ACTION IN THE MANUSCRIPT: we modified the Discussion:

Thus, A β -fibers were more likely to possess a node directly below the electrode surface, which might have **contributed to their greater** excitability, albeit not as **much** as the larger diameters of the A α -fibers.

REVIEWER 2 | Comment 13: "this point deserves further investigations." > Investigation

AUTHORS' ANSWERS: we thank the reviewer for his notification.

ACTION IN THE MANUSCRIPT: we corrected the text accordingly.

REVIEWER 2 | Comment 14: There are many words that are inappropriately hyphenated: Proprio-spinal, trans-synaptic, neuro-technologies, downward-oriented, cross-sections, Ia-afferents, Ia-fibers, supra-threshold, motor-axons.

AUTHORS' ANSWERS: we thank the reviewer for his notification.

ACTION IN THE MANUSCRIPT: we removed the hyphens from "proprio-spinal", "neuro-technologies", "downward-oriented" and "motor-axons". However, we kept them in "trans-synaptic", "cross-sections", "Ia-afferents", "Ia-fibers" and "supra-threshold" because we found them hyphenated in multiple references.

Reviewers' Comments:

Reviewer #1:

Remarks to the Author:

The authors have satisfactorily addressed my concerns. I think the decision to remove the kinematics data is sensible, and the data generally support the key claims that the authors have clarified. Therefore I am happy to recommend publication at this stage.

Reviewer #2:

Remarks to the Author:

The authors have responded well to my remaining concerns.

COMPUTATIONAL AND EXPERIMENTAL ANALYSIS OF THE RECRUITMENT OF UPPER-LIMB MOTONEURONS WITH EPIDURAL ELECTRICAL STIMULATION OF THE PRIMATE CERVICAL SPINAL CORD

TO EDITOR: No further edits required by the reviewers

Reviewer #1

The authors have satisfactorily addressed my concerns. I think the decision to remove the kinematics data is sensible, and the data generally support the key claims that the authors have clarified. Therefore I am happy to recommend publication at this stage.

Reviewer #2

The authors have responded well to my remaining concerns.